**Registered Report**

# Psychological inoculation strategies to fight climate disinformation across 12 countries

**Tobia Spampatti** [1,2] ✉, **Ulf J. J. Hahnel** [1,3], **Evelina Trutnevyte** [4] & **Tobias Brosch** [1,2]

Decades after the scientific debate about the anthropogenic causes of climate change was settled, climate disinformation still challenges the scientific evidence in public discourse. Here we present a comprehensive theoretical framework of (anti)science belief formation and updating to account for the psychological factors that influence the acceptance or rejection of scientific messages. We experimentally investigated, across 12 countries ($N = 6,816$), the effectiveness of six inoculation strategies targeting these factors—scientific consensus, trust in scientists, transparent communication, moralization of climate action, accuracy and positive emotions—to fight real-world disinformation about climate science and mitigation actions. While exposure to disinformation had strong detrimental effects on participants' climate change beliefs ($\delta = -0.16$), affect towards climate mitigation action ($\delta = -0.33$), ability to detect disinformation ($\delta = -0.14$) and pro-environmental behaviour ($\delta = -0.24$), we found almost no evidence for protective effects of the inoculations (all $\delta < 0.20$). We discuss the implications of these findings and propose ways forward to fight climate disinformation.

The sixth report of the Intergovernmental Panel on Climate Change (IPCC) unequivocally declared that climate change is real and that humans are driving it[1,2]. Whereas 97–99% of climate scientists agree about the human causation of climate change[3–5], one third of the global population doubts or denies its anthropogenic roots[6–8]. This can be traced back to half a century of disinformation by the climate change countermovement, comprising fossil fuel corporations and their front groups, scientists-for-hire and lobbied politicians, who have contested climate science and are now delaying necessary climate mitigation actions[9–17]. This multi-million-dollar public relations effort[18–22] operates mainly via popular communication avenues[3,23,24], such as traditional[25–27] and social media[28–30], to shape climate discourse and political decision-making[17,29]. Their claims take up legitimate concerns that people express—such as high costs or uncertain efficacy of climate action—but qualify as disinformation because these concerns are intentionally distorted and amplified into misleading claims[31,32] such as bad-faith questioning of the scientific consensus[33], overemphasizing

the socio-financial burden of climate mitigation policies[14,34] and scaremongering citizens into inaction through climate doomism[14]. Unfortunately, climate disinformation can be more persuasive than scientifically accurate information[35–38] (see also ref. 39).

People process scientific messages not as neutral information processors but rather by weighing the messages against their prior convictions[40–45], against desired outcomes[46–49], against affective associations[50,51] and through the lens of their sociocultural and ideological contexts[52–55] (see reviews in refs. 56–60). When these psychological factors are misaligned with scientific information about climate change, antiscience beliefs fester[39,61] and become resistant to correction[57]. Two recent reviews offer distinct yet complementary perspectives on how (dis)information and (anti)science beliefs hinge on different communicational bases and psychological drivers. Philipp-Muller and colleagues[61] identified the different communicational bases on which (anti)science beliefs can build: the sources of scientific messages, the scientific messages themselves, the recipients of the scientific

[1]Swiss Centre for Affective Sciences, University of Geneva, Geneva, Switzerland. [2]Faculty of Psychology and Educational Sciences, University of Geneva, Geneva, Switzerland. [3]Faculty of Psychology, University of Basel, Basel, Switzerland. [4]Renewable Energy Systems, University of Geneva, Geneva, Switzerland. ✉e-mail: tobia.spampatti@unige.ch

**Table 1 | Comprehensive framework of (anti)science belief formation and updating**

| | | | Core communicational bases | | |
|---|---|---|---|---|---|
| | | | Sources of scientific messages | Scientific messages themselves | Recipients of scientific messages |
| **Psychological drivers** | Cognitive pathway | Driver | Consideration of scientific sources | Match/mismatch with prior beliefs | (Lack of) analytical thinking and/or deliberation |
| | | Proposed intervention | Scientific consensus inoculation | Transparent communication inoculation | Accuracy inoculation |
| | Socio-affective pathway | Driver | Trust in scientific sources | Match/mismatch with moral convictions | Emotional state during message processing |
| | | Proposed intervention | Trust inoculation | Moralization inoculation | Positive emotions inoculation |

The table shows the interplay between the communicational bases and psychological drivers of (anti)science belief formation and updating, and the theory-based psychological inoculations designed to address each entry point. Note that the boundaries between the cognitive and socio-affective pathways are permeable, and the effects of most interventions meant to address one pathway will very probably spill out to the other pathway of scientific (dis)information processing. For example, we consider the transparent communication inoculation to act on the cognitive driver 'match/mismatch with prior beliefs'; however, its effects can spill over towards the socio-affective driver 'trust in scientific sources'[112,113].

messages and the recipients' epistemic style. In parallel, Ecker and colleagues[57] grouped the psychological drivers influencing (dis)information belief formation and revision into cognitive and socio-affective drivers, depending on the psychological pathways they act on to facilitate or hinder belief formation and updating. Overall, both analyses affirmed that people's capacity and motivation to process information and disinformation—(dis)information henceforth—is conditional to the (mis)alignment of scientific information about climate change with specific communicational and/or psychological factors[57,61].

Here we adapt these factors to construct a comprehensive framework of (anti)science belief formation and updating (Table 1). In this framework, the processing of scientific (dis)information is mapped onto its core communicational bases[61]: sources, messages and recipients. These communicational bases are the entry points[62] where different psychological factors can influence (anti)science belief formation and updating through cognitive or socio-affective pathways[57].

Such a framework allows both the systematic mapping of the different entry points of scientific (dis)information, and the targeted, theory-based design of a comprehensive set of psychological intervention strategies using cognitive and socio-affective pathways to protect people from being influenced by disinformation. Among possible interventions, psychological inoculations have been identified as one of the most promising approaches to fighting climate disinformation[23,63]. They consist of pre-emptive warnings about incoming disinformation coupled with psychological resources[64]—counterarguments[38] and/or rhetorical techniques[65–67]—to resist disinformation[68]. We now review how each factor identified in the framework can engender acceptance or rejection of scientific messages, and we propose a set of theory-based psychological inoculations targeting these factors, with the aim of fighting scientific disinformation about climate change.

The first entry point to (anti)science belief formation and updating is the source of scientific messages about climate change. At the cognitive pathway level, it has been shown that the perception that the scientific community agrees about anthropogenic climate change provides diagnostic information that people can use to strengthen their acceptance of climate science. According to the gateway belief model[69], accurate information about the scientific consensus makes people more accepting of climate science and of climate actions[70]. Since the infamous 'Luntz memo'[71] coached Republican politicians to question the scientific consensus about climate change, countermovement actors have been painting the scientific community as divided and biased about the reality of climate change[17,33]. The result of this strategy has been that people neglect current scientific sources, perceiving the scientific consensus to be magnitudinally lower than the actual consensus (the false consensus effect[57,69]), climate science to be unsettled and climate action to be therefore unnecessary[58].

People can thus be psychologically inoculated with arguments explaining the scientific consensus to protect against disinformation at the source basis via the cognitive pathway[38]. At the socio-affective pathway level, trust in the sources of scientific messages is essential for increasing information processing and climate policy support[72–80]. People update their beliefs when scientific messages are delivered by trusted sources[73,81–83], whereas people who distrust mainstream and scientific information sources are more susceptible to misinformation and to holding wrong beliefs[84–86]. Moreover, trust in climate stakeholders moderates the association between believing in climate change and supporting mitigation policies such as carbon taxes: people who distrust political climate stakeholders oppose mitigation policies[87–89], whereas people who trust them support mitigation policies[75,76,87,89] (see also ref. 90). Scientists themselves are the most trusted sources of scientific information[78,91,92], and trust in science and scientists predicts support for climate mitigation behaviours more strongly than trust in other climate stakeholders[93,94] (see also ref. 72). Emphasizing the trustworthiness of scientists can make this trust more salient[80], potentially curbing disinformation uptake[95] and thus protecting against disinformation at the source basis via the socio-affective pathway. Two non-peer-reviewed, preregistered studies support this idea, as pre-emptively making trust in key stakeholders of the energy transition salient protected support for renewable energy from multiple negative persuasive attacks[96].

The second entry point to (anti)science belief formation and updating is the scientific message itself. People process scientific (dis)information on the basis of the (un)intuitiveness of the messages[39,57,97] and the (mis)alignment with their own worldviews, moral values and political ideologies[98–102]. At the cognitive pathway level, when people detect a conflict between their prior beliefs and incoming scientific messages[43], they resist scientific information by generating counterarguments[103–107]. Unaddressed counterarguments can cement policy opposition[108,109] (cf. ref. 110), especially when people's legitimate concerns—such as the costs of climate actions[111]—are turned into exaggerated counterarguments to stifle climate policies[14,33]. To protect against disinformation at the message basis via the cognitive pathway, counterarguing can be addressed by transparently communicating the pros and cons of debated policies[112–116]. Transparently addressing concerns while highlighting positive outcomes was recently found to increase COVID-19 vaccination intentions and trust in the source of the transparent communication, more than messages ignoring vaccination concerns; the elicited changes were moreover resistant to a subsequent conspiracy message attacking the vaccine[112]. At the socio-affective pathway level, scientific messages are resisted when they are misaligned with people's moral values[117,118]. Multiple studies show that when people's moral convictions are questioned by scientific

messages, "moral convictions have the power to bend people's factual beliefs, trust in authorities, and evaluations of procedures"[119] (p. 87) (see also refs. [120–123]), which may result in the rejection of scientific evidence. To protect against disinformation at the message basis via the socio-affective pathway, one can link the importance of climate action to a diversity of worldviews and moral orientations by framing scientific messages in moral terms (for example, refs. [124–127]). Linking climate action to morality can moreover increase the likelihood that people will take action[128]: emblematically, climate activist Greta Thunberg cited moral conviction as her primary driver for the climate strike movement[118].

The third entry point to (anti)science belief formation and updating is the message recipient. People who rely on intuitive thinking are more likely to believe and share misinformation[129–131], whereas people who rely on reflective, deliberative thinking tend to hold more accurate beliefs[132–135] (see review in ref. [43]). According to this research, most people are accurate in determining the truthfulness of information when making judgements deliberately[134] (see also refs. [109,136]). However, they are easily distracted away from deliberation, thus engaging with (dis)information without actively considering their factual basis[137]. To protect against disinformation at the recipient basis via the cognitive pathway, people can be directed to thinking deliberately by prompting them to evaluate incoming information by its factual accuracy[138,139]. Untested in the climate domain (but see a similar intervention in ref. [140]), accuracy prompts robustly decreased the influence of misinformation on political belief and fake news sharing[134] in the lab, in the field[135] and across countries[141]. At the socio-affective level, the processing of scientific (dis)information is influenced by the emotional state of the recipient[142]. Emotions are a filter that guides people towards relevant and valued information in a noisy environment[143–145], and their motivational properties direct and support individuals' behaviour[146]. On the one hand, correlational evidence suggests that emotion-laden misinformation spreads more widely in social networks[147], and that people tend to believe misinformation more when it contains emotional content[148,149]. On the other hand, emotions have been found to foster belief updating and climate-related behaviour[50,144,150–152]. Positive emotions motivate discounting of counterattitudinal information[40] and have been suggested as an antidote to overcome a lack of motivation to parse misinformation[153]. Moreover, multiple recent reviews[144,146,154,155] argue that the anticipation and experience of positive emotions elicited by acting pro-environmentally[144,154–158] increase pro-environmental behavioural intentions as well as actual behaviour[158–161]. To protect against disinformation at the recipient basis via the socio-affective pathway, the saliency of experienced positive emotions in the context of climate action can be increased, which should increase resistance to disinformation as well as the likelihood of acting pro-environmentally.

In summary, here we integrated previous analyses into a comprehensive framework of the communicational and psychological factors influencing (anti)science belief formation and updating. On the basis of this integrated, theory-driven perspective, we introduce a set of broad-spectrum psychological inoculations to protect against climate disinformation that act on each of the identified entry points and pathways:

- A scientific consensus inoculation explaining that among climate scientists there is virtually no disagreement that humans are causing climate change
- A trust inoculation making salient the trustworthiness of IPCC scientists in terms of climate change science and mitigation actions
- A transparent communication inoculation addressing the pros and cons of climate mitigation action
- A moralization inoculation creating a stronger link between climate mitigation actions and the diversity of moral convictions

- An accuracy inoculation reorienting participants towards judging incoming information by its factual accuracy
- A positive emotions inoculation eliciting positive emotions towards climate mitigation actions

We investigated the effectiveness of these six broad-spectrum psychological inoculation strategies to protect against climate disinformation in a multi-country, multi-intervention study, against a sequence of 20 real climate disinformation messages spread by members of the climate change countermovement on the social media platform Twitter. We assessed the protective effect of the inoculations on participants' climate change beliefs[162], appraisal of climate mitigation action and truth discernment capacity—that is, their capacity to correctly distinguish between true and false information[163]. We moreover investigated whether the protective effects of the psychological inoculations extend to actual pro-environmental behaviour[164]. The participants were presented with 20 real climate disinformation statements that were selected on the basis of an initial validation study ($N = 504$, available at https://osf.io/m58zx). The participants saw multiple disinformation statements to assess whether the psychological inoculations are capable of protecting against not only one but multiple occurrences of climate disinformation (which mirrors the preponderance of climate disinformation in certain epistemic communities[30]). After each disinformation message, the participants rated their current affect towards climate actions (we measured affect towards, rather than political support for, climate mitigation actions because affective reactions predate and motivate policy appraisals and climate-friendly behaviour[50,51,144]). After having viewed all 20 disinformation statements, the participants also reported their perceptions concerning the reality, causes and consequences of climate change[162], performed a version of a validated pro-environmental behaviour task with actual environmental consequences[164], and performed a truth discernment task with true and false climate statements. Compared with a passive disinformation control condition where participants were only confronted with the disinformation, we expected the inoculations to significantly protect participants' affect towards climate action ($H_{1A,B}$), with a treatment effect bigger than that for the 'standard approach' of fact-checking political topics[165]. We moreover expected the protective effect to extend to people's climate change beliefs ($H_2$), performance in the pro-environmental behaviour task ($H_3$) and truth discernment capacity ($H_4$). We collected responses from 12 countries across the globe, 7 of which are non-WEIRD (Western, Educated, Industrialized, Rich and Democratic), to be able to make stronger claims about the generalizability of the six psychological inoculations[57,60,166,167]. We furthermore investigated treatment heterogeneity[168] by assessing the effectiveness of the inoculations depending on their thematic match with the climate disinformation statements (for example, testing whether the scientific consensus inoculation protects especially well against disinformation that targets the scientific consensus), and depending on participants' tendency for intuitive/deliberative thinking. Not only has this tendency previously been shown to directly influence belief in (anti)scientific topics[43] and to moderate accuracy prompting[135], but it may moreover underlie people's overall tendency to rely on the socio-affective (for intuitive thinkers) or the cognitive (for deliberative thinkers) pathways to enact belief updating and revision[43,57]. We hypothesized this tendency to be a moderator depending on its match with the inoculation pathway: cognitive-based inoculation would be more effective for people with a tendency for deliberative thinking, whereas socio-affective-based inoculation would be more effective for people with a tendency for intuitive thinking ($H_{secondary 1}$). The aim of the study was to introduce interventions that can comprehensively address the communicational bases and the main psychological drivers of (anti)science belief formation and updating and thus to provide new interventions in the fight against climate disinformation.

**Table 2 | Text of the six inoculations**

| Cognitive inoculation | Socio-affective inoculation |
|---|---|
| Scientific consensus inoculation | Trust inoculation |
| When confronted with such misleading information about the science of climate change and the actions to mitigate it, remember that the IPCC, the most comprehensive review on the scientific agreement behind climate change and climate action, found that among thousands of climate scientists with the highest degrees of expertise 'there is virtually no disagreement that humans are causing climate change'. Studies have shown that the consensus about anthropogenic climate change among expert scientists ranges from 97% to 99%. IPCC scientists from all cultural backgrounds and nations stated in the report that 'It is unequivocal that human influence has warmed the atmosphere, ocean and land' and they are in agreement that urgent climate action is needed for a better planet and society. | When confronted with such misleading information about the science of climate change and the actions to mitigate it, remember that the IPCC is the most authoritative scientific body in the world assessing the knowledge about climate change and climate action and that the majority of citizens of multiple countries trust scientists. Climate scientists have the highest degrees of expertise and are committed to open and transparent review by other scientists and governments around the world, and value rigorous and balanced scientific information above all else. IPCC scientists come from all cultural backgrounds and nations, to reflect a diverse range of views and expertise in their work and to ensure an objective and complete assessment of the scientific evidence about climate change, to recommend actions and policies for a better planet and society. |
| Transparent communication inoculation | Moralization inoculation |
| When confronted with such misleading information about the science of climate change and the actions to mitigate it, remember that the IPCC scientists are open about the fact that climate actions will require substantial funding and a significant overhaul of our way of life to keep our planet livable. They also disclosed that there is some uncertainty about if and how these climate actions may reduce our quality of life, but they still concluded with confidence that limiting irreversible climate-induced risks with climate action is less risky than not acting at all. Acting is hard, they admit, but it is through these scientifically supported actions that we can protect our planet, reduce inequality, and generate sustainable growth. | When confronted with such misleading information about the science of climate change and the actions to mitigate it, remember that the IPCC scientists provide valuable and authoritative advice about actions that our communities and nations must take to responsibly keep our planet livable for us and for future generations. As citizens of this earth, we have a moral responsibility to protect our homeland and our community from climate-induced risks and harms, and to stop defiling our pristine natural environment. Through these scientifically supported actions, we can protect our planet, create a more just and fair society with decent living conditions for everyone, and generate sustainable growth beneficial for us, our nations, the world, and generations to come. |
| Accuracy inoculation | Positive emotions inoculation |
| When confronted with such misleading information about the science of climate change and the actions to mitigate it, remember that it is important to be able to accurately recognize these misinformation to avoid being influenced by them. One good strategy to distinguish between good and bad information is to ask yourself: 'do I think this information is accurately describing the state of the science of climate change? Is this information not at all accurate, not very accurate, somewhat accurate, or very accurate?'. When you evaluate the information you see on any media about climate change, think about this accuracy question to get in the right frame of mind. | When confronted with such misleading information about the science of climate change and the actions to mitigate it, remember that climate actions are vital actions that will keep our planet livable for the next generation. Actions such as eating delicious and healthy meals with a lower carbon footprint or taking a bike ride instead of getting stuck in traffic are scientifically supported ways to make you happier and more fulfilled in your daily life. When you evaluate the information you see on any media about climate change, imagine the positive changes you can create with climate action, and think about how good you will feel when doing so. |

Cross-condition differences are underlined.

## Results

### Manipulation check

All psychological inoculations (Table 2) except the scientific consensus inoculation (equivalence test: $t(1,651.06) = -2.527$; $P = 0.006$; $\delta = 0.08$; 90% confidence interval (CI), $(-0.01, 0.23)$) increased participants' motivation to resist persuasion. Participants who received the trust inoculation ($t(1,704.11) = 2.233$; $P = 0.03$; $\delta = 0.11$; 95% CI, $(0.02, 0.26)$), the transparent communication inoculation ($t(1,697.49) = 2.844$; $P = 0.005$; $\delta = 0.14$; 95% CI, $(0.05, 0.29)$), the moralization inoculation ($t(1,730) = 5.333$; $P < 0.001$; $\delta = 0.26$; 95% CI, $(0.21, 0.45)$), the accuracy inoculation ($t(1,687.69) = 3.844$; $P < 0.001$; $\delta = 0.19$; 95% CI, $(0.12, 0.35)$) or the positive emotions inoculation ($t(1,702.58) = 3.258$; $P < 0.001$; $\delta = 0.16$; 95% CI, $(0.08, 0.32)$) reported more motivation to resist persuasion than participants in the passive control condition.

### Effects of the climate disinformation statements

Compared with the pure control condition, participants in the passive control condition (who received 20 climate disinformation statements without any psychological inoculation; Table 3) reported significantly less positive affect towards climate mitigation action ($t(1,676.83) = -6.774$; $P < 0.001$; $\delta = -0.33$; 95% CI, $(-10.48, -5.77)$) and significantly lower belief in the reality of climate change ($t(1,689.62) = -3.990$; $P < 0.001$; $\delta = -0.19$; 95% CI, $(-0.28, -0.10)$) and in its anthropogenic causes ($t(1,705.95) = -2.496$; $P = 0.013$; $\delta = -0.12$; 95% CI, $(-0.21, -0.03)$), but not in the negativity of its consequences ($t(1,708.79) = -1.942$; $P = 0.052$; $\delta = -0.09$; 95% CI, $(-0.18, 0.001)$; equivalence test: $t(1,708.79) = 2.205$; $P = 0.014$; 90% CI, $(-0.01, -0.16)$). Their overall truth discrimination was significantly worse ($t(1,717.26) = -2.877$; $P = 0.004$; $\delta = -0.14$; 95% CI, $(-0.52, -0.10)$). Exposure to climate disinformation reduced pro-environmental behaviour, as participants in the passive control condition completed significantly less pages of the pro-environmental behaviour task ($t(1,713.68) = -5.030$; $P < 0.001$; $\delta = -0.24$; 95% CI, $(-0.80, -0.39)$; $H_{control 1A–D}$ supported).

### Inoculation effects on affect–preregistered analyses

The protective effects of the inoculations on affect towards climate mitigation action after the 20 climate disinformation statements was, if present, significantly smaller than $\delta = 0.20$ (Supplementary Fig. 1; $H_{1A}$ not supported). There was suggestive evidence that the positive emotions inoculation had a small protective effect compared with the passive control condition, but the effect was significantly smaller than $\delta = 0.20$ (one-tailed $t$-test: $t(1,702.7) = 1.862$; $P = 0.03$; 95% CI, $(0.28, \infty)$; equivalence test: $t(1,702.68) = -2.243$; $P = 0.012$; 95% CI, $(0.28, 4.54)$). Comparisons between the passive control condition and the other psychological inoculations yielded no significant differences for the scientific consensus inoculation (one-tailed $t$-test: $t(1,667) = 0.854$; $P = 0.43$; 95% CI, $(-1.05, \infty)$; equivalence test: $t(1,667.05) = -3.239$; $P < 0.001$; 90% CI, $(-1.46, 3.72)$), the trust inoculation (one-tailed $t$-test: $t(1,706.9) = 1.011$; $P = 0.16$; 95% CI, $(-0.82, \infty)$; equivalence test: $t(1,705.36) = 3.272$; $P < 0.001$; 90% CI, $(-1.02, 3.69)$), the transparent communication inoculation (one-tailed $t$-test: $t(1,685.78) = -0.2899$; $P = 0.61$; 95% CI, $(-2.53, \infty)$; equivalence test: $t(1,685.78) = -3.769$; $P < 0.001$; 90% CI, $(-1.03, 3.29)$), the moralization inoculation (one-tailed $t$-test: $t(1,730) = 1.533$; $P = 0.06$; 95% CI, $(-0.15, \infty)$; equivalence test: $t(1,730) = -2.630$; $P = 0.004$; 90% CI, $(-0.15, 4.12)$) or the accuracy inoculation (one-tailed $t$-test: $t(1,683.2) = -0.150$; $P = 0.56$; 95% CI, $(-2.36, \infty)$; equivalence test: $t(1,683.24) = 3.960$; $P < 0.001$; 90% CI, $(-2.36, 1.97)$).

The multilevel model analyses of affect towards mitigation action across the disinformation provision mirrored the previous finding: there was no evidence of a significant interaction between trial and condition ($F(6, 112,994) = 1.4749$, $P = 0.18$; Fig. 1 and

**Table 3 | The 20 climate disinformation statements and their coding**

| Coding | Disinformation tweet |
| --- | --- |
| Science_1 | As more wind and solar are added they raise electricity prices and destabilize electric grids. Because they are part-time unreliable weather dependent sources. We want full-time electricity. Not part-time like third world countries. All for silly expensive net zero. CA pays more. |
| Science_2 | The current exceptional warming and cooling your seeing is due to the location of the Jet Stream. It's become very wavy due to the lack of Solar Energy going into the Oceans and nothing to do with Man Made CO2 |
| Science_3 | Today's 'global warming' is estimated to be an otherwise unmeasurable 0.4°C (0.72°F) over the 1979-2000 average... despite 50% of all manmade emissions. No 2022 weather event was unprecedented or can be blamed on CO2 emissions. |
| Science_4 | This is a portrait of climate fraud, posturing as the saviours of the world. They are a breed of crooks, getting rich by ripping off gullible western nations. The UN led climate hoax has been running since 1988. They want us to believe a pack of lies about earth's climate. |
| Science_5 | Too often, academic reports on climate use highly skewed data that seem to have been carefully selected to support aggressive environmental regulations. One recent and much-cited Lancet report appears deliberately deceptive. |
| Science_6 | The climate hoax devised by the UN, supported by rich elitists is endorsed by our treacherous leaders is an attack on freedoms & rights. Climate cultism is a form of global self hatred. It aims to punish western nations by transferring huge reparations to the developing world. |
| Science_7 | Top NASA Climate Modeler Admits Predictions Are 'Mathematically Impossible' |
| Science_8 | Lots of links of studies of the Medieval Warm Period that climate science deniers (alarmists) want to pretend did not exist. Because there is no explanation for natural warming during this time. Studies point out temp was warmer back then, than now. |
| Science_9 | According to global warming theory the poles should warm significantly if carbon dioxide is driving temperatures Just the opposite is occurring in the southern hemisphere. |
| Science_10 | The evidence for manmade climate change is so thin they cannot debate it. They hide behind the lie of consensus. There is no room for consensus in science. The basis is a provable hypothesis. There is not a single peer reviewed study that proves manmade CO2 is causing warming. |
| Action_1 | At Climate Summit, Elites Chow Down on Gourmet Meats While Telling Us to Eat Bugs |
| Action_2 | FACT CHECK Results of the Biden administration's extreme climate agenda cutting emissions by 44% by 2030. Annual Jobs Lost: 1.2 MILLION. Lost Economic Growth: $7.7 TRILLION. Increase in Electric Bills: 23% Increase in Gas Prices: 2$ PER YEAR |
| Action_3 | The war on 'fossil fuels' is absurd considering the vast fields of coal/oil/gas everywhere on earth. The mantle is brimming over with it. A United Nations bid for control, cash & power has led to an energy crisis that looms as the biggest self-inflicted disaster in human history. |
| Action_4 | Death and privation caused by the lack of affordable energy caused by Green Energy policies will not affect the Elites at all. They want us to eat bugs, do a lot less as they carry on with their lives just as they are doing now. Climate scamsters. They should lead by example. |
| Action_5 | You are lying. Fossil fuels gave us cheap energy for decades so billions live longer healthier happier lives. Many technologies like carbon capture, filters fuel additives etc reduces emissions. Banning fossil fuels is creating fuel poverty and harming people |
| Action_6 | Energy literacy starts with the knowledge that renewable energy is only intermittent electricity generated from unreliable breezes and sunshine, as wind turbines and solar panels cannot manufacture anything for the 8 billion on this planet. |
| Action_7 | Imagine sacrificing 500 high-paying coal jobs, ranging up to $60,000/yr, for the climate hoax. Even if you believe in the hoax, global emissions are up 5% from pre-pandemic levels -- 90% because of China. Emissions from a single mine are insignificant. |
| Action_8 | Europe's transition to renewable energy and net zero carbon is not working, except to make life hard on average European citizens. |
| Action_9 | Willfully-blind ignorance about the consequences of [the rush to green policies] – deep recessions, broken societies and millions more going hungry – doesn't make them any less immoral. The road to hell is paved with good intentions. Bingo. |
| Action_10 | Solar and wind are far more expensive than established reliable stable secure electricity from pure hydro coal gas nuclear. That's why your shift to unreliable, unstable, expensive solar and wind; is devastating families; and exporting manufacturing jobs |

To prevent climate change countermovement actors from understanding the net persuasive appeal of each disinformation statement, the identifying numbers of each statement differ from the identifying numbers in the collected data. The correct matching is known only to the authors.

Supplementary Table 1; $H_{IB}$ not supported) or of a main effect of condition ($F(6, 6,978) = 1.9400$, $P = 0.07$; Fig. 2). Supplementary control analyses provided no evidence that these non-effects were dependent on matches between specific inoculations and climate disinformation statements (Supplementary Tables 2–4).

**Effects of the inoculations on affect towards climate mitigation action−exploratory analyses**

As psychological inoculations are usually tested against a single disinformation statement, not multiple ones[38,65], we exploratorily tested whether the psychological inoculations protected participants' affect towards climate mitigation action against the influence of the very first climate disinformation statement only. Except for the participants inoculated with the transparent communication inoculation, all other inoculated participants reported significantly more positive affect than the participants in the passive control condition. Participants in the passive control condition showed a significant decrease in their affect towards climate mitigation from the pre-intervention level after the first climate disinformation statement (paired $t$-test:

$t(852) = -3.316$; $P < 0.001$; $\delta = -0.11$; 95% CI, ($-3.03$, $-1.02$)). In contrast, participants in all inoculation conditions reported their affect towards climate mitigation action as unmoved from the pre-intervention level after the first climate disinformation statement (scientific consensus inoculation, $t$-test: $t(1,674.5) = 2.526$; $P = 0.012$; $\delta = 0.12$; 95% CI, (0.67, 5.30); paired equivalence test: $t(823) = -5.608$; $P < 0.001$; 90% CI, ($-1.28$, 1.09); trust inoculation, $t$-test: $t(1,706.63) = 2.150$; $P = 0.032$; $\delta = 0.10$; 95% CI, (0.22, 4.82); paired equivalence test: $t(850) = 5.010$; $P < 0.001$; 90% CI, ($-1.76$, 0.57); transparent communication inoculation, $t$-test: $t(16,897.65) = 1.935$; $P = 0.053$; $\delta = 0.09$; 95% CI, ($-0.03$, 4.56); paired equivalence test: $t(846) = 4.336$; $P < 0.001$; 90% CI, ($-2.07$, 0.11); moralization inoculation, $t$-test: $t(1,725.49) = 3.000$; $P = 0.003$; $\delta = 0.14$; 95% CI, (1.20, 5.75); paired equivalence test: $t(878) = -4.815$; $P < 0.001$; 90% CI, ($-1.90$, 0.37); accuracy inoculation, $t$-test: $t(1,687.71) = 2.075$; $P = 0.042$; $\delta = 0.10$; 95% CI, ($-0.09$, 4.73); paired equivalence test: $t(836) = 4.525$; $P < 0.001$; 90% CI, ($-1.98$, 0.48); positive emotions inoculation, $t$-test: $t(1,699.95) = 3.432$; $P < 0.001$; $\delta = 0.17$; 95% CI, (1.70, 6.24); paired equivalence test: $t(845) = 4.911$; $P < 0.001$; 90% CI, ($-0.43$, 1.53)). These exploratory analyses suggest that the inoculations were able

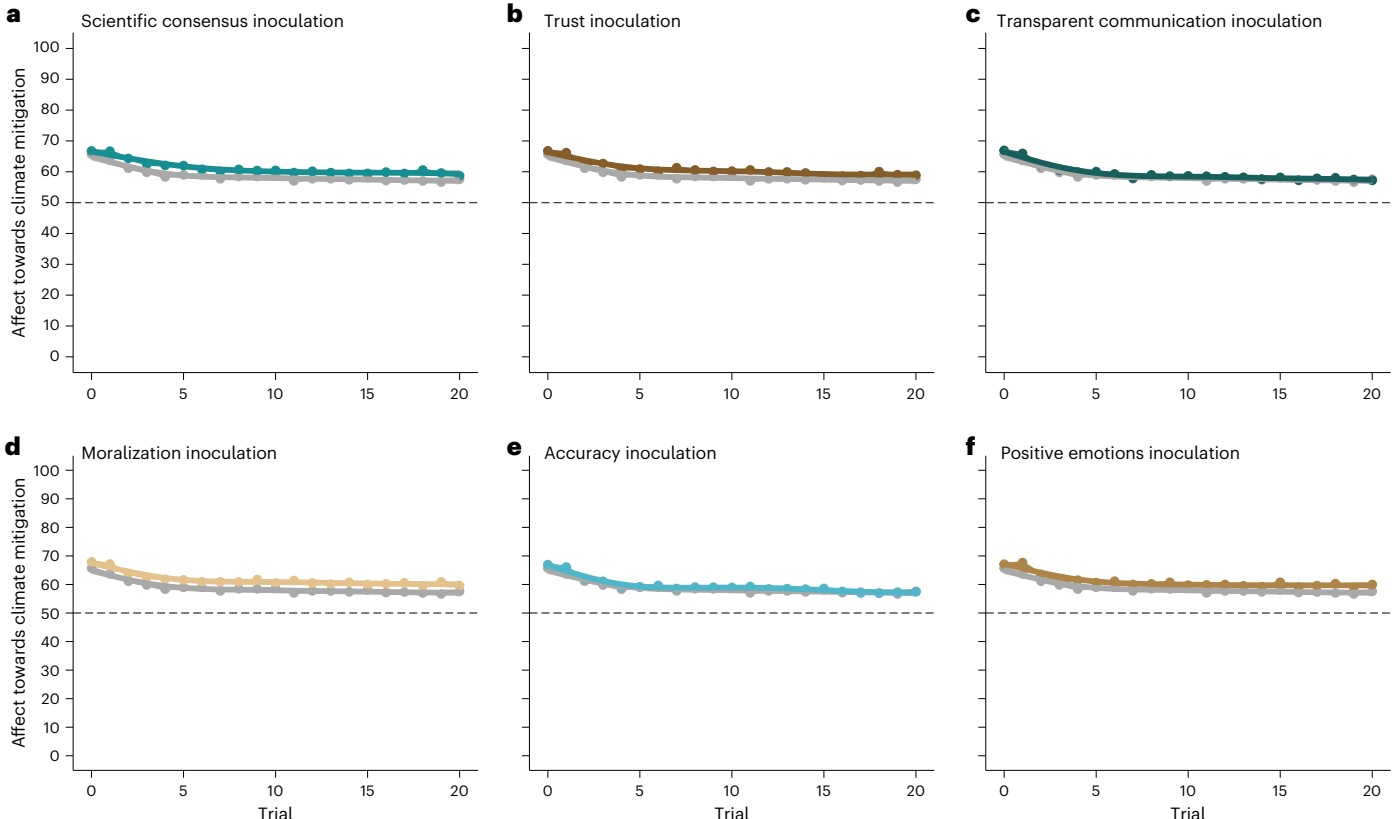

**Fig. 1 | Mean affect towards climate mitigation action across the provision of 20 climate disinformation statements, by condition ($N = 6816$). a,** Contrast between the scientific consensus inoculation and passive disinformation control conditions: $t_{\text{two-sided}}(6,978) = 2.550$; $P = 0.01$; $\beta = 2.78$; 95% CI, (0.64, 4.87). Two-way interaction between conditions and trial: $t_{\text{two-sided}}(113,000) = -1.493$; $P = 0.14$; $\beta = -0.04$; 95% CI, (−0.01, 0.01). **b,** Contrast between the trust inoculation and passive disinformation control conditions: $t_{\text{two-sided}}(6,978) = 2.130$; $P = 0.033$; $\beta = 2.28$; 95% CI, (0.18, 4.38). Two-way interaction between conditions and trial: $t_{\text{two-sided}}(113,000) = -0.976$; $P = 0.33$; $\beta = -0.03$; 95% CI, (−0.08, 0.03). **c,** Contrast between the transparent communication inoculation and passive disinformation control conditions: $t_{\text{two-sided}}(6,978) = 0.928$; $P = 0.35$; $\beta = 1.00$; 95% CI, (−1.11, 3.10). Two-way interaction between conditions and trial: $t_{\text{two-sided}}(113,000) = -1.943$; $P = 0.052$; $\beta = -0.06$; 95% CI, (−0.11, 0.001). **d,** Contrast between the moralization inoculation and passive disinformation control conditions: $t_{\text{two-sided}}(6,978) = 2.517$; $P = 0.011$; $\beta = 2.68$; 95% CI, (0.59, 4.76). Two-way interaction between conditions and trial: $t_{\text{two-sided}}(113,000) = -0.114$; $P = 0.91$; $\beta = -0.003$; 95% CI, (−0.06, 0.05). **e,** Contrast between the accuracy inoculation and passive disinformation control conditions: $t_{\text{two-sided}}(6,978) = 1.080$; $P = 0.033$; $\beta = 1.16$; 95% CI, (−0.95, 3.27). Two-way interaction between conditions and trial: $t_{\text{two-sided}}(113,000) = -1.985$; $P = 0.047$; $\beta = -0.06$; 95% CI, (−0.11, −0.001). **f,** Contrast between the positive emotions inoculation and passive disinformation control conditions: $t_{\text{two-sided}}(6,978) = 2.339$; $P = 0.02$; $\beta = 0.01$; 95% CI, (0.41, 4.61). Two-way interaction between conditions and trial: $t_{\text{two-sided}}(113,000) = -0.280$; $P = 0.78$; $\beta = -0.01$; 95% CI, (−0.06, 0.05). Each panel represents one condition and its contrast with the passive control condition (shown in dark grey). The y axis represents mean affect towards climate mitigation action, with values higher than 50 related to feeling more positively towards climate mitigation action, and values lower than 50 related to feeling more negatively towards climate mitigation action. The dashed line represents the 'neutral' anchor point (affect = 50). The x axis represents the trial number, with trial = 0 representing affect pre-intervention, and the numbers from 1 to 20 representing each climate disinformation statement received. The light grey bands represent the mean-centred standard errors produced by model fitting with a GAM function. Colour palette by the MetBrewer package[169]. The full-sized figures are in the Supplementary Information, Supplementary Figs. 2–7.

to protect participants at least against the negative impact of the first climate disinformation statement.

### Inoculation effects on climate change beliefs—preregistered analyses

Belief in climate change was generally high across the 12 countries ($\text{mean}_{\text{belief}} = 4.04 \pm 0.85$ on a scale from 1 to 5; Fig. 2a–c). Preregistered multilevel models did not find evidence that inoculated participants believed more in the reality of climate change (main effect of condition: $F(6, 5,901.1) = 1.1460$, $P = 0.33$), in its anthropogenic causes (main effect of condition: $F(6, 5,901.1) = 0.3824$, $P = 0.89$) or in the negativity of its consequences (main effect of condition: $F(6, 5,901.1) = 0.2911$, $P = 0.94$) than participants in the passive control conditions (the full model results are available in Supplementary Table 5). Equivalence tests confirmed that, if present at all, any effect of the inoculations on the overall belief in climate change was smaller than $\delta = 0.20$ (H$_2$ not supported) (scientific consensus inoculation: $t(1,670.08) = -3.971$;

$P < 0.001$; 90% CI, (−0.08, 0.09); trust inoculation: $t(1,705.17) = -4.038$; $P < 0.001$; 90% CI, (−0.08, 0.9); transparent communication inoculation: $t(1,694.76) = -3.413$; $P < 0.001$; 90% CI, (−0.10, 0.04); moralization inoculation: $t(1,694.76) = -3.413$; $P = 0.002$; 90% C(−0.02, 0.12); accuracy inoculation: $t(1,686.82) = -3.727$; $P < 0.001$; 90% CI, (−0.05, 0.09); positive emotions inoculation: $t(1,702.63) = -3.817$; $P < 0.001$; 90% CI, (−0.06, 0.08)).

### Inoculation effects on behaviour—preregistered analyses

Participants' engagement overall resulted in 10,969 trees being planted. The multilevel model predicting the performance in the pro-environmental behaviour task[164] showed a main effect of condition ($\chi(6) = 17.074$, $P = 0.009$) but yielded no evidence that inoculated participants accurately completed more task pages than participants in the passive control condition (the contrasts between conditions and the passive control were all not significant; Fig. 2e and Supplementary Table 6). Following the preregistration, upon visual inspection we

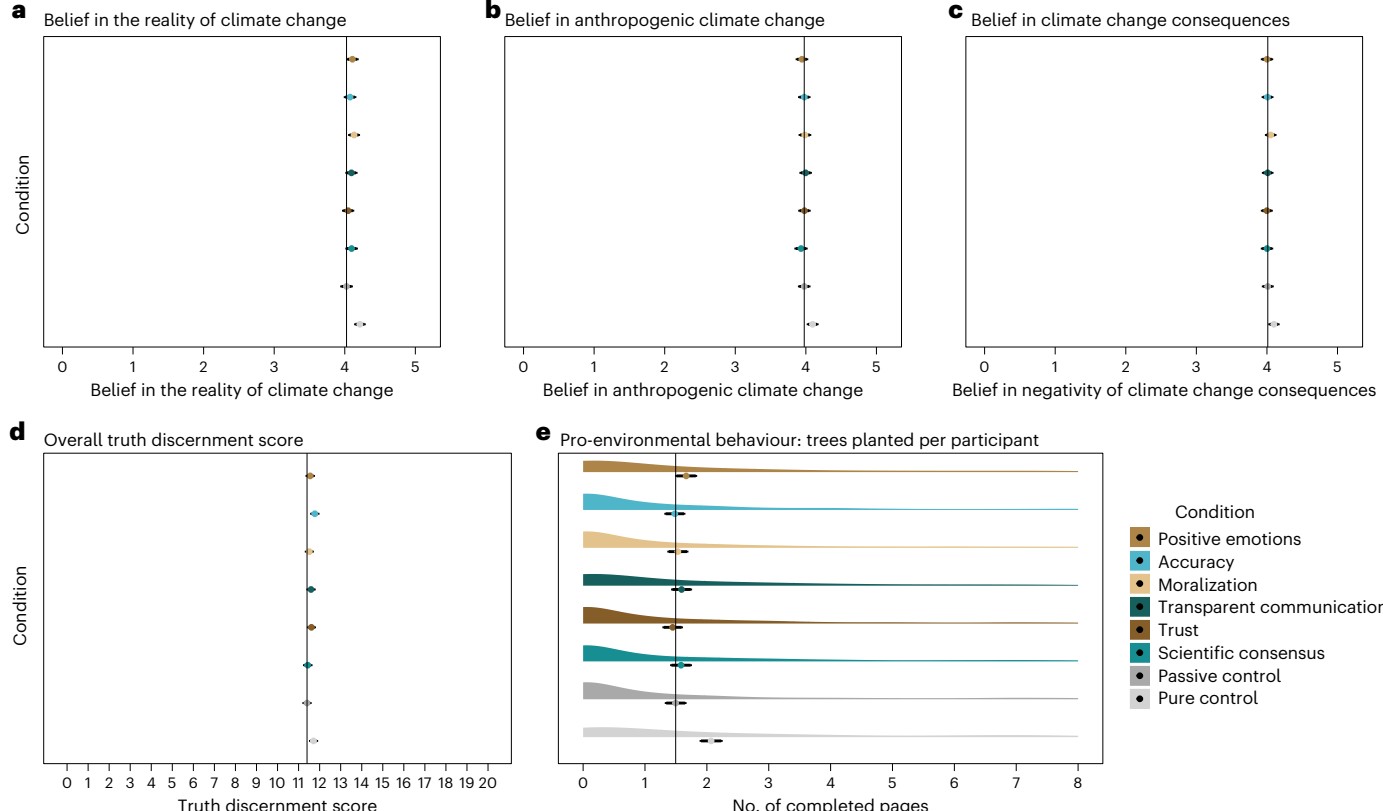

**Fig. 2 | Variables of interest, by condition (*N* = 6,816). a**, Belief in the reality of climate change[162] (*N* = 6,816). The *x* axis represents mean belief in climate change. Passive disinformation control condition: two-sided *t*-test; $t(1,689.62) = -3.990$; $P < 0.001$; $\delta = -0.19$; 95% CI, $(-0.28, -0.10)$. Contrast between conditions: $F(6, 5,901.1) = 1.1460$, $P = 0.33$. **b**, Belief in the anthropogenic nature of climate change[162]. The *x* axis represents mean belief in anthropogenic climate change. Passive disinformation control condition: two-sided *t*-test; $t(1,705.95) = -2.496$; $P = 0.013$; $\delta = -0.12$; 95% CI, $(-0.21, -0.03)$. Contrast between inoculation and passive disinformation control conditions: $F(6, 5,901.1) = 0.3824$, $P = 0.89$. **c**, Belief in the negativity of the consequences of climate change[162] (*N* = 6,816). The *x* axis represents mean belief in the negativity of climate change consequences. Passive disinformation control condition: equivalence test; $t(1,708.79) = 2.205$; $P = 0.014$; 90% CI, $(-0.01, -0.16)$. Contrast between conditions: $F(6, 5,901.1) = 0.2911$, $P = 0.94$. **d**, Climate truth discrimination score[163] (*N* = 6,816). The *x* axis represents the mean number of correct categorizations of true and false statements about climate change. Passive disinformation control condition: two-sided *t*-test; $t(1,717.26) = -2.877$; $P = 0.004$; $\delta = -0.14$; 95% CI, $(-0.52, -0.10)$. Accuracy inoculation: one-sided *t*-test; $t(5,936.2) = 3.360$; $P < 0.001$; $\beta = 0.36$; 95% CI, $(0.15, 0.57)$. Scientific consensus inoculation:

equivalence test; $t(1,670.06) = -3.782$; $P < 0.001$; 90% CI, $(-0.15, 0.22)$. Trust inoculation: equivalence test; $t(1,705.24) = -2.195$; $P = 0.014$; 90% CI, $(-0.03, -0.39)$. Transparent communication inoculation: equivalence test; $t(1,697.78) = -2.387$; $P = 0.009$; 90% CI, $(-0.01, -0.37)$. Moralization inoculation: equivalence test; $t(1,726.29) = -3.151$; $P = 0.001$; 90% CI, $(-0.07, 0.29)$. Positive emotions inoculation: equivalence test; $t(1,701.69) = -2.759$; $P = 0.003$; 90% CI, $(-0.03, 0.37)$. **e**, WEPT performance[164]; the response distribution is shown to highlight the flooring effect (*N* = 6,816). The *x* axis represents the mean number of WEPT pages participants completed with 90% accuracy (a preregistered inclusion criterion)—that is, trees planted per participant. Passive disinformation control condition: two-sided *t*-test; $t(1,713.68) = -5.030$; $P < 0.001$; $\delta = -0.24$; 95% CI, $(-0.80, -0.39)$. Contrast between conditions: $\chi(6) = 11.7805$, $P = 0.07$. In each panel, the *y* axis represents the experimental conditions: pure control condition (light grey), passive control condition (dark grey), scientific consensus inoculation (light green), trust inoculation (dark brown), transparent communication inoculation (dark green), moralization inoculation (gold), accuracy inoculation (light blue) and positive emotions inoculation (light brown). The error bars represent the mean-centred 95% CIs. Colour palette by the MetBrewer package[169].

identified a flooring effect of the Work for Environmental Protection Task (WEPT) response distribution. Model diagnostics confirmed that the distribution was zero-inflated[169,170] ($P < 0.001$). We thus conducted a supplementary multilevel with a zero-inflated intercept ($z = -4.02$; $P < 0.001$; $\beta = -0.12$; 95% CI, $(-0.17, -0.06)$), which curtailed the inflation. However, this analysis also yielded no evidence that the psychological inoculations protected pro-environmental behaviour from the significant decrease due to the climate disinformation statements (main effect of condition: $\chi(6) = 11.7805$, $P = 0.07$; $H_3$ not supported).

**Inoculation effects on truth discrimination—preregistered analyses**

We summed all responses correctly identifying true and false statements[163] (Table 4) as the dependent variable for the multilevel model.

The analysis suggested a significant main effect of condition ($F(6, 5,936.4) = 2.4338$, $P = 0.024$; Fig. 2d). Contrasts revealed that the effect was driven by the accuracy inoculation ($t(5,936.2) = 3.360$; $P < 0.001$; $\beta = 0.36$; 95% CI, $(0.15, 0.57)$), as participants inoculated to reflect on the accuracy of climate-change-related information were significantly better at discriminating true and false statements about the topic. Equivalence tests showed that, if present at all, the protective effects of the other inoculations were significantly smaller than $\delta = 0.20$ (Supplementary Fig. 8; $H_4$ partial support) (scientific consensus inoculation: $t(1,670.06) = -3.782$; $P < 0.001$; 90% CI, $(-0.15, 0.22)$; trust inoculation: $t(1,705.24) = -2.195$; $P = 0.014$; 90% CI, $(-0.03, -0.39)$; transparent communication inoculation: $t(1,697.78) = -2.387$; $P = 0.009$; 90% CI, $(-0.01, -0.37)$; moralization inoculation: $t(1,726.29) = -3.151$; $P = 0.001$; 90% CI, $(-0.07, 0.29)$; positive emotions inoculation: $t(1,701.69) = -2.759$; $P = 0.003$; 90% CI, $(-0.03, 0.37)$).

**Table 4 | Generated climate headlines for the truth discrimination task**

| Coding | Climate-relevant news headline |
|---|---|
| True_Supporting_1 | Earth's average temperature continues to rise, setting new record highs each decade. |
| True_Supporting_2 | Human activities, such as burning fossil fuels, are the main cause of climate change. |
| True_Supporting_3 | Climate change is leading to more intense and frequent natural disasters. |
| True_Supporting_4 | The transportation sector is a significant contributor to greenhouse gas emissions. |
| True_Supporting_5 | Rising seas could displace hundreds of millions of people by the end of the century. |
| True_Delaying_1 | Projections of Regional Impacts of Climate Change are Subject to Uncertainty. |
| True_Delaying_2 | Transportation Sector Transition to Electric Vehicles Can Cost Billions in Infrastructure Upgrades. |
| True_Delaying_3 | Brazil Missing Paris Agreement Targets with Deforestation and Agricultural Expansion Driving Up Emissions. |
| True_Delaying_4 | China's continued construction of coal-fired power plants threatens progress on climate goals. |
| True_Delaying_5 | Developing Countries Require $40 Billion Annually to Mitigate Climate Change. |
| False_Supporting_1 | Climate change will cause the extinction of up to 75% of all species on Earth. |
| False_Supporting_2 | Global temperatures may rise by up to 20°C by the end of the century, potentially resulting in widespread drought and famine due to climate change. |
| False_Supporting_3 | The Earth may enter a period of 'runaway warming' that cannot be stopped, which could lead to the collapse of civilization due to climate change. |
| False_Supporting_4 | Germany Leads the Way in Renewable Energy, with Nearly 65% of Electricity Generated from Renewables. |
| False_Supporting_5 | Climate Catastrophe: Entire Cities to be Submerged by Rising Seas Within Decades. |
| False_Delaying_1 | Extreme Weather: Natural variability, not human activity, is the main driver of extreme weather events. |
| False_Delaying_2 | The Climate Challenge Can Be Addressed Through Innovation and Technology Advancements in Fossil Fuels. |
| False_Delaying_3 | Carbon Dioxide is Not a Pollutant, but a Benefit to the Environment. |
| False_Delaying_4 | Catastrophic Consequences of Global Warming are Inevitable and Unavoidable. |
| False_Delaying_5 | Renewable Energy is Costly and Inefficient, and Should Not be Subsidized. |

'True/False' refers to true or false statements; 'Supporting/Delaying' refers to statements supporting or opposing climate science and action.

## Inoculation effects on truth discrimination—exploratory analyses

We conducted additional exploratory analyses because recent studies found that psychological inoculations may increase participants' bias towards judging any statement as false[67,171]. First, decomposing the effect of the accuracy inoculation across four types of climate statements (climate support versus delay, true or false) showed that accuracy-inoculated participants were better at discerning false statements delaying climate action ($t(1,743.74) = 2.204$; $P = 0.024$; $\delta = 0.11$; 95% CI, (0.02, 0.27)), but not the other types of statements—that is,

true statements delaying climate action ($t(1,745) = 3.557$; $P < 0.001$; 90% CI, (−0.16, 0.07)), true statements supporting climate action ($t(1,744.07) = 2.614$; $P = 0.005$; 90% CI, (−0.19, 0.01)) and false statements supporting climate action ($t(1,744.69) = 4.042$; $P < 0.001$; 90% CI, (−0.10, 0.09)).

Second, we applied signal detection theory[172] to further scrutinize performance in the truth discrimination task. In brief, signal detection theory posits that stimulus detection is contingent on people's discriminant ability and their overall response bias towards reporting all (dis)information as true or as false. We conducted these analyses because recent studies found that psychological inoculations may increase participants' bias towards judging any statement as false[67,171]. We therefore conducted additional $t$-tests and equivalence tests to predict participants' discriminatory ability and response bias. In line with the preregistered results, we found that only the accuracy inoculation increased participants' discriminatory ability ($t(1,687.71) = 3.386$; $P < 0.001$; $\delta = 0.17$; 95% CI, (0.02, 0.06)). In contrast to findings from previously published studies, equivalence tests suggested that the inoculations did not make participants more biased towards considering all task statements as false (scientific consensus inoculation: $t(1,675) = 3.502$; $P < 0.001$; 90% CI, (−0.02, 0.01); trust inoculation: $t(1,702.07) = 2.534$; $P = 0.006$; 90% CI, (−0.03, 0); transparent communication inoculation: $t(1,696.08) = 3.491$; $P < 0.001$; 90% CI, (−0.02, 0.01); moralization inoculation: $t(1,716.27) = 2.499$; $P = 0.006$; 90% CI, (−0.03, 0); accuracy inoculation: $t(1,681.97) = 3.459$; $P < 0.001$; 90% CI, (−0.02, 0.10); positive emotions inoculation: $t(1,703) = 3.978$; $P < 0.001$; 90% CI, (−0.01, 0.02)).

## Moderation by CRT-2 score—preregistered analysis

As preregistered, we aggregated the cognitive (scientific consensus, transparent communication and accuracy) and the socio-affective (trust, moralization and positive emotions) inoculations into two factors and tested whether the participants' tendency for intuitive/deliberative thinking (that is, their scores on the Cognitive Reflection Task, version 2 (CRT-2)[173]) moderated the inoculations' protective effect on affect towards climate mitigation action. We did not find evidence of a significant moderation of CRT-2 scores on the overall effectiveness of the inoculation groups ($t(5,963) = 0.515$; $P = 0.61$; $\beta = 0.15$; 95% CI, (−0.41, 0.70); Supplementary Table 7), but we found suggestive evidence that the preregistered three-way interaction between CRT-2 scores, the psychological drivers and trial was significant ($t(96,787) = −2.101$; $P = 0.036$; $\beta = −0.02$; 95% CI, (−0.03, −0.001); Supplementary Table 8). Visual inspection of this relationship suggested that participants who rely more on deliberative thinking and are inoculated with a cognitive inoculation are less affected by each of the climate disinformation statements, but participants who rely more on deliberative thinking and are inoculated with a socio-affective inoculation are more affected by each of the climate disinformation statements (Fig. 3).

A preregistered supplementary multilevel model analysing the moderation of CRT-2 scores on each condition suggested that the moderation was primarily driven by the scientific consensus ($F$-ratio = 4.087, $P = 0.043$), transparent communication ($F$-ratio = 6.265, $P = 0.012$) and accuracy inoculations ($F$-ratio = 5.316, $P = 0.021$), and that the more participants relied on intuitive thinking, the more the trust inoculation protected them ($F$-ratio = 4.102, $P = 0.043$; Supplementary Table 9 and Supplementary Figs. 9–15; H₅ moderate support).

## Moderation by political ideology—exploratory analyses

As political conservativism is a main predictor of climate skepticism[13] and resistance to misinformation interventions[173] but its effect on psychological inoculations is unclear[38], we explored whether political ideology moderated the protective effects of any psychological inoculation against the 20 climate disinformation statements using a multilevel model (Supplementary Table 10). We did not find evidence that political ideology influenced overall affect ($t(6,977) = 0.452$;

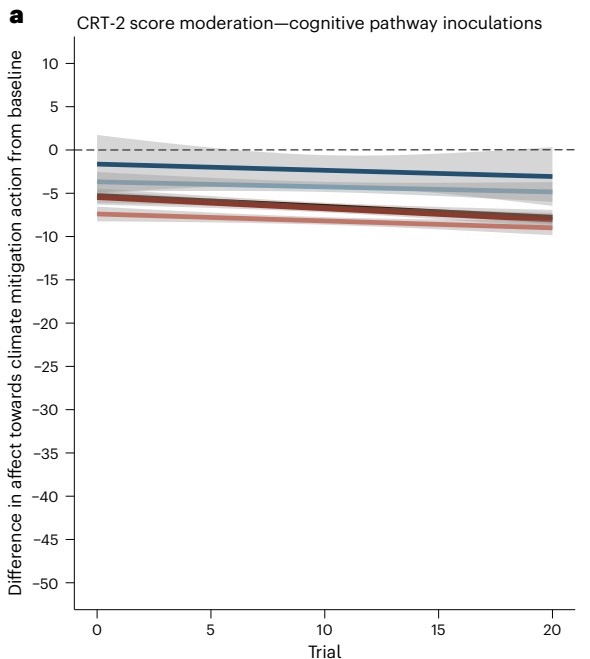

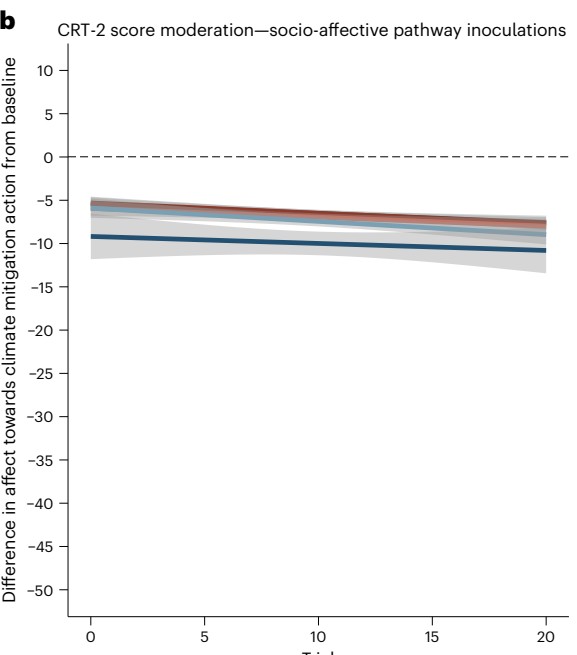

**Fig. 3 | How participants' (*N* = 6,816) tendency for deliberative versus intuitive thinking affected the effects of the inoculations relying on the cognitive or socio-affective pathway in protecting affect towards climate mitigation action against 20 climate disinformation statements. a,b,** Each panel represents one pathway and each level of tendency for deliberative thinking (CRT-2 score of 3 represented light blue, CRT-2 score of 4 represented in dark blue) and intuitive thinking (CRT-2 score of 0 represented in dark red, CRT-2 score of 1 represented in red). The *y* axis represents the mean difference in affect towards climate mitigation action from baseline (pre-inoculation and pre-disinformation provision) in the interval between −50 and +10, to better visualize the three-way interaction between pathway, trial and CRT-2 scores (multilevel model three-way interaction contrast: $t_{\text{two-sided}}(96,787) = -2.101$;

$P = 0.036$; $\beta = -0.02$; 95% CI, (−0.03, −0.001)). Values increasing from 0 are related to feeling overall more positively towards climate mitigation action, and values decreasing below 0 related to feeling overall more negatively towards climate mitigation action. The dashed line represents no mean difference from baseline. The *x* axis represents the trial number, with trial = 0 representing affect pre-intervention, and the numbers 1 to 20 representing each climate disinformation statement received. The light grey bands represent the mean-centred standard errors produced by fitting a linear model. Panel **a** shows the cognitive pathway (scientific consensus inoculation, transparent communication inoculation and accuracy inoculation; *N* = 2,508); panel **b** shows the socio-affective pathway (trust inoculation, moralization inoculation and positive emotions inoculation; *N* = 2,587). Colour palette by the MetBrewer package[169].

$P = 0.65$; $\beta = 0.13$; 95% CI, (−0.43, 0.69)) or the trial-by-trial decrease in affect towards climate mitigation action after each climate disinformation statement ($t(11,300) = -0.871$; $P = 0.38$; $\beta = -0.007$; 95% CI, (−0.02, 0.01)). However, we found suggestive evidence that political ideology moderated the trial-by-trial effectiveness of the psychological inoculations ($F(112,994) = 2.1846$, $P = 0.041$). In particular, the most liberal participants inoculated via the scientific consensus inoculation (*F*-ratio = 12.807, $P < 0.001$) and the positive emotions inoculation (*F*-ratio = 5.623, $P = 0.02$) were the most protected from the climate disinformation statements (Supplementary Figs. 15–21). The moderation was not significantly different between the two inoculations and the passive control condition (scientific consensus: $t(11,300) = -1.948$; $P = 0.051$; $\beta = -0.21$; 95% CI, (−0.04, 0.0001); positive emotions: $t(11,300) = -1.122$; $P = 0.26$; $\beta = -0.12$; 95% CI, (−0.03, 0.01)) and was limited to affect towards climate mitigation action (Supplementary Tables 10 and 11).

## Discussion

In our study, conducted in 12 diverse countries (Fig. 4), we found strong evidence that climate disinformation powerfully influences how people feel about, think of and evaluate climate change and climate actions. Presentation of climate disinformation not only influenced participants' beliefs in anthropogenic climate change and evaluations of climate action but also reduced their ability to discriminate between true and false climate statements. Moreover, processing climate disinformation directly affected pro-environmental behaviour, lowering participants' performance in a validated behavioural task with positive

real-world climate consequences[164]. Although exploratory analyses suggested that the psychological inoculations may have been able to counteract the negative impact of the first climate disinformation statement, we found no confirmatory evidence that the six psychological inoculations had a protective effect against the 20 climate disinformation statements, except that the accuracy inoculation significantly protected truth discernment ability. One potential explanation for the lack of significant effects in most preregistered analyses is that the true inoculation effects against the 20 climate disinformation statements may have been lower than our a priori effect size of interest. This effect size was based on the meta-analytic effect of fact-checking one disinformation statement in general[165]; it should be noted, however, that a more recent meta-analysis found no significant effect and high heterogeneity of fact-checking misinformation about scientific evidence, especially for polarized scientific topics such as climate change[174].

Even though our results do not yield evidence for promising main effects of the inoculations, they explained significant (albeit suggestive) trial-by-trial treatment heterogeneity by a preregistered moderator—that is, the tendency for intuitive or deliberative thinking. At the theoretical level, these moderation effects confirm the utility of using a comprehensive model of (anti)science belief formation and updating. A tendency for intuitive thinking increased the protection granted by the trust inoculation—that is, an inoculation acting on the socio-affective pathway—while a tendency for deliberative thinking increased the protection given by all three psychological inoculation strategies acting on the cognitive pathway (scientific consensus, transparent communication and accuracy inoculations).

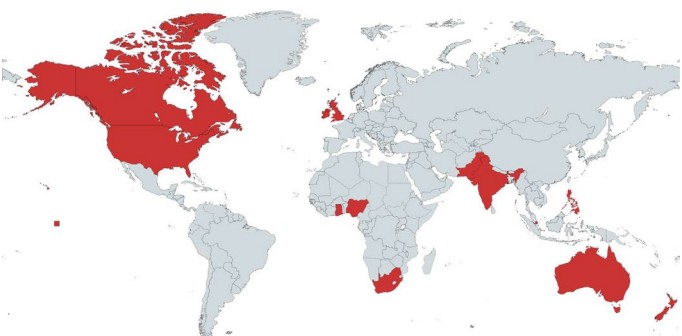

**Fig. 4 | The country distribution of the data collection.** Red represents the countries where we collected samples of *N* = 568 participants each. Figure created with mapchart.net.

These results echo calls to systematically investigate the treatment heterogeneity of behavioural interventions across multiple moderators of interest[168,174,175]. They furthermore suggest that the theoretical framework presented here can be used not only to identify coherent sets of psychological inoculation strategies but also to identify moderators explaining treatment heterogeneity due to a tendency for specific thinking styles. Taken together, our results suggest that the protective effects of psychological inoculations may be conditional to tailoring them to the pathways of (anti)scientific belief formation and updating that are prevalent in the target population.

At the applied level, our findings indicate that psychological inoculations have no unintended consequences—for example, backfire effects[67,171]—but may have no or only very limited capacity to protect against multiple disinformation statements related to climate change[67] (cf. ref. 176). Even the strategy of reinforcing the inoculated protection with follow-up 'booster shots'[68,103,177] may be untenable, as our evidence suggests that they would already be needed after encountering a single climate disinformation statement. Weighting the influence and discursive presence[30,178,179] of climate disinformation with the partial effectiveness of psychological inoculations, our findings join the recent discussion suggesting that behavioural science interventions do not seem to be efficacious enough to tackle systemic problems such as climate disinformation by themselves[59,60,180–182]. Systemic interventions, such as content moderation, virality circuit breakers[180], deplatforming[178,183] or changing online engagement metrics towards the accuracy of information[184], may be better at curbing climate disinformation. However, systemic actions are enforceable only by the same platforms that might be incentivized to let climate disinformation spread[184–190]. For this reason, both behavioural and systemic intervention approaches need to be further developed and applied, especially given the direct evidence provided here that climate disinformation drastically decreases climate-relevant judgements and behaviours.

These conclusions are to be weighed against the study limitations. First, our sample was smaller than the one required to detect an effect closer to a more conservative estimate of the effect of psychological inoculations[67]. Second, while we aimed for more generalizability across populations than previous studies[166], we pragmatically recruited participants from English-speaking countries and presented them one-size-fits-all psychological inoculations. Fully generalizable interventions would also require non-English-speaking participants, with inoculations adapted to the context of delivery. Third, all psychological inoculations were delivered as texts to be passively read, instead of engaging, multimodal videos or activities that could have motivated more processing[67,191].

While the psychological inoculations evaluated in this Registered Report were of limited efficacy, the framework proposed here can be helpful to systematize future psychological inoculation research to fight climate disinformation. It may generate new psychological inoculations, which can result in more systematic research[192] on psychological inoculations, their moderators and climate-relevant outcomes. For example, one of the reviewers suggested developing a 'pluralistic ignorance' inoculation, whereby participants are made aware that most of their peers support climate mitigation action, and that awareness should increase support of climate action[193,194]. Many-labs and mega-studies approaches, promising recent frameworks for systematically creating and testing multiple interventions[195–197] with large sample sizes that could help the detection of small intervention effects, can be applied to test sets of theory-guided psychological inoculation strategies against a validated set of climate disinformation statements, measuring climate-relevant outcomes and mapping the heterogeneity generated by model-identified individual differences. Such a combination could produce the next generation of psychological inoculations, which may yield better protection from climate disinformation.

## Methods
### Ethics information
The study was approved by the University Commission for Ethical Research in Geneva (CUREG2.0) of the University of Geneva, Switzerland (ID: CUREG-2022-05-43-Corr-Brosch). The participants explicitly consented to the study at the beginning of the survey; they were compensated for their time.

### Design
The study followed a mixed design. The participants were randomly assigned to one of eight different between-participants conditions: pure control (no inoculation and no disinformation), passive disinformation control (disinformation without inoculation), scientific consensus inoculation, trust inoculation, transparent communication inoculation, moralization inoculation, accuracy inoculation and positive emotions inoculation. We chose a passive disinformation control condition over an active or positive control to better mimic real-life information environments, where climate disinformation is most frequently encountered passively and in multiple occurrences. The participants and experimenters were blind to the name and aim of the condition that the participants were randomized into (double blind). The experiment contained 20 within-participant repeated measures of affect towards climate mitigation actions, assessed after each of the 20 climate disinformation statements.

**Procedure.** The participants accessed the survey through an anonymous link made available by the panel provider and provided their explicit consent to the study. After consenting, the participants reported their demographics (gender, age, education and political orientation (single-item, ten-point scale: 1 = 'extreme liberalism/left' to 10 = 'extreme conservativism/right')), completed a baseline measure of affect towards climate action and completed the CRT-2 (ref. 198), in random order. A two-strikes-out attention check ('Please select '3' to make sure you are paying attention') was presented; failing it triggered a warning with a ten-second time penalty. Inattentive participants received the attention check for a second time, and participants found inattentive again afterwards were screened out of the survey (*N* = 16). Attentive participants were then randomly allocated to one of the eight conditions and received the assigned intervention or, for participants in the passive disinformation control condition, were directly forwarded to the following section of the experiment. Participants in the pure control condition received neither the interventions nor the climate disinformation statements. All interventions were presented sequentially in four screens, with a 5–20 s time lock (depending on the content length of each screen) that did not allow the participants to manually proceed to the next screen until the time had elapsed. A manipulation check measuring the participants' motivation to resist persuasion[170] followed. Afterwards, the participants received 20 real climate disinformation statements in the form of anonymous tweets,

in randomized order with a 2 s time lock, and reported their affect towards actions to mitigate climate change after each disinformation statement. Following the disinformation provision, the participants completed the climate change perceptions scale[162], a modified version of the WEPT[164,199] and the truth discernment task, all described in full detail below. Finally, we probed the participants' understanding of the experimental aims with an open-ended question to account for potential demand effects. The survey ended with an extended debriefing that contained a reminder of the scientific consensus behind climate change with a link to the latest IPCC report. The survey duration was about 21 minutes.

**The six inoculations.** The inoculations were adapted to the same presentation format, as textual stimuli divided into two paragraphs. All inoculations contained an opening paragraph referring to the IPCC assessment of anthropogenic climate change, 'In their latest assessment, the Intergovernmental Panel on Climate Change (IPCC) has declared that anthropogenic climate change is happening, and urgent action is needed to prevent irreversible negative effects on the planet and society', followed by a pre-emptive warning of incoming threatening information[38]: 'However, some politically-motivated groups use misleading tactics to try to convince the public that there is a lot of disagreement among scientists and that climate action is useless or harmful to society'. The second paragraph contained the inoculation itself (Table 2), introduced by the sentence: 'When confronted with such misleading information about the science of climate change and the actions to mitigate it,...'. To minimize any differences between experimental conditions other than the theory-derived experimental variations, we created a reference text and maintained the thematic information to be as similar as possible across inoculations operating at the same communication basis, but varying the salience of aspects that make the different psychological drivers salient. Both the scientific consensus inoculation and the trust inoculation operated by changing the perception of the source of scientific messages about climate change, either by stressing the consensus about anthropogenic climate change within the scientific community[38] or by making the trustworthiness of IPCC scientists salient[96]. Both the transparent information inoculation and the moralization inoculation operated by emphasizing specific aspects of the presented climate mitigation actions. For the transparent communication inoculation, a transparent discussion of the societal costs of climate action including pros and cons of climate action preceded the disinformation; for the moralization inoculation, the importance of climate action was framed using moralizing words taken from the Moral Foundation Dictionary[200] to increase its link with people's moral convictions[201,202]. Finally, both the accuracy inoculation and the positive emotions inoculation operated by changing the internal state of the recipient. In the accuracy inoculation, we reframed the original accuracy prompt[135] into a passive psychological inoculation[191] where the participants were prompted to consider accuracy when evaluating the information, whereas in the positive emotions inoculation, the participants were prompted to consider positive emotions related to acting in a climate-friendly way. The complete text for all interventions is provided in Table 2, with cross-condition differences underlined.

**Disinformation provision.** The disinformation provision section of the experiment consists of 20 different actual disinformation statements collected from Twitter and pre-validated in a pilot study (Table 3). We followed a two-step procedure for the selection of the climate disinformation statements. First, we created a list of the available Twitter handles of members of the climate change countermovement by reviewing academic and journalistic resources that identified actors that have been spreading disinformation about climate change science and policies[11,13–16] (https://www.desmog.com/climate-disinformation-database/). We augmented this list with all the Twitter-active members of

the 'World Climate Declaration' (https://clintel.org/world-climate-declaration/), a document with 500 signatories—at the time of data collection—that misinforms the public about anthropogenic climate change. Through academic access to the Twitter API, we collected all the tweets by these users, first from account creation until April 2022, and a second time from 31 October to 20 November 2022, the week leading to and the two weeks of the UNFCCC Conference of the Parties 27. Second, we randomly selected and manually coded 20,000 of their tweets according to climate relatedness (1 = 'not at all related to climate change' to 4 = 'absolutely related to climate change'), disinformation (1 = 'not at all disinformation about climate change' to 4 = 'absolutely disinformation about climate change') and delay (1 = 'not at all a delay argument about climate change policies' to 4 = 'absolutely a delay argument about climate change policies') status, following the coding schema and instructions by Coan et al.[33] and Lamb et al.[14]. From a final pool of $N = 1,033$ tweets identified as climate related and disinformation/delay, we identified $N = 79$ tweets that were understandable without requiring background information and not including country-specific aspects. These 79 tweets were pretested with a representative sample of $N = 504$ British participants on the data collection platform Prolific (https://www.prolific.co/), in terms of their impact on affect towards climate action and 12 further variables—for example, perceived political slant—that may affect processing of disinformation about political topics[203] (the full list of disinformation, with a description of the pretesting design and all validation materials and data, can be found at https://osf.io/m58zx). Among these 79 statements, we selected $N = 20$ disinformation statements (Table 3) that deviated the least from the mean ratings across all 15 validation measures, evenly divided between 10 disinformation statements about climate science and 10 disinformation statements to delay climate action (according to coding criteria developed in previous research[14,33]). During the experiment, the participants were presented with all 20 selected climate disinformation statements in randomized order. Each statement was presented as an anonymous tweet, with the default user image, no identifying information and no engagement metrics. After each disinformation statement, the participants rated their affect towards climate actions on a visual analogue scale.

**Affect towards climate mitigation actions.** We measured the participants' affect towards actions to mitigate climate change with a visual analogue scale adapted from previous research[96,204] ('In general, what kind of feelings do you have when you think about actions to mitigate climate change?'; 0 = 'very negative', 50 = 'neutral', 100 = 'very positive'; scale anchored at 50).

**Climate change beliefs.** We assessed the participants' beliefs about climate change with the climate change perception scale[162], a validated scale that encompasses different dimensions of the appraisal of climate science and the consequences of climate change. While the published scale is composed of five different subscales and related factors, the authors note that the climate change perception scale allows for the selection of subscales of interest[162]. We therefore focused on the three subscales measuring participants' belief in the reality of climate change, the causes of climate change and the consequences of climate change. Climate change beliefs were collected with nine items (reality subscale: 'I believe that climate change is real'; 'Climate change is NOT occurring' (reverse scored); 'I do NOT believe that climate change is real' (reverse scored); Cronbach's $\alpha = 0.81$; causes subscale: 'Human activities are a major cause of climate change'; 'Climate change is mostly caused by human activity'; 'The main causes of climate change are human activities'; Cronbach's $\alpha = 0.91$; consequences subscale: 'Overall, climate change will bring more negative than positive consequences to the world'; 'Climate change will bring about serious negative consequences'; 'The consequences of climate change will be very serious'; 1 = 'strongly disagree',

7 = 'strongly agree'; Cronbach's $\alpha$ = 0.82; items for each subscale were mean-scored).

**The pro-environmental behaviour task.** We used a shortened version[199] of the WEPT[164,199], a validated, multi-trial web-based procedure to measure actual pro-environmental behaviour. In this task, participants can voluntarily choose to exert effort by screening numerical stimuli for the occurrence of target numbers beginning with an even digit and ending with an odd digit (for example, '23'). In this version of the WEPT, the participants were able to complete up to eight different numerical screenings of 60 numbers per page. The participants' willingness to engage in the screening task was prompted before each new page with a yes/no question: participants who answered positively were directed to screening the numbers; participants who answered negatively were directed to the following section of the study. In the instructions, we explicitly explained to the participants that each screening page they accurately completed would result in an actual tree being planted by an environmental organization, the Eden Reforestation Project (https://www.edenprojects.org/), with whom we partnered to plant trees. In other words, the participants were able to create actual environmental benefits (measured in terms of trees planted by the environmental organization) at an actual behavioural cost (personal time)[164,199]. They were able to track their tree-planting progress, from zero to up to eight trees, with an image presented between the pages of the numerical screenings. We measured their pro-environmental behaviour in terms of the number of pages that each participant completed while correctly screening more than 90% of the target numbers[164,199].

**Truth discernment task.** Inspired by recent work presenting a measure of domain-general news veracity discernment[163], we developed a climate-specific truth discernment task in which the participants had to categorize 20 statements mentioning climate-related topics as either false or real statements ('Please categorize the following statements as either "False Statement" or "Real Statement"'; binary choice: 'Real' or 'False'; item and response order randomized). These 20 statements were equally divided between true and false headlines and between supporting and opposing climate science and action. All statements were generated by interacting with an AI tool (ChatGPT version 4, by OpenAI). Over 300 true and false statements mentioning climate change or climate mitigation actions were initially created. The statements were then fact-checked and condensed into a longlist out of which 10 true and 10 false statements were selected to be included in the truth discernment task. The final statements are presented in Table 4 (the full list of generated statements is available in the Open Science Framework repository).

**Tendency for intuitive versus deliberative thinking (CRT-2).** We assessed the participants' tendency for intuitive versus deliberative/reflective thinking using version 2 (ref. 198) of the CRT[205]. This task comprises four open-ended, verbal problems that have an intuitive but incorrect answer and require reflection to correctly answer: 'If you're running a race and you pass the person in second place, what place are you in?' (intuitive answer: first; correct answer: second); 'A farmer had 15 sheep and all but 8 died. How many are left?' (intuitive answer: seven; correct answer: eight); 'Emily's father has three daughters. The first two are named April and May. What is the third daughter's name?' (intuitive answer: June; correct answer: Emily); and 'How many cubic feet of dirt are there in a hole that is 3' deep x 3' wide x 3' long?' (intuitive answer: 27; correct answer: none). We used the CRT-2 instead of the traditional version because it shares less variance with numerical skills[198]. Numeracy skills vary across countries[206] and could therefore confound the original measure of tendency for intuitive thinking. We computed the CRT-2 score as the number of correct answers given, ranging from 0 to 4, where lower scores represent an increasing tendency for intuitive

thinking, and higher scores represent an increasing tendency for deliberative thinking.

**Manipulation check.** Following the psychological inoculations literature[207], we measured motivation to resist persuasion as a theoretically and experimentally validated manipulation check[208] with the four-item motivational threat measure proposed by Banas and Richards[209] ('Indicate your level of agreement with the following statements': 1 = 'strongly disagree', 7 = 'strongly agree'; 'I want to defend my current attitudes from attack'; 'I feel motivated to think about why I hold the beliefs I do about climate change'; 'I feel motivated to resist persuasive messages about climate change'; 'I want to counterargue conspiracy theories about climate change'; items were mean scored as Cronbach's $\alpha$ = 0.76). We expected motivation to resist persuasion to increase for participants receiving the different inoculations, compared with participants in the passive disinformation control condition.

**Demand effects check.** We probed the participants' understanding of the aim of the experiment by asking them, 'Could you please describe what you think the aim of the experiment was?' Two coders then rated the participants' belief in the experimental objective with a multiple-choice question ('To what degree do you think the participant believed we were testing interventions to fight climate disinformation?'; 0 = 'They seemed very convinced we were not testing interventions to fight climate disinformation'; 1 = 'They seemed somewhat convinced we were not testing interventions to fight climate disinformation'; 2 = 'They seemed unsure if we were testing interventions to fight climate disinformation'; 3 = 'They seemed somewhat convinced we were testing interventions to fight climate disinformation'; 4 = 'They seemed very convinced we were testing interventions to fight climate disinformation'; the ratings were averaged, and differences in scoring were discussed and resolved).

## Sampling plan

We collected the sample with quotas for gender and age from the panel provider Market Science Institute. The sample comprised participants from 12 countries, $N$ = 568 participants per country, for a total of $N$ = 6,816 (mean age, 39.15 ± 14.17; $N$ = 3,555 female) participants. We identified the required sample size a priori, with G*Power (version 3.0; ref. 210), to have 95% power to detect a difference between any intervention condition and the passive disinformation control condition of $\delta$ = 0.20 in a one-tailed $t$-test with $\alpha$ = 0.005, for all main hypotheses separately. We selected the smallest effect size of interest (SESOI) from the lower bound of the confidence interval of the meta-analytically identified effect size[211] of fact-checking interventions on political topics[165], as we reasoned that a new disinformation intervention would be of interest if and only if it has an effect that is larger than already available interventions such as fact-checks. Incidentally, a recent paper showed that the effects of more established psychological inoculations on sharing intentions of manipulative content is $\delta$ = 0.20 (ref. 191), increasing our confidence in the practical interest of this SESOI.

**Countries.** We recruited participants based in the United States, Canada, the United Kingdom, Ireland, Australia, New Zealand, Singapore, the Philippines, India, Pakistan, Nigeria and South Africa (Fig. 4) to generalize our findings on the effectiveness of the six psychological inoculations across the globe and in non-WEIRD contexts. We settled on 12 countries as the minimum number of countries to provide a reasonably accurate statistical estimation for country-level variation in our dependent variables as a random effect in multilevel models[212] rather than conducting cross-country comparisons. The 12 countries were furthermore chosen pragmatically for English being the main or one of the official languages, to maintain the climate disinformation

statements in their original language and therefore maintain the highest ecological validity.

**Data inclusion.** Participants were removed from the survey and replaced with new respondents when they (1) did not consent to the study ($N = 118$), (2) did not finish the study ($N = 1,037$) or (3) failed the two-strikes-out attention check ($N = 16$). All incomplete responses and complete responses where participants no longer consented to the study at the end of the survey were removed; all other responses were included in the data analyses.

## Analysis plan

The data were analysed with the most recent version of R available at the time of data collection completion (version 4.1.3), with packages lme4 (ref. [213]), lmerTest[214], TOSTER[215], DHARMa[170] and emmeans[216]. Unless specified, we tested the hypotheses with multilevel models.

**Manipulation check.** We analysed the differences between the passive disinformation control condition and the six inoculation conditions in terms of the motivation to resist persuasion[177] with a set of six independent-sample, one-tailed $t$-tests. We expected all inoculated participants to report significantly more motivation to resist messages countering climate science and climate mitigation action than participants in the passive disinformation control condition.

**Primary hypotheses.** Gender, age and political ideology were added as covariates in all models. All random effects of multilevel models were weighed separately with Akaike information criterion (AIC) model comparison, and the random effect structures with an AIC value within 2 of the best model's AIC were used in each analysis.

We analysed changes in affect towards climate mitigation actions during the disinformation provision as the dependent variable with a multilevel model. We specified three random effects: an intercept for participant; an intercept for country, to account for the variance associated with each country[212,190]; and a random intercept for the internal numbering of the climate disinformation statements, to proactively account for any variance associated with each particular climate disinformation statement[190], as some differences across validation measures remained after the selection of the set of disinformation stimuli. Unless otherwise specified, we specified the following as fixed effects: condition (factor, seven levels, dummy coded with 0 = passive disinformation control condition as the reference contrast), trial (continuous variable, from 1 to 20) and the two-way interactions of trial with condition.

We analysed performance in the modified version of the pro-environmental behaviour task as the dependent variable with a multilevel model. Unless otherwise specified, we specified condition (factor, seven levels, dummy coded with 0 = passive disinformation control condition as the reference contrast) as the fixed effect.

To test whether inoculated participants had more positive affect towards climate mitigation action than participants in the control condition after receiving 20 climate disinformation statements, we first compared affect toward climate action at the end of the intervention (that is, after the 20th disinformation statement) of the participants in the passive disinformation control condition with that of the participants in each inoculation condition separately, with a one-tailed independent-sample $t$-test with $\alpha$ corrected to 0.005. As none of the contrasts were significant, we first visually inspected the affect curve of the 20 measurements of affect across the processing of the 20 climate disinformation statements and did not visually identify ceiling or flooring effects for the intervention conditions. We then tested whether the difference between the passive disinformation control condition and the inoculation of interest is smaller than our SESOI ($\delta = 0.20$) with equivalence testing (RQ1; see the Design Table of the Stage 1 Protocol).

To test whether inoculated participants had more positive affect towards climate mitigation action than participants in the passive

disinformation control condition after each one of the 20 climate disinformation statements, we analysed changes in affect towards climate mitigation actions during the disinformation provision with a multilevel model (RQ1; see the Design Table of the Stage 1 Protocol). We specified three random effects: an intercept for participant; an intercept for the internal numbering of the climate disinformation statements, to account for the variance associated with each disinformation statement[190]; and an intercept for country (factor, alphabetically coded), to account for the variance associated with each country[212]. We specified the following as fixed effects: condition (factor, seven levels, dummy coded with 0 = passive disinformation control condition as the reference contrast), trial (continuous variable, from 1 to 20) and the two-way interactions of trial with condition to test whether inoculated participants had more positive affect towards climate mitigation action than participants in the passive disinformation control condition after each climate disinformation statement.

To test whether inoculated participants reported believing more in the reality, causes and consequences of climate change than participants in the passive disinformation control condition after receiving 20 climate disinformation statements, we analysed the climate change perception subscales with three multilevel models (RQ3; see the Design Table of the Stage 1 Protocol). We specified one random effect: an intercept for country (factor, alphabetically coded), to account for the variance associated with each country[190]. We specified condition (factor, seven levels, dummy coded with 0 = passive disinformation control condition as the reference contrast) as the only fixed effect besides the covariates. We first visually inspected the raincloud distribution of the responses of each climate change perception subscale[162], to visually identify ceiling or flooring effects. Upon visual confirmation of a normal distribution, we tested whether the difference between the passive disinformation control condition and the inoculation of interest was smaller than our SESOI ($\delta = 0.20$) with equivalence testing.

To test whether inoculated participants completed more pages in the WEPT than participants in the passive disinformation control condition after receiving 20 climate disinformation statements, we analysed the performance in the modified version of the WEPT with a multilevel model, with the number of completed pages as the dependent variable (RQ2; see the Design Table). We did not specify the expected distribution of the WEPT responses in the Stage 1 report. For transparency, we assumed the data would be Poisson distributed, as the WEPT dependent variable was a count; for completeness, we present the more common linear multilevel model in Supplementary Table 6 (the results do not differ). We specified one random effect: an intercept for country (factor, alphabetically coded), to account for the variance associated with each country[212]. We specified condition (factor, seven levels, dummy coded with 0 = passive disinformation control condition as the reference contrast) as the only fixed effect besides the covariates. We plotted the WEPT performance data and identified a flooring effect.

To test whether inoculated participants have higher news veracity discernment[163] than participants in the passive disinformation control condition after receiving 20 climate disinformation statements, we analysed the performance in the truth discernment task with a multilevel model (RQ4; see the Design Table). We calculated news veracity discernment as the sum of correct identifications of true and false climate-related statements[163]. We specified one random effect: an intercept for country (factor, alphabetically coded), to account for the variance associated with each country[212]. We specified condition (factor, seven levels, dummy coded with 0 = passive disinformation control condition as the reference contrast) as the only fixed effect besides the covariates. We tested whether the difference between the passive disinformation control condition and the inoculation of interest was smaller than our SESOI ($\delta = 0.20$) with equivalence testing. We furthermore calculated the real news detection and the false news detection scores[163], to investigate whether the inoculations influenced

only one of the two underlying factors of the general news veracity discernment score.

**Secondary hypothesis.** For the secondary hypothesis analysis, we limited our sample to those participants who received one of the six inoculations ($N = 5,112$). We analysed affect towards climate action mitigation during the disinformation provision with the multilevel model used for Hypothesis $H_{IA}$. We added the CRT-2 score (continuous, range from 0 to 4) as a fixed predictor; we substituted the 'condition' variable with a 'drivers' factor (two levels: socio-affective and cognitive), each containing the corresponding psychological inoculations (socio-affective: trust inoculation, moralization inoculation and positive emotions inoculation; cognitive: scientific consensus inoculation, transparent communication inoculation and accuracy inoculation); and we added the two-way interactions of driver with CRT-2 score, the two-way interactions of trial with CRT-2 score and the three-way interaction of driver, trial and CRT-2 score.

**Control analyses.** For the $H_{\text{control A–D}}$ analyses, we limited our sample to those participants who participated in the pure control and passive disinformation control conditions ($N = 1,704$). We tested whether consecutively presenting the 20 real climate disinformation statements decreased participants' (A) affect towards climate mitigation action, (B) beliefs in climate change, (C) performance in the modified version of the WEPT[164] and (D) truth discernment. We compared affect toward climate action, belief in climate change, WEPT performance and truth discernment at the end of the intervention (that is, after the 20th disinformation statement) of the participants in the passive disinformation control condition with those of the participants in the pure control condition, separately, with a one-tailed independent-sample $t$-test with $\alpha$ corrected to 0.005. For hypothesis $H_{\text{control IA-bis}}$, we conducted an additional one-tailed, paired-sample $t$-test within the passive disinformation control condition, with affect towards climate mitigation action as the dependent variable.

To account for potential demand effects, we introduced the 'demand effects check' measure as a control variable for $H_{IA,B}$. If our participants were influenced by demand effects, we would have expected the variable to moderate the effectiveness of the psychological inoculations, such that participants who received a psychological inoculation and understood the experimental aim would have reported more positive affect towards climate action overall. We added the 'demand effects check' score (continuous, range from 0 to 4) as a fixed predictor as a main effect and a two-way interaction with condition. We conducted a second multilevel model within the passive disinformation control condition, to assess whether demand effects might influence the disinformation provision. We specified three random effects: an intercept for participant; an intercept for the internal numbering of the climate disinformation statements, to account for the variance associated with each disinformation statement[190]; and an intercept for country (factor, alphabetically coded), to account for the variance associated with each country[212]. We specified the following as fixed effects: 'demand effects check' (continuous, range from 0 to 4), trial (continuous variable, from 1 to 20) and the two-way interactions of trial with demand effects check. Only $N = 78$ (1.14%) participants reported having guessed the aim of the study. There was no evidence that potential demand effects moderated the effectiveness of the psychological inoculations or the disinformation provision (Supplementary Tables 13–15).

Finally, although the six psychological inoculations presented here were conceptualized as broad-spectrum inoculations[191], it was possible that the content of specific climate disinformation statements matched the thematical content of specific psychological inoculations more closely than others, and that this thematic match would have increased the protective effect of the psychological inoculation. To address this possibility, we manually coded whether specific climate disinformation statements are thematic matches with one of the different psychological inoculations (Supplementary Table 2). To compare the effectiveness of the psychological inoculation between matching and unmatching climate disinformation statements, we analysed changes in affect towards climate mitigation actions during the disinformation provision with four additional multilevel models, one for each psychological inoculation where we could identify at least one thematic match. We specified four random effects: a slope per trial, an intercept per participant, an intercept per climate disinformation statement and an intercept per country. We specified the following as fixed effects: condition (factor, two levels, specific psychological inoculation and passive disinformation control), trial (continuous variable, from 1 to 20) and the interaction between 'thematic match' (factor, two levels, matching and not matching) and condition. If a thematic match between climate disinformation statements and specific psychological inoculations did indeed increase the protective effects of the inoculation, we expected the interaction to be significant, and the simple slopes to highlight a significant difference between thematically matching and thematically non-matching climate disinformation statements in the inoculation condition, so that the difference in affect would have been smaller for climate disinformation statements that are thematic matches of the psychological inoculation.

## Protocol registration
The Stage 1 protocol for this Registered Report was accepted in principle on 20 April 2023. The protocol, as accepted by the journal, can be found at https://figshare.com/s/f431f656b53ec90396c0.

## Reporting summary
Further information on research design is available in the Nature Portfolio Reporting Summary linked to this article.

## Data availability
The anonymized data, Qualtrics files and stimuli are available on the Open Science Framework at https://osf.io/m58zx.

## Code availability
The R code necessary to reproduce our results is available on the Open Science Framework at https://osf.io/m58zx.

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

## Acknowledgements

We thank the Canton of Geneva and the Service Industriels de Genève for funding this project (grant no. UN10765 awarded to T.B., U.J.J.H. and E.T.). Moreover, U.J.J.H. thanks the Swiss National Science Foundation for providing individual career funding (SNSF Eccellenza PCEFPI_203283). We also thank C. Mumenthaler for help with the Twitter data collection and K. Doell for help on the WEPT version used in this study. The funders had no role in study design, data collection and analysis, decision to publish or preparation of the manuscript.

## Author contributions

T.S.: conceptualization, methodology, software, formal analysis, investigation, resources, data curation, writing (original draft), visualization and project administration. U.J.J.H.: conceptualization, methodology, resources, writing (review and editing), supervision and funding acquisition. E.T.: conceptualization, writing (review and editing), supervision and funding acquisition. T.B.: conceptualization, methodology, resources, writing (review and editing), supervision, funding acquisition and project administration.

## Competing interests

The authors declare no competing interests.

## Additional information

**Correspondence and requests for materials** should be addressed to Tobia Spampatti.

| | |
|---|---|

# Reporting Summary

## Statistics

For all statistical analyses, confirm that the following items are present in the figure legend, table legend, main text, or Methods section.

| n/a | Confirmed | |
|---|---|---|
| ☐ | ☒ | The exact sample size (*n*) for each experimental group/condition, given as a discrete number and unit of measurement |
| ☐ | ☒ | A statement on whether measurements were taken from distinct samples or whether the same sample was measured repeatedly |
| ☐ | ☒ | The statistical test(s) used AND whether they are one- or two-sided<br>*Only common tests should be described solely by name; describe more complex techniques in the Methods section.* |
| ☐ | ☒ | A description of all covariates tested |
| ☐ | ☒ | A description of any assumptions or corrections, such as tests of normality and adjustment for multiple comparisons |
| ☐ | ☒ | A full description of the statistical parameters including central tendency (e.g. means) or other basic estimates (e.g. regression coefficient) AND variation (e.g. standard deviation) or associated estimates of uncertainty (e.g. confidence intervals) |
| ☐ | ☒ | For null hypothesis testing, the test statistic (e.g. *F*, *t*, *r*) with confidence intervals, effect sizes, degrees of freedom and *P* value noted<br>*Give P values as exact values whenever suitable.* |
| ☐ | ☒ | For Bayesian analysis, information on the choice of priors and Markov chain Monte Carlo settings |
| ☐ | ☒ | For hierarchical and complex designs, identification of the appropriate level for tests and full reporting of outcomes |
| ☐ | ☒ | Estimates of effect sizes (e.g. Cohen's *d*, Pearson's *r*), indicating how they were calculated |

*Our web collection on statistics for biologists contains articles on many of the points above.*

## Software and code

Policy information about availability of computer code

| Data collection | We used the online Qualtrics XM Platform (2020 version) for data collection, accessed through institutional license dispensed to the corresponding author. |
|---|---|
| Data analysis | We used open software R (Version 4.1.3, 2023.03.0+386). Within the R environment, we used packages "cowplot" (Version 1.1.1), "ggdist" (Version 3.1.1), "lmerTest" (Version 3.1.3), "lme4" (Version 1.1-27.1), "TOSTER" (Version 0.3.4), "MuMIn" (Version 1.46.0), "sjmisc" (Version 2.8.9), "sjPlot" (Version 2.8.10), "effects" (Version 4.2.1), "car" (Version 3.1.1), "pscl" (Version 1.5.5), "DescTools" (Version 0.99.27), "forcats" (Version 0.5.1), "stringr" (Version 1.4.0), "dyplr" (Version 1.0.10), "purr" (Version 0.3.4), "readr" (Version 2.1.2), "tidyr" (Version 1.2.0), "tibble" (Version 3.1.6), "tidyverse" (Version 1.3.1), "gmodels" (Version 2.18.1), "psych" (Version 2.2.3), "mlogit" (Version 1.1.1), "Hmisc" (Version 4.6.0), "QuantPsych" (Version 1.5), "MASS" (Version 7.3.56), "boot" (Version 1.3.28), "sandwich" (Version 3.0.1), "cocor" (Version 1.1.3), "MSwM" (Version 1.5), "emmeans" (Version 1.8.2), "Rmisc" (Version 1.5), "plyr" (Version 1.8.8), "lattice" (Version 0.20.45), "raincloudplots" (Version 0.2.0), "ggplot" (Version 2_3.3.5), "MetBrewer" (Version 0.2.0), "r2glmm" (Version 0.1.2), "visreg" (Version 2.7.0), "glmTMB" (Version 1.1.7), "DHARMa" (Version 0.4.5). |

For manuscripts utilizing custom algorithms or software that are central to the research but not yet described in published literature, software must be made available to editors and reviewers. We strongly encourage code deposition in a community repository (e.g. GitHub). See the Nature Portfolio guidelines for submitting code & software for further information.

## Data

Policy information about availability of data

All manuscripts must include a data availability statement. This statement should provide the following information, where applicable:

- Accession codes, unique identifiers, or web links for publicly available datasets
- A description of any restrictions on data availability
- For clinical datasets or third party data, please ensure that the statement adheres to our policy

Anonymized data, Qualtrics files and stimuli are available on the OSF at https://osf.io/m58zx/?view_only=95fd430f4b7e4ee99c9c8b472e31d6b3.

## Research involving human participants, their data, or biological material

Policy information about studies with human participants or human data. See also policy information about sex, gender (identity/presentation), and sexual orientation and race, ethnicity and racism.

| | |
|---|---|
| Reporting on sex and gender | Only self-reported gender identity was collected (1-item: "Which gender do you identify with?" Responses: "Male"; "Female"; "non-binary / third gender"; "prefer not to answer". As per our preregistration, no participant was excluded from their response to this question. No sex data was collected. Gender self-report was utilized as a covariate in all multilevel models reported in the manuscript. |
| Reporting on race, ethnicity, or other socially relevant groupings | We did not collect categorization data for socially relevant groupings. |
| Population characteristics | Mean participant age = 39.15 ± 14.17, n=3555 female. Participants were quota-sampled by age and gender within twelve (12) countries: USA, Canada, UK, Ireland, Australia, New Zealand, Singapore, Philippines, India, Pakistan, Nigeria, and South Africa |
| Recruitment | Participants were recruited for the online survey by the panel provider Market Science Institute (https://site.msi-aci.com/). A potential selection bias that might impact the generizability of the results is that participation to the survey was conditional to having a device (laptop, tablet, or phone) with an internet connection. |
| Ethics oversight | The study was approved by the Ethics Committee of the University of Geneva (https://cureg.unige.ch/en/). |

Note that full information on the approval of the study protocol must also be provided in the manuscript.

## Field-specific reporting

Please select the one below that is the best fit for your research. If you are not sure, read the appropriate sections before making your selection.

☐ Life sciences  ☒ Behavioural & social sciences  ☐ Ecological, evolutionary & environmental sciences

For a reference copy of the document with all sections, see nature.com/documents/nr-reporting-summary-flat.pdf

## Behavioural & social sciences study design

All studies must disclose on these points even when the disclosure is negative.

| | |
|---|---|
| Study description | Quantitative experimental study. Mixed design. Participants were randomly assigned to one of eight different between-participants conditions: pure control (no inoculation, no disinformation), passive disinformation control (disinformation without inoculation), scientific consensus inoculation, trust in scientists inoculation, transparent communication inoculation, moralization of climate action inoculation, accuracy inoculation, and positive emotion inoculation. We chose a passive disinformation control condition over an active or positive control in order to better mimic real-life information environments, where climate disinformation is most frequently encountered passively and in multiple occurrences. Participants and experimenters were blind to the name and aim of the condition that participants were randomized into (double blind). The experiment contained twenty within-participants repeated measures of affect towards climate mitigation actions, assessed after each of the twenty climate disinformation statements. |
| Research sample | We collected one quota-based sample (age and gender) from each of twelve (12) countries: USA, Canada, UK, Ireland, Australia, New Zealand, Singapore, Philippines, India, Pakistan, Nigeria, and South Africa. We collected the sample with quota for gender and age from the panel provider Market Science Institute. The sample comprised of participants from twelve countries, n = 568 participants per country, for a total of N = 6816 (mean age = 39.15 ± 14.17, n=3555 female) participants. We choose these countries for two reasons. First, as all the real and validated climate disinformation material (see Table 3 in the manuscript) was in the English language, we selected these countries as they all list English as one of their official languages. Second, we selected these countries to have a broad representativeness across continents and beyond WEIRD samples. |
| Sampling strategy | We collected one quota-based sample (age and gender) from each of twelve (12) countries: USA, Canada, UK, Ireland, Australia, New |

| Sampling strategy | Zealand, Singapore, Philippines, India, Pakistan, Nigeria, and South Africa. |
|---|---|

We collected the sample with quota for gender and age from the panel provider Market Science Institute. The sample comprised of participants from twelve countries, n = 568 participants per country, for a total of N = 6816 (mean age = 39.15 ± 14.17, n=3555 female) participants. We identified the required sample size a-priori, with G*Power (Version 3.0), in order to have 95% power to detect a difference between any intervention condition and the passive disinformation control condition of δ = 0.20 in a one-tailed t-test with α = .005, for all main hypotheses separately. We selected the smallest effect size of interest (SESOI) from the lower bound of the confidence interval of the meta-analytically identified effect size of fact-checking interventions on political topics, as we reasoned that a new disinformation intervention would be of interest if and only if it has an effect that is larger than already available interventions such as fact-checks. Incidentally, a recent paper showed that the effects of more established psychological inoculations on sharing intentions of manipulative content is δ = 0.20, increasing our confidence in the practical interest of this SESOI.

**Data collection**

Participation was entirely online, the survey being accessed through an anonymous link.
Participants and experimenters were blind to the name and aim of the condition that participants were randomized into (double blind).
Procedure: Participants accessed the survey through an anonymous link made available by the panel provider, and provided their explicit consent to the study. After consenting, participants reported their demographics (gender, age, education, and political orientation: single-item, 10-point scale: 1=[Extreme liberalism/left] to 10=[Extreme conservativism/right]), completed a baseline measure of affect towards climate action, and completed the Cognitive Reflection Task, Version 2, in random order. A two-strikes-out attention check ([Please select "3" to make sure you are paying attention]) was presented; failing it triggered a warning with a 10-seconds time penalty. Inattentive participants received the attention check for a second time, and participants found inattentive again afterwards were screened out of the survey (n=16). Attentive participants were then randomly allocated to one of the eight conditions and received the assigned intervention or, for participants in the passive disinformation control condition, were directly forwarded to the following section of the experiment. Participants in the pure control condition received neither the interventions nor the climate disinformation statements. All interventions were presented sequentially in four screens, with a 5-20s time lock (depending on the content length of each screen) that did not allow participants to manually proceed to the next screen until the time had elapsed. A manipulation check measuring participants' motivation to resist persuasion followed. Afterwards, participants received twenty real climate disinformation statements in form of anonymous tweets, in randomized order with a 2s time lock, and report their affect towards actions to mitigate climate change after each disinformation statement. Following the disinformation provision, participants completed the climate change perceptions scale, a modified version of the Working for Environmental Protection Task, and the truth discernment task, all described in full detail below. Finally, we probed participant's understanding of the experimental aims with an open-ended question to account for potential demand effects. The survey ended with an extended debriefing that contained a reminder of the scientific consensus behind climate change with a link to the latest IPCC report. Survey duration was about 21 minutes.

**Timing**

Data collection started the 15th of May, 2023, and terminated on the 27th of May, 2023.

**Data exclusions**

No participant was excluded from data analysis.

**Non-participation**

In total, n = 1171 participants dropped out/declined participation (USA n = 113, Canada n = 94, UK n = 143, Ireland n = 113, Australia n = 104, New Zealand n = 66, Singapore n = 58, Philippines n = 109, India n = 94, Pakistan n = 128, Nigeria n = 6, and South Africa n = 143).

**Randomization**

Participants were randomly assigned to one of eight different between-participants conditions: pure control (no inoculation, no disinformation), passive disinformation control (disinformation without inoculation), scientific consensus inoculation, trust in scientists inoculation, transparent communication inoculation, moralization of climate action inoculation, accuracy inoculation, and positive emotion inoculation.

# Reporting for specific materials, systems and methods

We require information from authors about some types of materials, experimental systems and methods used in many studies. Here, indicate whether each material, system or method listed is relevant to your study. If you are not sure if a list item applies to your research, read the appropriate section before selecting a response.

## Materials & experimental systems

| n/a | Involved in the study |
|---|---|
| ☒ | ☐ Antibodies |
| ☒ | ☐ Eukaryotic cell lines |
| ☒ | ☐ Palaeontology and archaeology |
| ☒ | ☐ Animals and other organisms |
| ☒ | ☐ Clinical data |
| ☒ | ☐ Dual use research of concern |
| ☒ | ☐ Plants |

## Methods

| n/a | Involved in the study |
|---|---|
| ☒ | ☐ ChIP-seq |
| ☒ | ☐ Flow cytometry |
| ☒ | ☐ MRI-based neuroimaging |

