## [Peer Review File · Nature Human Behaviour]

Peer Review Information

Journal: Nature Human Behaviour

Manuscript Title: Psychological inoculation strategies to fight climate disinformation across 12 countries

Corresponding author name(s): Tobia Spampatti

Reviewer Comments & Decisions:

Decision Letter, initial version:

3rd February 2023

Dear Mr Spampatti,

Thank you once again for your manuscript, entitled "Psychological inoculation strategies to fight climate disinformation around the world," and for your patience during the peer review process.

Your manuscript has now been evaluated by 4 reviewers, whose comments are included at the end of this letter. Although the reviewers find your protocol to be of interest, they also raise some important concerns. We are very interested in the possibility of proceeding further with your submission in Nature Human Behaviour, but would like to consider your response to these concerns in the form of a revised manuscript before we make a decision on in principle acceptance and Stage 2 submission.

To guide the scope of the revisions, the editors discuss the referee reports in detail within the team, including with the chief editor, with a view to (1) identifying key priorities that should be addressed in revision and (2) overruling referee requests that are deemed beyond the scope of the current study. We hope that you will find the prioritised set of referee points to be useful when revising your study. Please do not hesitate to get in touch if you would like to discuss these issues further.

1. All reviewers suggest changes to the experimental design to ensure the reliability and validity of your findings. They make several suggestions for additional (control) conditions and measures. While we appreciate that not all of the suggested changes are feasible, we believe that some of the concerns are fundamental. Specifically, we ask that you follow the advice of Reviewers 2 and 4, and include a truth discernment measure and a pure control condition to fully assess the specific effects of misinformation and to ensure that the disinformation messages are indeed persuasive. Furthermore, Reviewer 4 requests that, in addition to a behavioural measure, you include a measure that assesses that the inoculation treatments will help people resist or identify misinformation specifically.

2. In addition, Reviewers 2 and 3 raise important concerns about the possibility of demand-effects and potential confounds. Please revise your methodology to specifically address these concerns.

3. Finally, our reviewers raise several questions about the methodological and statistical choices. In your revised manuscript, please ensure that all methodological and statistical choices are explicitly motivated.

In sum, we invite you to revise your Stage 1 Registered Report taking into account reviewer and editor comments. Please highlight all changes in the manuscript text file.

* Include a "Response to reviewers" document detailing, point-by-point, how you addressed each referee comment. If no action was taken to address a point, you must provide a compelling argument. This response will be sent back to the reviewers along with the revised manuscript.

* Ensure that you use our template for Stage 1 Registered Reports to prepare your revised manuscript: https://www.nature.com/documents/NHB_Template_RR_Stage1.docx. Failure to ensure that your revised Stage 1 submission meets our requirements as specified in the template will result in your submission being returned to you, which will delay its consideration.

* In your cover letter, please include the following information:

--An anticipated timeline for completing the study if your Stage 1 submission is accepted in principle.

--A statement confirming that you agree to share your raw data, any digital study materials, computer code (if relevant), and laboratory log for all eventually published results.

--A statement confirming that, following Stage 1 in principle acceptance, you agree to register your approved protocol on the Open Science Framework (<https://osf.io/>) or other recognised repository, either publicly or under private embargo, until submission of the Stage 2 manuscript.

--A statement confirming that if you later withdraw your paper, you agree to the Journal publishing a short summary of the pre-registered study under a section Withdrawn Registrations.

[REDACTED]

We hope to receive your revised manuscript within four to eight weeks. If you cannot send it within this time, please let us know. We will be happy to consider your revision so long as nothing similar has been accepted for publication at Nature Human Behaviour or published elsewhere.

Nature Human Behaviour is committed to improving transparency in authorship. As part of our efforts in this direction, we are now requesting that all authors identified as 'corresponding author' on published papers create and link their Open Researcher and Contributor Identifier (ORCID) with their account on the Manuscript Tracking System (MTS), prior to acceptance. ORCID helps the scientific community achieve unambiguous attribution of all scholarly contributions. You can create and link your ORCID from the home page of the MTS by clicking on 'Modify my Springer Nature account'. For more information please visit <http://www.springernature.com/orcid>.

Sincerely,

Samantha Antusch

Samantha Antusch, PhD
Senior Editor
Nature Human Behaviour

Reviewer expertise:

Reviewer #1: open science and RRs

Reviewer #2: (climate change) misinformation ; interventions

Reviewer #3: open science and RRs

Reviewer #4: (climate change) misinformation ; interventions

Reviewers' Comments:

Reviewer #1:

Remarks to the Author:

In "Psychological inoculation strategies to fight climate disinformation around the world" the authors propose a worthwhile and theoretical justified attempt to inoculate participants against climate disinformation. Not only is this topic of high interest and relevance, knowing whether- or not- these techniques are effective will be worthwhile to policy makers. Below are my comments and suggestions, I hope that the editors and authors find them helpful.

As a matter of personal policy, I sign my reviews:
David Mellor, PhD

Director of Policy, Center for Open Science
<https://orcid.org/0000-0002-3125-5888>

Two of the three most important features of the authors' registered report are well constructed: the quality control steps, mentioned below, and the proposed analysis plan. The third feature, theoretical applicability, I am least able to judge (as my professional training is outside of the field)- HOWEVER, I do feel obligated to comment on what I see as a theoretical oversight that could lead to another intervention. My understanding is that one's social circle is highly relevant to belief in climate change; ie if one's peers have bought into disinformation then they are much more likely to do some themselves. A quick literature search uncovered this, for example:

Stevenson KT, Peterson MN, Bondell HD. The influence of personal beliefs, friends, and family in building climate change concern among adolescents. *Environmental Education Research*. 2019;25(6):832-845. doi:10.1080/13504622.2016.1177712

I believe (and suspect that many readers will likewise believe) that this is likely an important driver of this problem. To address this, I would request that the authors come up with another inoculation. If the authors can have ethical clearance to deceive, I'd suggest that the manipulation tell the participant that the vast majority of people in their country (being specific to the country that they reside) see disinformation as false. If the authors do not have the ability to deceive participants, then asking the participant to imagine their closest friends and family taking some climate-friendly action. It's possible that either of those inoculations may be insufficient to actually shift one's image of their peers' opinions, and would welcome the authors' suggestions about other methods. IF the authors believe that such a manipulation is not possible, or is too far outside of their proposed theoretical framework, I would still like them to address this plausible explanation. Finally, if they do add such an inoculation, as I recommend, then please address the likely need for a larger sample size (or remove one of the existing 6- although I do not necessarily recommend doing so).

Quality controls:

The authors provide a comprehensive set of rules for quality control- namely the attention checks and all possible outcomes of the procedure failing the manipulation check. These procedures (as long as sufficient numbers pass the attention checks) ensure the interpretability and utility of results even in the event that none show promise in reducing the harm of disinformation- which will be as important to policy makers as a finding that one particular strategy ends up being a magic bullet against disinformation (which I expect will not happen because of my own pessimism on this issue).

Data and code availability statements are appropriate.

Reviewer #2:

Remarks to the Author:

Thank you for the opportunity to review this study proposal manuscript titled "Psychological inoculation strategies to fight climate disinformation around the world". Overall, I believe the work proposed here would provide a significant contribution to the literature, especially with regards to the theoretical framework detailed at the beginning. I have a number of recommendations that I believe

would strengthen the validity and applicability of the findings once collected, and I hope you will consider including them. These are detailed below:

1) Recipients of scientific messages cells

I am very fond of your conceptualisation for the use of inoculation to change the internal states of participants, specifically hoping to induce stronger analytical thinking and positive emotions in participants. However, I would like to see some more justification for these two variables, especially over the inclusion of other possible options.

First, while accuracy nudges do appear to hold some weight, I believe mixing the theoretical ingredients of inoculation in with this poses a different question. Essentially, you are inoculating against intuitive thinking. While this is not necessarily missing from your explanation, my point is that I believe more consideration should be given to how exactly this might be the same/different to the classic accuracy nudge approach, considering the established effectiveness of both approaches separately.

Second, while I like the approach of inducing positive emotions in participants, I can't help but wonder whether there may be a more effective strategy for this theoretical cell (table 1). You back-up the inclusion of positive emotions well, but I am interested in other—I suppose more motivational—variables. For example, what about inoculation to reduce anxiety in this context (see Jackson et al., 2017, for an example). Basically, my point here is that I am wondering whether there are more influential (and specific) variables that could be addressed here than a general sense of positive emotion.

2) The experimental stimuli

Having looked over your proposed inoculation stimuli, I am satisfied that they definitely include the appropriate ingredients of inoculation. However, I would like to raise a concern about the standardised introduction texts to "minimize any differences between experimental conditions...". While this should certainly improve validity, I do wonder whether including information about how "...it is unequivocal that anthropogenic climate change is happening..." introduces a confound. Specifically, I am concerned that mentioning the unequivocal evidence on climate change communicates a very similar piece of information to the consensus inoculation message. My worry here is that you may not be able to determine whether your other inoculations worked in their own right, risking simply improving participants' perceptions of scientific consensus in all conditions through this standardised introduction.

3) Measures

I can understand why you would include measures of intentions to plant trees and affective perceptions of the disinformation, but I do wonder whether you are losing some very important information by not including measures of truth discernment alongside your disinformation stimuli. That is, it may be conceptually possible to have a negative affect about disinformation, but also believe it to be true. While this is of course unlikely, I would recommend the measurement of truth discernment alongside your other measures to more robustly assess the efficacy of your interventions. On this note, I would recommend a more efficient strategy of creating your disinformation attack stimuli. Maertens et al. (2022) have created and psychometrically validated a Misinformation Susceptibility Test (MIST), which presents participants with AI-generated fake and real news posts, and asks participants to determine whether they are "fake" or "real" on a binary choice scale. This would mean you can avoid the time-consuming process of combing the Twitter posts, alongside including a validated measure of misinformation susceptibility.

4) The measurement and alteration of CRT/accuracy

Recent work (Roozenbeek et al., 2022) has called the role of CRT in misinformation susceptibility into question. Specifically, it appears that actively open-minded thinking (AOT) accounts for greater variance in the cognitive mechanisms associated with accuracy that make people more/less susceptible to misinformation. Therefore, I would urge you to reconsider your inoculation with regards to accuracy, at least including a measure of AOT to address these contemporary findings.

References

Jackson, B., Compton, J., Thornton, A. L., & Dimmock, J. A. (2017). Re-thinking anxiety: Using inoculation messages to reduce and reinterpret public speaking fears. *PLOS ONE*, 12(1), e0169972. <https://doi.org/10.1371/journal.pone.0169972>

Maertens, R., Götz, F. M., Golino, H., Roozenbeek, J., Schneider, C. R., Kerr, J. R., ... Linden, S. (2021, July 6). The Misinformation Susceptibility Test (MIST): A psychometrically validated measure of news veracity discernment. <https://doi.org/10.31234/osf.io/gk68h>

Roozenbeek, J., Maertens, R., Herzog, S. M., Geers, M., Kurvers, R., Sultan, M., & Van der Linden, S. (2022). Susceptibility to misinformation is consistent across question framings and response modes and better explained by myside bias and partisanship than analytical thinking. *Judgment and Decision Making*, 17(3), 547-573. <https://doi.org/10.1017/s1930297500003570>

Reviewer #3:

Remarks to the Author:

Thank you for the invitation to review this Stage 1 Registered Report. The manuscript firstly presents a theoretical framework describing various factors influencing belief formation and updating in the context of climate change. Six "inoculation strategies" that may protect individuals from climate change disinformation are then created, based on this framework. Finally, the authors propose a study to test the effectiveness of these strategies. The basic design of the study is as follows: (1) participants will be randomly assigned to receive one of the six inoculation strategies or a passive control group; (2) each participant will be presented with 20 real examples of climate change disinformation; (2) after each disinformation message is presented, participants will self-report their personal affect towards climate change; (3) after all disinformation messages are presented, participants will perform a pro-environmental behavioural task.

Firstly, I should highlight that although I am a psychologist with expertise in methodology, I do not have expertise in the general domain of climate change messaging. Below I have responded to the journal's prompts for reviewers.

The significance of the research question(s) and relevance for a broad, multidisciplinary audience

The study addresses an important topic with broad, multidisciplinary appeal.

The extent to which the proposed study can satisfactorily answer the research question(s)

I think this is a well designed study that can provide a useful answer to the research questions. The use of simple interventions and real disinformation will enhance the real-world applicability of the results.

The logic, rationale, and plausibility of the proposed hypotheses

Everything seems fine here. Everything is clearly presented.

The soundness and feasibility of the methodology and analysis pipeline (including statistical power analysis)

- Generally the study appears to be well designed. I have a few comments/questions:
- I'm quite concerned about demand characteristics - imagining myself in a participant's shoes, it seems pretty obvious what the researchers are trying to do and how they want me to respond. I'm not sure how that could be best addressed. Perhaps a more natural task could be created that doesn't make the intervention and topic of interest so obvious, for example, embedding the climate-change related information in a fake Twitter timeline consisting of other non-climate information and asking users to scroll through.
- Related to the above, the current design uses a passive control condition (participants skip straight to the disinformation messages). I wonder if an active control condition - which could have the benefit of reducing participant reactivity effects (e.g., Hawthorne effects) - is feasible in this context. For example, perhaps control participants could read a general (non-climate related) news message? It could be that the authors prefer a passive control because it more closely resembles the practical circumstances in which inoculation strategies will be deployed. Either way, perhaps the rationale for the control group design could be included in the text?
- Its not clear to me why each participant views 20 disinformation messages and responds to the the affect question after each. It would be good to hear more about the rationale for that. An alternative would be to give each participant a single randomly selected disinformation message.
- One of the exclusion criteria is "iv) respond with the same response for more than 90% of the questions". What is the justification for this criterion? It seems reasonable to me that a participant might respond consistently if they are not influenced by the disinformation messages.

Whether the clarity and degree of methodological detail would be sufficient to replicate exactly the proposed experimental procedures and analysis pipeline

Yes, it appears so.

Whether the authors provide a sufficiently clear and detailed description of the methods to prevent undisclosed flexibility in the experimental procedures or analysis pipeline

Yes it appears so.

Whether the authors have considered sufficient outcome-neutral conditions (e.g. absence of floor or ceiling effects; positive controls) for ensuring that the results obtained are able to test the stated hypotheses

Yes, this seems satisfactory.

Reviewer #4:

Remarks to the Author:

I very much enjoyed reviewing this Registered Report. I think the topic of climate disinformation is of obviously of great societal importance and this is by far the most comprehensive assessment of the potential of psychological inoculation against climate disinformation I've seen. I appreciated the novel theoretical framework and the high level of technical detail provided in the RR, which allows for a proper assessment of its overall value.

I am definitely supportive of this work, I think it has great value for the literature across multiple disciplines (climate, behaviour, misinformation etc) but I do see some areas of potential concern that I think could benefit from further reflection as well as some more minor comments:

(1) The first is a potential mismatch between the level at which the intervention operates (inoculation) and the dependent variable (climate action) and whether this constitutes a fair test of the intervention. For example, the literature is pretty silent on the extent to which inoculation can actually move behavior. Personally, I am not sure about the extent to which inoculation is going to protect behavioral action. Inoculation interventions are typically geared at (a) relevant beliefs and attitudes and (b) helping people spot the misleading argument or manipulation attempt. At present, the authors are not measuring either. I appreciate the manipulation check about motivation to resist persuasion (to see if the inoculation worked as intended) but I did not see any measures of whether inoculation will help people resist or identify misinformation specifically, which is the level at which inoculation usually operates.

(2) Relatedly, there should be a match between the content of the inoculation message and the disinformation attack, otherwise we're not testing inoculation as a main effect but rather the extent to which it can function as an "umbrella" or "broad-spectrum" vaccine insofar it protects against a range of disinformation not necessarily mentioned in the inoculation itself. For example, if you inoculate someone against X, and then attack them with Y, can we really conclude anything about inoculation if it didn't work? I.e. if people don't have the right mental defences you cannot expect them to generate immunity, especially at the behavioral level. Maybe the authors have already thought about this, but one helpful revision could be to link the Twitter disinformation messages to the relevant inoculation condition in the Table so that it is clear which disinformation corresponds to which hypothesis/condition. For example, I saw some disinformation about the scientific consensus, which would nicely match with the consensus inoculation condition etc. I think this should be clarified better for the disinformation messages. Having said this, it's not a problem if you want to test how far the inoculation stretches across different issues but this should then be explicitly recognized as a secondary hypothesis.

(3) I wondered about the need for a pure control condition to establish that the passive fact-check and inoculation work as intended? Moreover, inoculation can only be effective if the disinformation is persuasive in the first place otherwise there is nothing to inoculate against, so is it not important to illustrate that disinformation had a negative effect on the DVs in a pure control condition (no inoculation, no fact-check)? If misinformation does nothing, then there's nothing to inoculate against.

The authors say they pilot tested the disinformation tweets, but in order to not waste a huge amount of resources, they might at least want to check whether the disinformation is indeed negatively influencing the key DVs before proceeding.

(4) The forewarning seems to be taken literally from van der Linden et al. (2017), which the authors may want to mention insofar it helps provide prior validation of this wording.

(5) The power analysis seems correct to me, but, 500 per country, with 7 conditions, breaks down to about 70 people per condition, which is not that high. Especially if another pure control baseline is added. Is there any way to oversample?

(6) It is not mentioned whether the panel is representative of the population in each country or whether quotas will be used or just a standard panel.

(7) For a multi-level model, 12 countries is not a lot, some argue that if the participant-level n is large compared to the second-level (n) it can bias estimates, though this mostly seems to concern the country-level comparisons (see Bryan & Jenkins, 2016). Perhaps worth considering.

(8) The authors may find recent work about transparency, inoculation, and trust of interest to motivate the transparency condition (see Kerr et al., 2023).

(9) I wonder, if, in the context of climate change, it is useful to include ideology as a factor as well to establish whether politics reduces the effect of the inoculation condition in countries with known polarization on the issue vs others that are perhaps less polarized on climate change?

Refs

Bryan, M. L., and S. P. Jenkins. 2016. "Multilevel Modelling of Country Effects: A Cautionary Tale." *European Sociological Review* 32 (1): 3–22. doi:10.1093/esr/jcv059.

Kerr, J. R., Schneider, C. R., Freeman, A. L., Marteau, T., & van der Linden, S. (2022). Transparent communication of evidence does not undermine public trust in evidence. *PNAS Nexus*.

Author Rebuttal to Initial comments

Reviewer #1:

Remarks to the Author:

In “Psychological inoculation strategies to fight climate disinformation around the world” the authors propose a worthwhile and theoretical justified attempt to inoculate participants against climate disinformation. Not only is this topic of high interest and relevance,

knowing whether- or not- these techniques are effective will be worthwhile to policy makers. Below are my comments and suggestions, I hope that the editors and authors find them helpful.

As a matter of personal policy, I sign my reviews:

David Mellor, PhD

Director of Policy, Center for Open Science

<https://orcid.org/0000-0002-3125-5888>

Two of the three most important features of the authors' registered report are well constructed: the quality control steps, mentioned below, and the proposed analysis plan. The third feature, theoretical applicability, I am least able to judge (as my professional training is outside of the field)- HOWEVER, I do feel obligated to comment on what I see as a theoretical oversight that could lead to another intervention. My understanding is that one's social circle is highly relevant to belief in climate change; ie if one's peers have bought into disinformation then they are much more likely to do some themselves. A quick literature search uncovered this, for example:

Stevenson KT, Peterson MN, Bondell HD. The influence of personal beliefs, friends, and family in building climate change concern among adolescents. *Environmental Education Research*. 2019;25(6):832-845. doi:10.1080/13504622.2016.1177712

I believe (and suspect that many readers will likewise believe) that this is likely an important driver of this problem. To address this, I would request that the authors come up with another inoculation. If the authors can have ethical clearance to deceive, I'd suggest that the manipulation tell the participant that the vast majority of people in their country (being specific to the country that they reside) see disinformation as false. If the authors do not have the ability to deceive participants, then asking the participant to imagine their closest friends and family taking some climate-friendly action. It's possible that either of those inoculations may be insufficient to actually shift one's image of their peers' opinions, and would welcome the authors' suggestions about other methods. IF the authors believe that such a manipulation is not possible, or is too far outside of their proposed theoretical

framework, I would still like them to address this plausible explanation. Finally, if they do add such an inoculation, as I recommend, then please address the likely need for a larger sample size (or remove one of the existing 6- although I do not necessarily recommend doing so).

We would like to thank the reviewer for their expert appraisal of the methodological and analytical steps we have taken to ensure the quality, openness, and future reproducibility of our proposed contribution. In what concerns the suggested addition of another psychological inoculation to the proposed study, we would first like to point out that this comment highlights one of the strengths of the theoretical framework proposed in the manuscript: indeed, many additional inoculation strategies can be developed by the structure of the framework. The inoculation suggested by the reviewer is indeed promising, as second-order beliefs have been shown to influence relevant attitudes (Jost, 2018; Mildenberger & Tingley, 2019; Ruggeri et al., 2021). The suggested inoculation could be defined as a “pluralistic ignorance” inoculation aiming to change the perception that salient social groups believe in climate disinformation (Altay & Acerbi, 2022; Leviston et al., 2013). Within the framework proposed here, we would localize this inoculation in the cognitive pathway of the “sources of scientific messages”, structurally not unlike the *scientific consensus inoculation*, providing diagnostic information as to what salient social groups believe regarding climate science (Jost, 2018; Matthes et al., 2022; Ruggeri et al., 2021). With this in mind, we propose to add this suggestion (and an acknowledgement to Dr. Mellor) within a “future directions” paragraph in the discussion of the Stage 2 manuscript, illustrating that additional inoculations, including social-norm-or-belief-based interventions as proposed by Reviewer 1, can be developed based on the framework.

We hesitate to add an additional inoculation to our current experimental design for several reasons. First, adding yet another inoculation would require investing more resources, which have already been taxed by adding the pure control condition and the truth discrimination task. Second, we would not be able to replace the *scientific consensus inoculation* (which is located in the same cell of the theoretical framework) with the inoculation suggested by the reviewer, as the scientific consensus inoculation is the reference intervention for psychological inoculations against climate disinformation and thus needs to be kept in the design (Lewandowsky, 2021). As we are proposing a theoretical framework with 6 cells, testing 7 inoculations would not be balanced, and not as elegant as a symmetric approach. Third, ethically speaking, we would be concerned about deceiving our participants with the normative information suggested by the reviewer. However, a follow-up study with normative information about country-specific resistance to climate disinformation taken directly from the findings of the current study is

possible, and could thus be designed with evidence-based statistics on second-order beliefs without deceiving participants at a later stage.

Quality controls:

The authors provide a comprehensive set of rules for quality control- namely the attention checks and all possible outcomes of the procedure failing the manipulation check. These procedures (as long as sufficient numbers pass the attention checks) ensure the interpretability and utility of results even in the event that none show promise in reducing the harm of disinformation- which will be as important to policy makers as a finding that one particular strategy ends up being a magic bullet against disinformation (which I expect will not happen because of my own pessimism on this issue).

We thank the reviewer for their positive assessment of our quality controls, and we hope our findings will prove their pessimism wrong!

Data and code availability statements are appropriate.

Reviewer #2:

Remarks to the Author:

Thank you for the opportunity to review this study proposal manuscript titled "Psychological inoculation strategies to fight climate disinformation around the world". Overall, I believe the work proposed here would provide a significant contribution to the literature, especially with regards to the theoretical framework detailed at the beginning. I have a number of recommendations that I believe would strengthen the validity and applicability of the findings once collected, and I hope you will consider including them. These are detailed below:

We are grateful to the reviewer for the positive assessment of our project and of the proposed theoretical framework.

1) Recipients of scientific messages cells

I am very fond of your conceptualisation for the use of inoculation to change the internal states of participants, specifically hoping to induce stronger analytical thinking and positive emotions in participants. However, I would like to see some more justification for these two variables, especially over the inclusion of other possible options.

First, while accuracy nudges do appear to hold some weight, I believe mixing the theoretical ingredients of inoculation in with this poses a different question. Essentially, you are inoculating against intuitive thinking. While this is not necessarily missing from your explanation, my point is that I believe more consideration should be given to how exactly this might be the same/different to the classic accuracy nudge approach, considering the established effectiveness of both approaches separately.

We understand the reviewer's need for clarification concerning our version of the accuracy intervention in light of the changes from the classic prompt/nudge format. Our reasoning was methodologically motivated, as we aimed to keep the different interventions as similar as possible to remove potential confounds regarding delivery modality (psychological inoculations vs. prompt/nudge format; see Lorenz-Spreen et al., 2021 for a related discussion), and participants' engagement with the intervention material (active vs. passive; see Roozenbeek & van der Linden, 2019). In order to address the first confound, we opted to harmonize the delivery of all interventions and present them as psychological inoculations. To address the second confound, we chose a passive psychological inoculation format as the original authors had already implemented the accuracy prompt as a passive intervention in their Twitter field study, which curbed the spread of misinformation on social media (Pennycook et al., 2021). We now explicitly mention the changes made to the classic accuracy prompt/nudge when introducing the accuracy inoculation in the methodology section (p. 14):

In the accuracy inoculation, we reframed the original accuracy prompt^[136] into a passive psychological inoculation^[182] where participants will be prompted to consider accuracy when evaluating the information

Second, while I like the approach of inducing positive emotions in participants, I can't help but wonder whether there may be a more effective strategy for this theoretical cell (table 1). You back-up the inclusion of positive emotions well, but I am interested in other—I suppose more motivational—variables. For example, what about inoculation to reduce anxiety in this context (see Jackson et al., 2017, for an example). Basically, my point here is that I am wondering whether there are more influential (and specific) variables that could be addressed here than a general sense of positive emotion.

We thank the reviewer for their suggestion and for highlighting, as also Reviewer 1 did, that one of the strengths of our proposed theoretical framework is that multiple additional inoculations can be generated based on the framework. Indeed, we agree that multiple other emotional variables can be addressed in the context of psychological inoculations, climate anxiety being one of them. Two arguments make us hesitant to design an inoculation aiming to reduce climate anxiety. First, lowering climate anxiety requires that participants already are in a state of climate

anxiety that hinders pro-environmental behavior and that increases susceptibility to climate disinformation. Cross-cultural evidence of the prevalence of climate anxiety is mounting (e.g., Ickman et al., 2021; Leiserowitz et al., 2021), but its prevalence in the countries we are focusing on is not understood well enough, especially when it comes to the predicted detrimental effects on our DVs. Based on the available information about the distribution of climate anxiety, on average one quarter of the population reports not being specifically anxious or worried about climate change (Ickman et al., 2021), thus a climate anxiety inoculation would a priori be expected to not work for a significant portion of the sample. Second, the causal link between climate anxiety and pro-environmental behavior is not completely clear. Although anxiety might be detrimental in public speaking situations (Jackson et al., 2017), it has been argued that feeling anxious about climate change can drive pro-environmental friendly behavior (Verplanken et al., 2020), and that lowering anxiety might thus also lower pro-environmental behavior. Overall, the debate about the role of anxiety and other emotions in the context of climate action is not settled yet (Brosch, 2021). However, a substantial number of recent reviews all independently highlight the potential of positive emotions to motivate climate action (Brosch, 2021; Brosch & Steg, 2021; Schneider et al., 2021; Shiota et al., 2021) which we now more prominently reference in the revised manuscript (p. 9):

Moreover, multiple recent reviews^[145; 147; 155-156] argue that the anticipation and experience of positive emotions elicited by acting pro-environmentally^[145; 155-158] increases pro-environmental behavioral intentions as well as actual behavior^[158-160]. To protect against disinformation at the *recipient* basis via the socio-affective pathway, the saliency of experienced positive emotions in the context of climate action can be increased, which should increase resistance to disinformation as well as likelihood to act pro-environmentally.

2) The experimental stimuli

Having looked over your proposed inoculation stimuli, I am satisfied that they definitely include the appropriate ingredients of inoculation. However, I would like to raise a concern about the standardised introduction texts to "minimize any differences between experimental conditions...". While this should certainly improve validity, I do wonder whether including information about how "...it is unequivocal that anthropogenic climate change is happening..." introduces a confound. Specifically, I am concerned that mentioning the unequivocal evidence on climate change communicates a very similar piece of information to the consensus inoculation message. My worry here is that you may not be able to determine whether your other inoculations worked in their own right, risking simply improving participants' perceptions of scientific consensus in all conditions through this standardised introduction.

Thank you for catching this, we agree that the phrase “...it is unequivocal that anthropogenic climate change is happening...” which is present in the introductory part of all inoculations indeed introduces a confound in that it makes reference to the scientific consensus. We have addressed this by adapting the introductory text and replacing “*is unequivocal that anthropogenic climate change is happening*” with “*has declared that anthropogenic climate change is happening*”.

3) Measures

I can understand why you would include measures of intentions to plant trees and affective perceptions of the disinformation, but I do wonder whether you are losing some very important information by not including measures of truth discernment alongside your disinformation stimuli. That is, it may be conceptually possible to have a negative affect about disinformation, but also believe it to be true. While this is of course unlikely, I would recommend the measurement of truth discernment alongside your other measures to more robustly assess the efficacy of your interventions. On this note, I would recommend a more efficient strategy of creating your disinformation attack stimuli. Maertens et al. (2022) have created and psychometrically validated a Misinformation Susceptibility Test (MIST), which presents participants with AI-generated fake and real news posts, and asks participants to determine whether they are "fake" or "real" on a binary choice scale. This would mean you can avoid the time-consuming process of combing the Twitter posts, alongside including a validated measure of misinformation susceptibility.

We thank the reviewer for these important considerations. Before engaging with the main point raised by the reviewer, we would like to clarify that we are not measuring “intentions to plant trees”, but that one tree will actually be planted by the Eden Reforestation Project for each WEPT page completed by the participants. It is therefore a valid and validated measure of actual pro-environmental behavior with concrete environmental consequences (Lange, 2022). In the current revision, we have slightly modified the task taking into account a recent publication that adapted the experimental task to better capture between-participants differences in experimental designs (Lange & Truyens, 2022).

As suggested by the reviewer, we have now added a climate-specific measure of truth discernment to the experimental sequence to further unpack the different effects that inoculations

may have in the context of climate disinformation. Building upon the strategy used by Maertens and colleagues to develop the Misinformation Susceptibility Test (Maertens and colleagues, 2022), we have likewise engaged with OpenAI ChatGPT to create a truth discernment task that is specific to the climate domain and describe the task and its creation in the revised version of the manuscript (pp. 21-23).

Truth discernment task: Inspired by a preprint presenting a measure of domain-general news veracity discernment ^[165] we developed a climate-specific truth discernment task in which participants have to categorize 20 statements mentioning climate-related topics as either false or real statements ([Please categorize the following statements as either “False Statement” or “Real Statement”]; Binary choice: [Real]; [False], item and response order randomized.). These 20 statements are equally divided between true and false headlines and between supporting or opposing climate science and action. All statements were generated interacting with an AI tool (ChatGPT by OpenAI). Over 300 true and false statements mentioning climate change or climate mitigation actions were initially created. The statements were then fact-checked and condensed into a longlist out of which 10 true and 10 false statements were selected to be included in the truth discernment task. The final statements are presented in Table 4 (the full list of generated statement is available in the OSF repository).

Table 4– Truth discrimination task generated climate headlines.

Coding	Climate-relevant news headline
True_Supporting_1	Earth's average temperature continues to rise, setting new record highs each decade.
True_Supporting_2	Human activities, such as burning fossil fuels, are the main cause of climate change.
True_Supporting_3	Climate change is leading to more intense and frequent natural disasters.
True_Supporting_4	The transportation sector is a significant contributor to greenhouse gas emissions.
True_Supporting_5	Rising seas could displace hundreds of millions of people by the end of the century.

True_Delaying_1	Projections of Regional Impacts of Climate Change are Subject to Uncertainty
True_Delaying_2	Transportation Sector Transition to Electric Vehicles Can Cost Billions in Infrastructure Upgrades
True_Delaying_3	Brazil Missing Paris Agreement Targets with Deforestation and Agricultural Expansion Driving Up Emissions
True_Delaying_4	China's continued construction of coal-fired power plants threatens progress on climate goals.
True_Delaying_5	Developing Countries Require \$40 Billion Annually to Mitigate Climate Change
False_Supporting_1	Climate change will cause the extinction of up to 75% of all species on Earth.
False_Supporting_2	Global temperatures may rise by up to 20°C by the end of the century, potentially resulting in widespread drought and famine due to climate change.
False_Supporting_3	The Earth may enter a period of 'runaway warming' that cannot be stopped, which could lead to the collapse of civilization due to climate change.
False_Supporting_4	Germany Leads the Way in Renewable Energy, with Nearly 65% of Electricity Generated from Renewables
False_Supporting_5	Climate Catastrophe: Entire Cities to be Submerged by Rising Seas Within Decades
False_Delaying_1	Extreme Weather: Natural variability, not human activity, is the main driver of extreme weather events
False_Delaying_2	The Climate Challenge Can Be Addressed Through Innovation and Technology Advancements in Fossil Fuels
False_Delaying_3	Carbon Dioxide is Not a Pollutant, but a Benefit to the Environment

False_Delaying_4	Catastrophic Consequences of Global Warming are Inevitable and Unavoidable
False_Delaying_5	Renewable Energy is Costly and Inefficient, and Should Not be Subsidized

Note: “True/False” refers to true or false statements; “Supporting/Delaying” refers to statements supporting or opposing climate science and action.

In this context, we would like to note that some of the more novel inoculation strategies proposed in our registered report are not specifically oriented towards increasing news veracity discernment per se. In particular, we are less confident that socio-affective inoculation strategies will increase overall truth discernment, as their effect might rather be related to a rejection of all claims against climate science and action (regardless of whether they are true or false). We are however committed to transparently exploring the benefits as well as the unwanted side-effects of the different proposed interventions (IJzerman et al., 2020) with regards on their impact on truth discernment, so we have added a detailed section on “interpretations given to different outcomes” to the revised manuscript (see RQ₄ of the design table).

RQ₄: Do the six inoculations increase participants' news veracity discernment^[165]?

H₄: Compared to participants in the passive disinformation control condition, participants who have received one of the inoculations will have higher news veracity discernment.

We identified the required sample of $N = 5964$ (i.e., sample without the pure control condition), a-priori, with G*Power, in order to have 95% power to detect an overall difference between any intervention condition and the passive disinformation control condition of $\delta = 0.20$ in a one-tailed t-test with $\alpha = .005$, for hypothesis H₂. We selected the smallest effect size of interest from the lower bound of the confidence interval of the meta-analytically identified effect size of fact-checking interventions on political topics^[166]; we reasoned that a new misinformation intervention would be practically interesting if and only if it has an effect that is larger than already available benchmark

We will analyze the performance in the truth discernment task with a multilevel model. We will calculate news veracity discernment as the sum of correct identification of true and false climate-related statements^[165]. We will specify two random effects: intercept for participant; and intercept for country (factor, alphabetically coded), to account for the variance associated with each country^[185]. We will specify condition (factor, seven levels, dummy coded with 0 = passive disinformation control condition as the reference contrast) as the only fixed effect besides the covariates.

Should any of the contrasts between the passive disinformation control condition and the inoculations be not significant, we will first visually inspect the raincloud distribution of news veracity discernment, to visually identify ceiling or flooring effects. Upon visual confirmation of a normal distribution, we will test whether the difference between the passive disinformation control condition and the inoculation of interest is smaller than our smallest effect size of interest ($\delta = 0.20$) with equivalence testing.

Should any of the equivalence testing be significant at $p < .005$, we will interpret the result as confirmatory evidence that the inoculation of interest does not offer better protection than traditional fact-checking.

Should any of the equivalence testing be significant at $.005 < p < .05$, we will interpret the result as suggestive evidence that the inoculation of interest does not offer better protection than traditional fact-checking. We will furthermore calculate the real news detection and the false news detection scores^[164], to investigate as to whether some inoculation might have influence only one of the two underlying factors of the general news veracity discernment score.

		interventions such as fact-checks.		
--	--	------------------------------------	--	--

4) The measurement and alteration of CRT/accuracy

Recent work (Roozenbeek et al., 2022) has called the role of CRT in misinformation susceptibility into question. Specifically, it appears that actively open-minded thinking (AOT) accounts for greater variance in the cognitive mechanisms associated with accuracy that make people more/less susceptible to misinformation. Therefore, I would urge you to reconsider your inoculation with regards to accuracy, at least including a measure of AOT to address these contemporary findings.

We thank the reviewer for pointing us towards the lively debate around an important individual difference measure that can account for misinformation susceptibility. We recognize that recent findings indicate that the AOT provides a more interesting moderator in the context of truth discernment capacity, as a measure of avoiding “myside bias” and overconfidence in one’s own conclusion (Baron et al., 2022). Indeed, if our only interest would have been the impact of inoculation strategies on truth discernment, we would have proposed this moderator instead. However, CRT is a measure that allows assessing people’s overall tendency to rely on socio-affective or cognitive pathways of information processing. In the context of the theoretical framework put forward in our manuscript, which differentiates between cognitive and socio-affective inoculation pathways, CRT appeared to be the more interesting moderator to assess the impact of different inoculation strategies on climate-related truth discernment, attitudes, beliefs, and behavior. Concretely, we expect that CRT may capture differences in the effectiveness of the proposed psychological inoculations depending on the match between the pathway of entry and participants’ cognitive predispositions to think intuitively/deliberatively as assessed by the CRT (see $H_{\text{secondary 1}}$ in the Design Table). Given that CRT and AOT are only weakly correlated and are considered separate constructs (Baron, 2019; see also the mediation models in Bronstein et al., 2019), we would prefer to keep the CRT as key moderator of interest as AOT is not related to the two pathways. If the editor and reviewers disagree, we could envisage using the recently developed 4-Component Thinking Style (Newton et al., 2022), which contains an AOT measure and three additional subscales capturing both intuitive/reflective thinking.

Reviewer #3:

Remarks to the Author:

Thank you for the invitation to review this Stage 1 Registered Report. The manuscript firstly presents a theoretical framework describing various factors influencing belief formation and updating in the context of climate change. Six "inoculation strategies" that may protect individuals from climate change disinformation are then created, based on this framework. Finally, the authors propose a study to test the effectiveness of these strategies. The basic design of the study is as follows: (1) participants will be randomly assigned to receive one of the six inoculation strategies or a passive control group; (2) each participant will be presented with 20 real examples of climate change disinformation; (2) after each disinformation message is presented, participants will self-report their personal affect towards climate change; (3) after all disinformation messages are presented, participants will perform a pro-environmental behavioural task.

Firstly, I should highlight that although I am a psychologist with expertise in methodology, I do not have expertise in the general domain of climate change messaging. Below I have responded to the journal's prompts for reviewers.

The significance of the research question(s) and relevance for a broad, multidisciplinary audience

The study addresses an important topic with broad, multidisciplinary appeal.

The extent to which the proposed study can satisfactorily answer the research question(s)

I think this is a well designed study that can provide a useful answer to the research questions. The use of simple interventions and real disinformation will enhance the real-world applicability of the results.

Thank you very much for the positive appraisal of our work!

The logic, rationale, and plausibility of the proposed hypotheses

Everything seems fine here. Everything is clearly presented.

The soundness and feasibility of the methodology and analysis pipeline (including statistical power analysis)

- Generally the study appears to be well designed. I have a few comments/questions:

- 1. I'm quite concerned about demand characteristics - imagining myself in a participant's shoes, it seems pretty obvious what the researchers are trying to do and how they want me to respond. I'm not sure how that could be best addressed. Perhaps a more natural task could be created that doesn't make the intervention and topic of interest so obvious, for example, embedding the climate-change related information in a fake Twitter timeline consisting of other non-climate information and asking users to scroll through.**

We thank the reviewer for raising this concern about demand effects, which have indeed not been addressed sufficiently in the previous submission. As already outlined in our response to the editor, we have taken several measures to address demand effects which we will reiterate below for completeness. The embedding of the climate disinformation statements in a fake Twitter timeline suggested by the reviewer would indeed increase the ecological validity of our experimental design and potentially reduce demand characteristics. However, it is incompatible with one of the main aims of the investigation. We are not only interested in the overall effects of the proposed inoculations, but we furthermore aim to analyze just how each inoculation can affect (mis)information accumulation (H_{1B}). In other words, we want to assess whether inoculations affect the processing of only one climate disinformation statement, or whether their protective effects extend to multiple instances of disinformation. To do this, we will need to assess participant responses systematically after each instance of disinformation, which unfortunately would not be feasible when using a fake Twitter timeline. Nevertheless, we think that the four points discussed below and integrated in the revised version of the manuscript comprehensively address the issue of demand effects:

1. The new truth discernment task that was added in this revised version is a performance-based task, therefore less likely to be influenced by demand effects. To optimally design the task against the influence of demand effects, we experimentally crossed arguments in favor and against climate action with true and false statements, which avoids that participants interpret the task as asking for “blind support” for climate science and mitigation action. Similarly, the behavioral task already present in the first version of the manuscript is a performance-based task in which participants need to invest their time and effort for the benefit of the environment, it is as such unlikely to be overly influenced by demand effects.
2. In the context of the different part of our experimental sequence, a potential impact of demand effects would be most likely to be observed in participant responses during the repeated affect ratings recorded during the disinformation provision task. We are hesitant to introduce major changes to this experimental task. The task in its current form most closely mimics the real-life information environment in which climate disinformation is most frequently encountered, while leaving sufficient experimental control for a continuous measurement of the impact of repeated climate disinformation. We would moreover like to point out that the results obtained during the stimulus development and selection phase speak against a large impact of demand effects on these ratings: As indicated in Figure 1 and Table 1 in this response to the reviewers, the affective evaluations of the different climate disinformation statements were heterogeneously distributed across the measurement space, and were moreover congruent with individual difference measures of theoretical relevance. For instance, participants with higher trust in science felt less negative about each climate disinformation statement. If the participants had been operating under a demand effect (e.g., “I read negative information, so I am supposed to report that I feel bad”), we would have expected more homogeneous and more negative evaluations. We would moreover expect no influence of individual differences. These observations are also relevant light of recent studies that have found no evidence of demand effects in political topics (Mummolo & Peterson, 2019), or that found only modest demand effects even when the experimental objectives were explicitly cued (De Quidt et al., 2018).

3. We would moreover like to point out that the experimental design of the study is based on a between-participants design. Individual participants are therefore not aware of the complete design and the experimental hypotheses, which are based on the comparison of different experimental conditions (Mummolo & Peterson, 2019).
4. As a final safeguard against demand effects, we will explore potential demand effects at the end of our experiment. We will ask participants what they think the study aim was with an open-ended question at the end of the study. Akin to Coles and colleagues, 2022, two coders will then rate participants' belief in the experimental objective (*To what degree do you think the participant believed we were testing interventions to fight climate disinformation? 0 = They seemed very convinced we were not testing interventions to fight climate disinformation; 1 = They seemed somewhat convinced we were not testing interventions to fight climate disinformation; 2 = They seemed unsure if we were not testing interventions to fight climate disinformation; 3 = They seemed somewhat convinced we were testing interventions to fight climate disinformation; 4 = They seemed very convinced we were testing interventions to fight climate disinformation*) in order to quantify the prevalence of potential demand effects. In the revised manuscript, we have added a “demand effects check” factor in the multilevel model (main effect, two-way interaction with trial (for the passive disinformation control condition), and three-way interaction with trial and condition (for the intervention conditions)) with affect towards climate action as a DV.

Below are Figure 1 and Table 1 taken from the results of the pretest of the climate disinformation statements, mentioned in point 3 to the editor above, with the nationally representative sample of $N=50$ British participants (for a description of the pretest, see the “disinformation provision” subsection of the manuscript, p.16, and the description in the OSF repository). The figure and table showcase the heterogenous spread of the evaluations and their relationship with individual differences of theoretical interest (all data is available at: https://osf.io/m58zx/?view_only=95fd430f4b7e4ee99c9c8b472e31d6b3):

Figure 1: Boxplot for raw affective reactions towards climate action generated by each climate disinformation statement, reordered in descending order of affect towards climate mitigation action for visual purposes.

Discourses of delay of climate action are represented in red; disinformation about climate science is represented in grey. The x axis represents each climate disinformation statement, reorder in terms of descending affect towards climate mitigation action. The y axis represents the ratings of how people felt towards climate mitigation action after the climate disinformation statement, with values $-5 < x < 0$ representing feeling negative about climate mitigation actions, and values $0 < x < 5$ representing feeling positive about climate mitigation actions.

Table 1 – Correlations between affect towards climate science/action and the individual differences of interest collected in the pretest.

	Social conservatism	Economic conservatism	CRT	Trust in science	Climate change importance	climate change worry	climate change impacts on self (n=493)	climate change impacts on others (n=496)
Affect: climate science	-.29*** [-.37, -.20]	-.27*** [-.35, .19]	.08*** [.05, .11]	.27*** [.24, .30]	.31*** [.29, .34]	.27*** [.25, .30]	.26*** [.23, .28]	.28*** [.26, .31]
Affect: climate action	-.27*** [-.30, -.35]	-.27*** [-.35, -.18]	.07*** [.04, .10]	.27*** [.24, .29]	.32*** [.30, .35]	.29*** [.27, .32]	.22*** [.20, .25]	.29*** [.26, .31]

Note: * $p < .05$. ** $p < .01$. *** $p < .001$. 95% Confidence intervals in brackets. Social and economic conservatism in ascending values, such as higher values represent higher conservatism.

We have now included a section about demand effects in the measurement section of the revised manuscript (p. 24-25):

Demand effects check: We will probe participants’ understanding of the aim of the experiment by asking them “[Could you please describe what you think the aim of the experiment was?]”. Two coders will then rate participants’ belief in the experimental objective with a multiple-choice question ([To what degree do you think the participant believed we were testing interventions to fight climate disinformation?] 0 = [They seemed very convinced we were not testing interventions to fight climate disinformation]; 1 = [They seemed somewhat convinced we were not testing interventions to fight climate disinformation]; 2 = [They seemed unsure if we were not testing interventions to fight climate disinformation]; 3 = [They seemed somewhat convinced we were testing interventions to fight climate disinformation]; 4 = [They

seemed very convinced we were testing interventions to fight climate disinformation]; ratings will be averaged, and differences in scoring will be discussed and resolved).

We have furthermore added a paragraph describing the analyses to be conducted with the variable in the “Control analyses” subsection of the revised manuscript (p. 33):

In order to account for potential demand effects, we will introduce the “demand effects check” measure as a control variable for H_{1A-B} . Should our participants be influenced by demand effects, we would expect that the variable will moderate the effectiveness of the psychological inoculations, such that participants who receive a psychological inoculation and have understood the experimental aim will report more positive affect towards climate action overall. We will add the “demand effects check” score (continuous, range from 0 to 4) as a fixed predictor as a main effect and a two-way interaction with condition. We will conduct a second multilevel model within the passive disinformation control condition, in order to assess whether demand effects might influence the disinformation provision. We will specify three random effects: intercept for participant; intercept for the internal numbering of the climate disinformation statements, to account for the variance associated with each disinformation statement^[185]; intercept for country (factor, alphabetically coded), to account for the variance associated with each country^[185]. We will specify as fixed effects: “demand effects check” (continuous, range from 0 to 4), trial (continuous variable, from 1 to 20), and the two-way interactions of trial with demand effects check.

We however see the benefit of presenting the climate disinformation statements in a manner that closely resembles information received on Twitter. We therefore embedded the different statements in a tweet-like format (see Figure 2) without identifying cues (no image, username, or name) or engagement metrics.

Figure 2: Example of climate disinformation statement embedded in a tweet-like format.

We explicitly reference this new delivery mode in the “procedure” subsection of the revised manuscript (p.13):

All interventions will be presented sequentially in four screens, with a 5-20s time lock (depending on the content length of each screen) that will not allow participants to manually proceed to the next screen until the time has elapsed. A manipulation check measuring participants’ motivation to resist persuasion^[171] will follow. Afterwards, participants will receive twenty real climate disinformation statements **in the form of anonymous tweets**, in randomized order **with a 2s time lock**, and report their affect towards actions to mitigate climate change after each disinformation statement.

And in the “disinformation provision” subsection of the revised manuscript (p.17):

During the experiment, participants will be presented with all twenty selected climate disinformation statements in randomized order. **Each statement will be presented as an anonymous tweet, with the default user image, no identifying information, and no engagement metrics**. After each disinformation, participants will rate their affect towards climate actions on a visual analogue scale.

- 5. Related to the above, the current design uses a passive control condition (participants skip straight to the disinformation messages). I wonder if an active control condition - which could have the benefit of reducing participant reactivity effects (e.g., Hawthorne effects) - is feasible in this context. For example, perhaps control participants could read a general (non-climate related) news message? It**

could be that the authors prefer a passive control because it more closely resembles the practical circumstances in which inoculation strategies will be deployed. Either way, perhaps the rationale for the control group design could be included in the text?

We thank the reviewer for the clarification request and for the apt description of our implicit aim. We now explicitly justify the choice of the passive disinformation control condition over an active control condition in the revised manuscript (p. 12):

The study will follow a mixed design, with a between-participants randomized assignment to one of **eight** different conditions: **pure control (no inoculation, no disinformation)**, **passive disinformation control (disinformation without inoculation)**, scientific consensus inoculation, trust in scientists inoculation, transparent communication inoculation, moralization of climate action inoculation, accuracy inoculation, and positive emotion inoculation. **We chose a passive disinformation control condition over an active or positive control in order to better mimic real-life information environments, where climate disinformation is most frequently encountered passively and in multiple occurrences.** Participants and experimenters will be blind to the name and aim of the condition participants are randomized into (double blind). The experiment will contain twenty within-participants repeated measures of affect towards climate mitigation actions, assessed after each of the twenty climate disinformation statements.

We have now moreover added a *pure control* condition to the experimental design, in which participants are neither exposed to inoculations nor to climate disinformation. Comparison of this condition to the *passive disinformation control* condition (in which participants are exposed to disinformation, but no inoculation), will allow us to quantify the effects of climate disinformation statements on climate-relevant attitudes and behaviors.

- 6. It's not clear to me why each participant views 20 disinformation messages and responds to the affect question after each. It would be good to hear more about the rationale for that. An alternative would be to give each participant a single randomly selected disinformation message.**

We are hesitant to introduce major changes to the “disinformation provision” experimental task. The task in its current form most closely mimics the real-life information environment in which climate disinformation is most frequently encountered (not as single but as multiple consecutive occurrences), while leaving sufficient experimental control for a continuous measurement of the impact of repeated climate disinformation. The motivation underlying this choice is that we wanted to not only test the main effects of the proposed psychological inoculations, but also to test whether their protective effects can influence the way participants update their beliefs when they accumulate multiple pieces of disinformation. We deemed this important and interesting for three reasons: first, there is heightened societal concern about growing amounts of climate misinformation in our information environments (EU DisinfoLab, 2023; UN, 2022; Vaughan, 2022). Second, discourse polarization in social and legacy media around the climate change topic has been on the rise in recent years (Chinn et al., 2020; Falkenberg et al., 2022; Grustafson et al., 2019; Williams et al., 2015), making a better understanding of the effects of multiple climate disinformation and their corrections a priority (van Bavel et al., 2021). To illustrate this concern with an example, habitual viewers of Fox News might view more than a single climate disinformation (or, more likely, be “bombarded” by them) before corrections appear in their information environment. People belonging to such information environments – and other populations disproportionately at risk of susceptibility to disinformation. Third, psychological inoculations can have a cumulative protective effect which is currently under-researched. In a previous study (Spampatti et al., under review, Study 1), we found that the effects of the trust inoculation increased as participants received more negative persuasive attacks. We unfortunately lack the word space in the introduction to give justice to all the relevant points, so we now explicitly make a general statement relevance in the introduction of the revised manuscript, which can be expanded in the discussion should space allow (p. 10):

Concretely, participants will be presented with twenty real climate disinformation statements that were selected based on an initial validation study ($N=504$, available at: https://osf.io/m58zx/?view_only=95fd430f4b7e4ee99c9c8b472e31d6b3). After each disinformation, they will rate their current affect towards climate actions (we will measure affect towards, rather than political support for, climate mitigation actions because affective reactions predate and motivate policy appraisals and climate-friendly behavior^[50-51; 145]), to measure if psychological inoculations can not only protect against single, but also against multiple consecutive instances of climate disinformation (mirroring the preponderance of climate disinformation in certain epistemic communities^[29]).

- 7. One of the exclusion criteria is “iv) respond with the same response for more than 90% of the questions”. What is the justification for this criterion? It seems reasonable to me that a participant might respond consistently if they are not influenced by the disinformation messages.**

We thank the reviewer for raising this important methodological point: Indeed, using this criterion, participants who are not influenced by the disinformation messages might be systematically removed. We have now removed this exclusion criterion from the “Data inclusion” and “Exclusion criteria” sections of the manuscript.

Whether the clarity and degree of methodological detail would be sufficient to replicate exactly the proposed experimental procedures and analysis pipeline

Yes, it appears so.

Whether the authors provide a sufficiently clear and detailed description of the methods to prevent undisclosed flexibility in the experimental procedures or analysis pipeline

Yes it appears so.

Whether the authors have considered sufficient outcome-neutral conditions (e.g. absence of floor or ceiling effects; positive controls) for ensuring that the results obtained are able to test the stated hypotheses

Yes, this seems satisfactory.

Reviewer #4:

Remarks to the Author:

I very much enjoyed reviewing this Registered Report. I think the topic of climate disinformation is obviously of great societal importance and this is by far the most

comprehensive assessment of the potential of psychological inoculation against climate disinformation I've seen. I appreciated the novel theoretical framework and the high level of technical detail provided in the RR, which allows for a proper assessment of its overall value.

Thank you very much for these encouraging comments!

I am definitely supportive of this work, I think it has great value for the literature across multiple disciplines (climate, behaviour, misinformation etc) but I do see some areas of potential concern that I think could benefit from further reflection as well as some more minor comments:

(1) The first is a potential mismatch between the level at which the intervention operates (inoculation) and the dependent variable (climate action) and whether this constitutes a fair test of the intervention. For example, the literature is pretty silent on the extent to which inoculation can actually move behavior. Personally, I am not sure about the extent to which inoculation is going to protect behavioral action. Inoculation interventions are typically geared at (a) relevant beliefs and attitudes and (b) helping people spot the misleading argument or manipulation attempt. At present, the authors are not measuring either. I appreciate the manipulation check about motivation to resist persuasion (to see if the inoculation worked as intended) but I did not see any measures of whether inoculation will help people resist or identify misinformation specifically, which is the level at which inoculation usually operates.

We thank the reviewer for raising these theoretical and methodological points. We have now added a truth discernment task, allowing to test whether the inoculations help participants to spot misleading arguments (described in more detail in response 3 to Reviewer 4), as well as a measure of climate change perception scale assessing the perceived reality, causes, and consequences of climate change, allowing to test whether relevant beliefs and attitudes are protected. The manuscript has been revised to introduce the climate change perception scale as follows (p. 20-21):

Climate change beliefs: We will assess participants' beliefs about climate change with the climate change perception scale^[163], a validated scale that encompasses different dimensions of the appraisal of climate science and the consequences of climate change. While the published scale is composed of five different subscales and related factors, the authors note that the climate change perception scale allows for the selection of subscales

of interest ^[163]. We will therefore focus on the three subscales measuring participants' belief in the reality of climate change, the causes of climate change, and the consequences of climate change. Climate change beliefs will be collected with nine items (Reality subscale: [I believe that climate change is real]; [Climate change is NOT occurring] (reverse scored); [I do NOT believe that climate change is real] (reverse scored). Causes subscale [Human activities are a major cause of climate change]; [Climate change is mostly caused by human activity]; [The main causes of climate change are human activities]; Consequences subscale [Overall, climate change will bring more negative than positive consequences to the world].; [Climate change will bring about serious negative consequences]; [The consequences of climate change will be very serious]; 1 = [strongly disagree], 7 = [strongly agree]. Items per each subscale will be mean-scored if Cronbach's $\alpha > .70$, otherwise we will use only the first item as representative of the subscale^[163]).

Along with the accompanying statistical analyses:

RQ₃: Do the six inoculations protect participants' belief in A) the reality of climate change, B) the causes of climate change, and C) the consequences of climate change^[163], after twenty real climate disinformation statements?	H₃: Compared to participants in the passive disinformation control condition, participants who have received one of the inoculations will report higher belief in the reality of climate change (H_{3A}), the causes of climate change (H_{3B}), and the consequences of climate change (H_{3C}) after receiving all the twenty climate disinformation statements.	We identified the required sample of $N = 5964$ (i.e., sample without the pure control condition), a-priori, with G*Power, in order to have 95% power to detect an overall difference between any intervention condition and the passive disinformation control condition of $\eta^2 = 0.20$ in a one-tailed t-test with $\alpha = .005$, for hypothesis H₂. We selected the smallest effect size of interest from the lower bound of the confidence interval of the meta-analytically identified effect size of fact-checking interventions on political topics^[165]; we reasoned that	We will analyze the climate change beliefs ^[162] with a multilevel model per each subscale. We will specify two random effects: intercept for participant, and intercept for country (factor, alphabetically coded), to account for the variance associated with each country^[185]. We will specify condition (factor, seven levels, dummy coded with 0 = passive disinformation control condition as the reference contrast) as the only fixed effect besides the covariates.	Should any of the contrasts between the passive disinformation control condition and the inoculations be not significant, we will first visually inspect the raincloud distribution of the responses of each climate change perception subscale^[162], to visually identify ceiling or flooring effects. Upon visual confirmation of a normal distribution, we will test whether the difference between the passive disinformation control condition and the inoculation of interest is smaller than our smallest
---	--	---	---	--

		a new misinformation intervention would be practically interesting if and only if it has an effect that is larger than already available benchmark interventions such as fact-checks.	effect size of interest ($\delta = 0.20$) with equivalence testing. Should any of the equivalence testing be significant at $.005 < p < .05$, we will interpret the result as suggestive evidence that the inoculation of interest does not offer better protection than traditional fact-checking. Should any of the equivalence testing be significant at $p < .005$, we will interpret the result as confirmatory evidence that the inoculation of interest does not offer better protection than traditional fact-checking.
--	--	--	--

We should moreover have stressed in more detail in the manuscript that one of the proposed novel contributions of this study is indeed extending the evidence in favor of psychological inoculations to actual behavior. Changes in self-reported behavior following inoculations have been measured in the past (e.g., post-inoculation talk, Compton & Pfau, 2009; Ivanov et al., 2012) and, more recently in psychological inoculations against disinformation as well (e.g., post-inoculation talk: Maertens, 2023; willingness to share: Basol et al., 2022; Roozenbeek et al., 2022; composing a tweet: Tay et al., 2021). Here we will extend this to consequential pro-environmental behavior. We now make this point clearer in the revised manuscript (p. 10):

We will investigate the effectiveness of these six **broad-spectrum** psychological inoculation strategies to protect against climate disinformation in a multi-country, multi-intervention study, against a sequence of twenty real climate disinformation spread by members of the climate change countermovement on the social media Twitter. We will assess the protective effect of the inoculations on **participants' climate change beliefs**^[163],

appraisal of climate mitigation action, and truth discernment capacity, i.e., their capacity to correctly distinguish between true and false information^[165]. We will moreover investigate if the protective effects of psychological inoculations extend to actual pro-environmental behavior^[164].

(2) Relatedly, there should be a match between the content of the inoculation message and the disinformation attack, otherwise we're not testing inoculation as a main effect but rather the extent to which it can function as an "umbrella" or "broad-spectrum" vaccine insofar it protects against a range of disinformation not necessarily mentioned in the inoculation itself. For example, if you inoculate someone against X, and then attack them with Y, can we really conclude anything about inoculation if it didn't work? I.e. if people don't have the right mental defences you cannot expect them to generate immunity, especially at the behavioral level. Maybe the authors have already thought about this, but one helpful revision could be to link the Twitter disinformation messages to the relevant inoculation condition in the Table so that it is clear which disinformation corresponds to which hypothesis/condition. For example, I saw some disinformation about the scientific consensus, which would nicely match with the consensus inoculation condition etc. I think this should be clarified better for the disinformation messages. Having said this, it's not a problem if you want to test how far the inoculation stretches across different issues but this should then be explicitly recognized as a secondary hypothesis.

We thank the reviewer for this point, which allows us to clarify the contributions of our proposed study. Our intention was indeed to devise the umbrella, broad-spectrum inoculation they described. We now mention this in the introduction of the revised manuscript:

In summary, here we integrate previous analyses into a comprehensive framework of the communicational and psychological factors influencing (anti)science belief formation and updating. Based on this integrated, theory-driven perspective, we created a set of broad-spectrum psychological inoculations to protect against climate disinformation that will act on each of the identified entry points and pathways:

- a scientific consensus inoculation explaining that among climate scientists there is virtually no disagreement that humans are causing climate change;
- a trust inoculation making salient the trustworthiness of IPCC scientists in terms of climate change science and mitigation actions;
- a transparent communication inoculation transparently addressing the pros and cons of climate mitigation action;

- a moralization inoculation creating a stronger link between climate mitigation actions and the diversity of moral convictions;
- an accuracy inoculation reorienting participants towards judging incoming information by their factual accuracy;
- a positive emotions inoculation eliciting positive emotions towards climate mitigation actions.

We will investigate the effectiveness of these six **broad-spectrum** psychological inoculation strategies to protect against climate disinformation in a multi-country, multi-intervention study, against a sequence of twenty real climate disinformation spread by members of the climate change countermovement on the social media Twitter.

At the statistical level, we do account for the possibility of a better match of some of the inoculations with some specific climate disinformation statements. By adding a random factor for each climate disinformation statement, we statistically take into account the variance associated with the presentation of each unique climate disinformation statement. This will allow us to generalize the results of the multilevel models beyond the tested items (Judd et al., 2012). With this statistical consideration in place, the unique contribution of climate disinformation statements matching the content of some psychological inoculations can be attenuated. We therefore are confident the results of these analyses will be informative about the broad-spectrum protection of each of the six inoculations.

(3) I wondered about the need for a pure control condition to establish that the passive fact-check and inoculation work as intended? Moreover, inoculation can only be effective if the disinformation is persuasive in the first place otherwise there is nothing to inoculate against, so is it not important to illustrate that disinformation had a negative effect on the DVs in a pure control condition (no inoculation, no fact-check)? If misinformation does nothing, then there's nothing to inoculate against. The authors say they pilot tested the disinformation tweets, but in order to not waste a huge amount of resources, they might at least want to check whether the disinformation is indeed negatively influencing the key DVs before proceeding.

We thank the reviewer for these suggestions. As suggested, we have added a pure control condition. The manuscript now reads as follows (p. 12):

The study will follow a mixed design, with a between-participants randomized assignment to one of eight different conditions: pure control (no inoculation, no disinformation), passive disinformation control (disinformation without inoculation), scientific consensus inoculation, trust in scientists inoculation, transparent communication inoculation, moralization of climate action inoculation, accuracy inoculation, and positive emotion inoculation. We chose a passive disinformation control condition over an active or positive control in order to better mimic real-life information environments, where climate disinformation is most frequently encountered passively and in multiple occurrences. Participants and experimenters will be blind to the name and aim of the condition participants are randomized into (double blind). The experiment will contain twenty within-participants repeated measures of affect towards climate mitigation actions, assessed after each of the twenty climate disinformation statements.

We have added four explicit quality control hypotheses when it comes to the comparison between the pure and passive control conditions. We have now implemented these changes in the reviewed version of the manuscript, in the “Analysis plan” section and in the Design Table under RQ_{Control 1}. The added “Control analyses” subsection reads:

RQ_{Control 1}: does consecutively presenting the twenty real climate disinformation statements decrease participants A) affect towards climate mitigation action; B) beliefs in climate change; C) participants' performance in the modified version of the Work for Environmental Protection Task (WEPT^[162]); and D) truth discernment?	H_{Control 1}: Compared to participants in the pure control condition, participants in the passive disinformation control condition will report more negative affect towards climate mitigation action (H_{Control 1A}), believing less in anthropogenic climate change (H_{Control 1B}), complete less pages of the WEPT task (H_{Control 1C}), and have a worse truth discernment (H_{Control 1D}). We furthermore expect participants in the passive disinformation control condition to report more negative feelings about climate mitigation actions at the end of the disinformation provision,	As for our main hypotheses, we identified the required sample of $n = 852$ for the pure control condition, a-priori, with G*Power, in order to have 95% power to detect an overall difference with the passive disinformation control condition of $\eta^2 = 0.20$ in a one-tailed t-test with $\alpha = .005$, for hypothesis H_{Control 1}. For hypothesis H_{Control 1A-bis}, sensitivity analysis shows we will achieve 95% power to detect an effect as small as $\delta_z = 0.14$ ($\alpha = .005$) and $\delta_z = 0.12$ ($\alpha = .01$).	Hypothesis H_{Control 1A-D}: We will compare affect toward climate action, belief in climate change, WEPT performance, and truth discernment at the end of the intervention (i.e., after the twentieth disinformation statement) of the participants in the passive disinformation control condition with the participants in the pure control condition, separately, with a one-tailed independent-sample t-test with α corrected to .005. For hypothesis H_{Control 1A-bis}, we will conduct an additional one-tailed, paired-sample t-test within the passive disinformation	Hypothesis H_{Control 1A-D}: Should any of the contrasts be not significant, we will first visually inspect the dependent variables, to visually identify ceiling or flooring effects for the intervention conditions. After confirming the lack of ceiling/flooring effects, we will test whether the difference between the passive disinformation control condition and the pure control condition is smaller than our smallest effect size of interest with equivalence testing. Should any of the equivalence testing be significant at $.005 < p < .05$,
--	--	---	---	--

	compared to their baseline affect ($H_{\text{Control 1A-bis}}$).		control condition, with affect towards climate mitigation action as the dependent variable.	we will interpret the result as suggestive evidence that, if the twenty real climate disinformation statements have a detrimental effect on any of our dependent variables of interest, it is lower than $\alpha = 0.20$.
--	--	--	--	---

We have moreover added a baseline measure of affect towards climate mitigation action, to be collected before the psychological inoculations are delivered. This measure will enable us to complement the findings from pure-passive controls comparison with a pre-post within-participants measures of decline of affect towards climate mitigation actions, which will augment the precision of our estimates and provide more evidence for a casual effects of the climate disinformation statements (Clifford et al., 2021; Roher et al., in press; Roozenbeek et al., 2021; Swire-Thompson et al., 2020). The “Procedure” subsection of the revised manuscript now reads (p. 12):

Procedure: Participants will access the survey through an anonymous link made available by the panel provider, and will provide their explicit consent to the study. After consenting, participants will report their demographics (gender, age, and political orientation: single-item, 10-point scale: 1=[Extreme liberalism/left] to 10=[Extreme conservatism/right]), complete a baseline measure of affect towards climate mitigation action, and complete the Cognitive Reflection Task, Version 2 (CRT-2^[169]), in random order.

(4) The forewarning seems to be taken literally from van der Linden et al. (2017), which the authors may want to mention insofar it helps provide prior validation of this wording.

The reviewer is correct, we have added a mention to the study in the “The six inoculations” subsection of the revised manuscript. The manuscript now reads as follows (p. 13):

The six inoculations: The inoculations have been adapted to the same presentation format, as textual stimuli divided in two paragraphs. All inoculations will contain an opening paragraph referring to the IPCC assessment of anthropogenic climate change: “[In their latest assessment, the Intergovernmental Panel on Climate Change (IPCC) has declared that anthropogenic climate

change is happening, and urgent action is needed to prevent irreversible negative effects on the planet and society.]” followed by a preemptive warning of incoming threatening information^[38]: “[However, some politically-motivated groups use misleading tactics to try to convince the public that there is a lot of disagreement among scientists and that climate action is useless or harmful to society]”.

(5) The power analysis seems correct to me, but, 500 per country, with 7 conditions, breaks down to about 70 people per condition, which is not that high. Especially if another pure control baseline is added. Is there any way to oversample?

The reviewer raises a correct point of concern regarding the cell size per country per condition. However, as elaborated in the response to the editor and to comment 7 below, this is a concern only if an interaction between condition and country is analyzed as a fixed factor in the proposed multilevel models. Instead, we are collecting these data precisely to avoid making this comparison, but to generalize our findings beyond the country of implementation. Therefore, our aim is not to test this underpowered interaction. On a side note, it may currently be unachievable to preregister or justify any directional hypotheses for this interaction, as the misinformation field currently does not provide the a-priori knowledge to predict that x psychological inoculation will work better in country x but not in country y.

(6) It is not mentioned whether the panel is representative of the population in each country or whether quotas will be used or just a standard panel.

We will be collecting data from each country with gender and age quota. We opted for quota, rather than a representative sample, as the latter might not improve causal estimates of treatment effects (Coppock et al., 2018). The information has now been added in the “sampling plan” section of the revised manuscript (p. 25):

We will collect the sample with quota for gender and age from the panel provider Market Science Institute. The sample will comprise of participants from twelve countries, $n = 568$ participants per country, for a total of $N = 6816$ participants.

(7) For a multi-level model, 12 countries is not a lot, some argue that if the participant-level n is large compared to the second-level (n) it can bias estimates, though this mostly seems to concern the country-level comparisons (see Bryan & Jenkins, 2016). Perhaps worth considering.

Our choice of collecting data in 12 countries was based on the motivation to go beyond the classic WEIRD populations, thus increasing the generalizability of our findings beyond one single Western country. At the same time, we needed to be considerate of the resources needed to

obtain the necessary data. For instance, for a hypothetical experiment conducted in two countries and modeling a “country” term as a fixed – rather than random – predictor, twice the proposed sample would be needed to model the simple slopes of the “condition” factor within each country. While this would undoubtedly also be interesting, we cannot dedicate the necessary financial resources (in the order of tens of thousands of EUR/USD) to it, and thus opted to optimize our design according to the minimum number of countries required to be able to model the country variable as a random effect with acceptable accuracy. Based on these practical considerations, we are thus interested in generalizing our findings beyond the specific countries of implementation, rather than engaging in underpowered cross-country comparisons. We make this point more explicit in the revised manuscript (p. 25-26):

Countries: We will recruit participants based in the USA, Canada, UK, Ireland, Australia, New Zealand, Singapore, Philippines, India, Pakistan, Nigeria, South Africa, and Ghana (see Fig. 1) to generalize our findings on the effectiveness of the six psychological inoculations across the globe and in non-WEIRD contexts. We settled on twelve countries to provide the minimum number of countries to provide a reasonably accurate statistical estimation for country-level variation in our dependent variables as a random effect in multilevel models^[185] rather than conducting cross-country comparisons. The twelve countries were furthermore chosen pragmatically for English being the main or one of the official languages, to maintain the climate disinformation statements in their original language and therefore maintain the highest ecological validity.

(8) The authors may find recent work about transparency, inoculation, and trust of interest to motivate the transparency condition (see Kerr et al., 2023).

We thank the reviewer for recommending the addition of the interesting studies by Kerr and colleagues (2023), which were published after our initial submission. We now include and cite the reference (nr. 64) whenever relevant.

(9) I wonder, if, in the context of climate change, it is useful to include ideology as a factor as well to establish whether politics reduces the effect of the inoculation condition in countries with known polarization on the issue vs others that are perhaps less polarized on climate change?

We thank the reviewer for this suggestion. We had initially decided not to include the political ideology measure in the preregistered analyses because we considered any hypotheses about the role of this variable not to be appropriate for the Registered Report format. RR guidelines for hypotheses require strong evidence for the claim to be investigated: not only is the influence of political ideology on climate change highly heterogeneous across countries (Hornsey et al., 2018), but in the case of a potential effect of political ideology on the inoculations, the only

paper having investigated a moderating effect reported a null result (van der Linden et al., 2017). We therefore felt measuring a potential moderation effect unjustified. However, we decided to add political ideology to the measured variables as a covariate, and to assess its potential effects in a series of exploratory analyses that will be conducted after the preregistered ones.

Decision Letter, first revision:

3rd April 2023

Dear Mr Spampatti,

Thank you once again for your manuscript, entitled "Psychological inoculation strategies to fight climate disinformation around the world," and for your patience during the peer review process.

Your manuscript has now been evaluated by 4 reviewers, whose comments are included at the end of this letter. Although the reviewers find your protocol to be of interest, they also raise some important concerns. We are very interested in the possibility of proceeding further with your submission in Nature Human Behaviour, but would like to consider your response to these concerns in the form of a revised manuscript before we make a decision on in principle acceptance and Stage 2 submission.

Specifically, we ask that you revise your protocol to address all remaining concerns of Reviewer 4.

I have also attached a detailed editorial checklist to this decision letter. Please make sure that your Registered Report adheres to all requirements listed in the checklist. To guide the process, I highlighted the parts that need specific attention.

In sum, we invite you to revise your Stage 1 Registered Report taking into account reviewer and editor comments. Please highlight all changes in the manuscript text file.

* Include a "Response to reviewers" document detailing, point-by-point, how you addressed each referee comment. If no action was taken to address a point, you must provide a compelling argument. This response will be sent back to the reviewers along with the revised manuscript.

* Ensure that you use our template for Stage 1 Registered Reports to prepare your revised manuscript: https://www.nature.com/documents/NHB_Template_RR_Stage1.docx. Failure to ensure that your revised Stage 1 submission meets our requirements as specified in the template will result in

your submission being returned to you, which will delay its consideration.

* In your cover letter, please include the following information:

- An anticipated timeline for completing the study if your Stage 1 submission is accepted in principle.
- A statement confirming that you agree to share your raw data, any digital study materials, computer code (if relevant), and laboratory log for all eventually published results.
- A statement confirming that, following Stage 1 in principle acceptance, you agree to register your approved protocol on the Open Science Framework (<https://osf.io/>) or other recognised repository, either publicly or under private embargo, until submission of the Stage 2 manuscript.
- A statement confirming that if you later withdraw your paper, you agree to the Journal publishing a short summary of the pre-registered study under a section Withdrawn Registrations.

[REDACTED]

We hope to receive your revised manuscript within four to eight weeks. If you cannot send it within this time, please let us know. We will be happy to consider your revision so long as nothing similar has been accepted for publication at Nature Human Behaviour or published elsewhere.

Nature Human Behaviour is committed to improving transparency in authorship. As part of our efforts in this direction, we are now requesting that all authors identified as 'corresponding author' on published papers create and link their Open Researcher and Contributor Identifier (ORCID) with their account on the Manuscript Tracking System (MTS), prior to acceptance. ORCID helps the scientific community achieve unambiguous attribution of all scholarly contributions. You can create and link your ORCID from the home page of the MTS by clicking on 'Modify my Springer Nature account'. For more information please visit please visit www.springernature.com/orcid.

Sincerely,

Samantha Antusch

Samantha Antusch, PhD
Senior Editor
Nature Human Behaviour

Reviewer expertise:

Reviewer #1: open science and Registered Reports

Reviewer #2: (climate change) misinformation ; interventions

Reviewer #3: open science and Registered Reports

Reviewer #4: (climate change) misinformation ; interventions

Reviewers' Comments:

Reviewer #1:

Remarks to the Author:

Thank you for addressing the issues raised by each reviewer. My suggestion for adding an additional inoculation was not accepted by the authors for reasons that they justify appropriately. The framework that they are proposing does lend itself to this possibility in a future study, but excluding it from this one does not invalidate any of the proposed assessments, and so I see no justifiable reason to suggest any other revisions.

Furthermore, from my perspective, the comments made by other reviewers were appropriately addressed.

Reviewer #2:

Remarks to the Author:

Thanks for engaging with my comments so closely, I'm definitely satisfied that all of my concerns have been addressed. I would just like to add that I really like your incorporation of the AI-generated climate headlines, I believe that will be a huge contribution to the literature by itself!

Reviewer #3:

Remarks to the Author:

My comments have been adequately addressed

Reviewer #4:

Remarks to the Author:

I appreciate the thought the authors put into the revision, I particularly welcome the pure control group, the MIST-inspired truth discernment task and the use of equivalence testing.

I'm basically on board with the plan now but I remain somewhat skeptical about the main hypothesis being broad-spectrum protection rather than narrow spectrum.

I think the authors should still include a fairer test of inoculation in the form of a simple sub-analysis: e.g. for the scientific consensus inoculation, only evaluate it against the disinformation tweets that target the scientific consensus (narrow) and then separately against all other climate disinformation (spill-over or broad-spectrum) and so on and so on. That way, there's more nuance. For example, it is not entirely surprising to find that the broad spectrum hypothesis fails in some instances but it would be surprising if inoculation doesn't work against the disinformation it is specifically inoculating against. Separating out the main effect into narrow inoculation (matched on items) versus broad (non-matched) will be good, also for informing the wider literature. If, at this point, certain inoculation conditions do not have matching items in the discernment task, these could perhaps be added to provide a specific match for each condition (in terms of relevant disinformation).

Sorry to be a pain about this but I think it's important.

Author Rebuttal, first revision:

Reviewer #4:

Remarks to the Author:

I'm basically on board with the plan now but I remain somewhat skeptical about the main hypothesis being broad-spectrum protection rather than narrow spectrum.

I think the authors should still include a fairer test of inoculation in the form of a simple sub-analysis: e.g. for the scientific consensus inoculation, only evaluate it against the disinformation tweets that target the scientific consensus (narrow) and then separately against all other climate disinformation (spill-over or broad-spectrum) and so on and so on. That way, there's more nuance. For example, it is not entirely surprising to find that the broad spectrum hypothesis fails in some instances but it would be surprising if inoculation doesn't work against the disinformation it is specifically inoculating against. Separating out the main effect into narrow inoculation (matched on items) versus broad (non-matched) will be good, also for informing the wider literature. If, at this point, certain inoculation conditions do not have matching items in the discernment task, these could perhaps be added to provide a specific match for each condition (in terms of relevant disinformation).

We thank the reviewer for transparently sharing their positive assessment of the study and for reiterating their theoretical point of interest. Given the extensive pre-testing that went into the stimulus selection, we are hesitant to change the stimulus set at this stage. We moreover think that manually selecting additional experimental stimuli to match all inoculations to at least one climate disinformation statement is difficult (for instance, it seems almost impossible to find climate-specific disinformation that would be specifically matched to the accuracy inoculation), and, more importantly, would damage the out-of-sample generalizability we can achieve from the data-driven selection of the climate disinformation statements (see Judd et al., 2012). These points notwithstanding, we do agree with the reviewer that presenting the fairest possible test for the psychological inoculations is important, and we are happy to add this extra layer of detail in the supplementary analyses. We thus propose to compare the protective effects of psychological inoculations depending on their thematic match with specific climate disinformation statements in the previously selected and validated stimulus set. To this end, we coded whether the climate disinformation statements from our set of stimuli are thematic matches with one of the different psychological inoculations. We were able to identify thematically related climate disinformation statements for four out of the six inoculation strategies (the coding can be found in Table SM-1 in the Supplementary Information). For the analysis, we will set up four multilevel models (one per psychological inoculation thematically related with at least one climate disinformation statement) to test whether the inoculations are more effective in protecting affect towards climate mitigation actions against thematically matching climate disinformation statements compared to non-matching statements. Analyzing the interaction term “thematic match” with condition will allow us to disentangle the “broad” main effects of the interventions from the potential “narrow” effects suggested by the reviewer. We now introduce this additional analysis in the Analysis Plan Section (pp. 33-35):

Finally, although the six psychological inoculations presented here are conceptualized as broad-spectrum inoculations^[184], it is possible that the content of specific climate disinformation statements matches the thematical content of specific psychological inoculations more closely than others, and that this thematic match increases the protective effect of the psychological inoculation. To address this possibility, we manually coded whether specific climate disinformation statements are thematic matches with one of the different psychological inoculations (see Table SM-1 in the Supplementary Information). To compare the effectiveness of the psychological inoculation between matching and unmatching climate disinformation statements, we will analyze changes in affect towards climate mitigation actions during the disinformation provision with four additional multilevel models, one for each psychological inoculation where we could identify at least one thematic match. We will specify four random effects: a slope per trial, an intercept per participant, an intercept per climate disinformation

statement, and an intercept per country. We will specify as fixed effects: condition (factor, two levels, specific psychological inoculation and passive disinformation control), trial (continuous variable, from 1 to 20), and the interaction between “thematic match” (factor, two levels, matching and not matching) and condition. If a thematic match between climate disinformation statements and specific psychological inoculations does indeed increase the protective effects of the inoculation, we would expect the interaction to be significant, and the simple slopes to highlight a significant difference between thematically matching versus thematically non-matching climate disinformation statements in the inoculation condition, so that the difference in affect would be smaller for climate disinformation statements that are thematic matches of the psychological inoculation.

The manual coding of the thematic matching between psychological inoculations and climate disinformation statements is available in the newly created Supplementary Information file (pp. 1):

Table SM-1 – Summary of psychological inoculations potential thematic match with climate disinformation statements and truth discernment items.

Inoculation	Matching climate disinformation statement(s)
Scientific consensus	Science_8; Science_10
Trust in scientists	Science_4; Science_5; Science_6
Transparent communication	Action_8; Action_10
Moralization of climate action	Action_4; Action_5; Action_9
Accuracy	
Positive emotion	

Should the reviewer spot additional thematic matches between the psychological inoculations and the climate disinformation statements, we would be happy to add them to the table and the related analyses.

Moreover, disentangling thematically matched from thematically non-matched misinformation statements will allow to test for a different interpretation in case our H1a is not supported: Should a main effect of the inoculation not be supported by the analyses centered around H_{1A}, we will be able to test if, for the inoculations with at least one matching climate disinformation statement, there are significant differences between the passive disinformation control condition and the specific psychological inoculation within matching climate disinformation statements. We have now added this further check in the Design Table for Question RQ₁ (pp. 56-58):

RQ₁: Do the six inoculations protect participants affect towards climate action against twenty real climate disinformation statements?	H_{1A}: Compared to participants in the passive disinformation control condition, participants who have received one of the inoculations will express more positive affect towards climate action after receiving all the twenty climate disinformation statements. H_{1B}: Compared to participants in the passive disinformation control condition, participants who have received one of the inoculations will express more positive affect towards climate action after each trial presenting a single climate disinformation statement.	We identified the required sample of N = 5964 (i.e., sample without the pure control condition), a-priori, with G*Power (Version 3.0), in order to have 95% power to detect a difference between any intervention condition and the control condition of $\eta^2 = 0.20$ in a one-tailed t-test with $\alpha = .005$, for both main hypotheses. We selected the smallest effect size of interest from the lower bound of the confidence interval of the meta-analytically identified effect size of fact-checking interventions on political topics^[165];	Hypothesis H_{1A}: We will compare affect toward climate action at the end of the intervention (i.e., after the twentieth disinformation statement) of the participants in the passive disinformation control condition with the participants in each inoculation condition, separately, with a one-tailed independent-sample t-test with a corrected to .005. Hypothesis H_{1B}: We will analyze affect toward climate action during the disinformation provision with a multilevel model. We will specify three	Hypothesis H_{1A}: Should any of the contrasts between the control condition and the inoculations be not significant, we will first visually inspect the affect curve of the twenty measurements of affect across the processing of the twenty climate disinformation statements, to visually identify ceiling or flooring effects for the intervention conditions. After confirming the lack of ceiling/flooring effects, we will test whether the difference between the passive disinformation control condition and the inoculation of interest is smaller than our smallest effect size of interest ($\delta = 0.20$) with equivalence testing. Should any of the equivalence testing be significant at $.005 < p < .05$, we will interpret the result as suggestive evidence that the inoculation of interest does not offer better protection than traditional fact-checking. Should any of the equivalence testing be significant at $p < .005$, we will interpret the result as confirmatory evidence that the inoculation of interest does not offer better protection than traditional fact-checking.
--	--	---	---	---

we reasoned that a new misinformation intervention would be practically interesting if and only if it has an effect that is larger than already available benchmark interventions such as fact-checks.

random effects: intercept for participant; intercept for the internal numbering of the climate disinformation statements, to account for the variance associated with each disinformation statement^[185]; intercept for country (factor, alphabetically coded), to account for the variance associated with each country^[185]. We will specify as fixed effects: condition (factor, seven levels, dummy coded with 0 = passive disinformation control condition as the reference contrast), trial (continuous variable, from 1 to 20), and the two-way interactions of trial with condition to test hypothesis H_{1B}.

Should H_{1A} not be supported, we will test if, for the inoculations with at least one thematically matching climate disinformation statement (see Table SM-1 in Supplementary Information), there are significant differences between the passive disinformation control condition and the specific psychological inoculation only within matching climate disinformation statements. This comparison will be carried out with the multilevel models described in the final analysis of the Analysis plan section, expecting the interaction between condition and thematic matching to be significant, and for the simple slopes to show a significant difference between the specific psychological inoculation and the passive disinformation control condition only against matching climate disinformation statements (interaction simple slope α corrected to .125; see the Analysis plan section pp. 34-35).

Hypothesis H_{1B}: Should the trial main effect not be significant, we will first visually inspect the affect curve of the twenty measurements of affect across the processing of the twenty climate disinformation statements, to visually identify ceiling or flooring effects for the effects of the climate disinformation statements. Should the interaction between trial and condition be not significant but the hypothesis H_{1A} confirmed, we will conduct an additional one-tailed independent sample t-test between the passive disinformation control condition and the inoculation of interest, with the affect towards climate action after the first climate disinformation statement as the dependent

				variable. Should this t-test be significant, we will conclude that the inoculation of interest influences affect towards climate action only immediately, and not after each single climate disinformation statement. Should the equivalence test for hypothesis H_{1A} be significant, we will conclude that our data provides “suggestive evidence that the inoculation does not offer a better protection than traditional fact-checking”.
--	--	--	--	--

Decision Letter, second revision:

20th April 2023

Dear Mr Spampatti,

Thank you once again for submitting your revised Stage 1 Registered Report, entitled "Psychological inoculation strategies to fight climate disinformation across 12 countries." Everything is in order and I am delighted to say that we can offer acceptance in principle. You may progress to Stage 2 and complete the study as approved.

As you know, a condition of in-principle-acceptance is that the authors agree to deposit their Stage 1 accepted protocol in a repository, either publicly or under embargo until Stage 2 acceptance and publication. We are very keen to showcase our in-principle accepted protocols, so that our readers, reviewers, and potential authors can gain insight into the requirements of the format as well as an idea of the types of projects that are suitable for publication in Nature Human Behaviour. We have set up a space on figshare (https://springernature.figshare.com/registered-reports_NHB) to host all of our in-principle accepted protocols, which can either be made public or kept under embargo until Stage 2 acceptance (depending on author preference). This gives you the opportunity to have your work publicly associated with Nature Human Behaviour, and of course we will be very pleased to showcase your report if you agree to share it publicly.

Depositing the work on our figshare space does not preclude deposition of your Stage 1 protocol on

other depositories – your protocol can also be posted on OSF, Dataverse, Dryad or any other public repository of your choice. You also do not need to do anything – if you agree with posting your protocol on our figshare space, we will upload your protocol on your behalf and either set it public or place it under embargo, depending on your choice. Your protocol will be licensed under a CC BY license (Creative Commons Attribution 4.0 International License). The CC BY license allows for maximum dissemination and re-use of open access materials and is preferred by many research funding bodies. Under this license users are free to share (copy, distribute and transmit) and remix (adapt) the contribution including for commercial purposes, providing they attribute the contribution in the manner specified by the author or licensor (read full legal code:

<http://creativecommons.org/licenses/by/4.0/legalcode>) Please note that any use of <https://springernature.figshare.com> will be subject to the Figshare terms of use. Figshare has the right to enforce these terms and conditions where applicable. Use of third party services and sites will be subject to the relevant terms of use and will apply if we act on your behalf in this regard. Do let me know if you would like to take up this option or if you have any questions regarding the protocol deposition requirement.

Following completion of your study, we invite you to resubmit your paper for peer review as a Stage 2 Registered Report. Please note that your manuscript can still be rejected for publication at Stage 2 if the Editors consider any of the following to hold:

- The results were unable to test the authors' proposed hypotheses by failing to meet the approved outcome-neutral criteria
- The authors altered the Introduction, rationale, or hypotheses, as approved in the Stage 1 submission
- The authors failed to adhere closely to the registered experimental procedures
- Any post hoc (unregistered) analyses were either unjustified, insufficiently caveated, or overly dominant in shaping the authors' conclusions
- The authors' conclusions were not justified given the data obtained

We encourage you to read the complete guidelines for authors concerning Stage 2 submissions at <https://www.nature.com/nathumbehav/registeredreports>. Please especially note the requirements for protocol deposition, data sharing, and that withdrawing your manuscript will result in publication of a Retracted Registration.

In recognition of the time and expertise our reviewers provide to Nature Human Behaviour's editorial process, we would like to formally acknowledge their contribution to the external peer review of your manuscript entitled "Psychological inoculation strategies to fight climate disinformation across 12 countries". For those reviewers who give their assent, we will be publishing their names alongside the published article.

When you are ready, please use the following link to access your home page and submit your Stage 2 Registered Report:

[REDACTED]

*This url links to your confidential homepage and associated information about manuscripts you may have submitted or be reviewing for us. If you wish to forward this e-mail to co-authors, please delete

this link to your homepage first.

We expect your Stage 2 Registered Report to be submitted by the date specified in your latest cover letter. If unforeseen circumstances prevent submission by that date, please contact us as soon as possible to discuss any changes to the submission time-frame.

Thank you again for offering us this work and we look forward to receiving your Stage 2 Registered Report.

Yours sincerely,

Samantha Antusch

Samantha Antusch, PhD
Senior Editor
Nature Human Behaviour

Decision Letter, second revision:

29th August 2023

Dear Mr Spampatti,

Thank you once again for submitting your Stage 2 Registered Report, entitled "Psychological inoculation strategies to fight climate disinformation across 12 countries," and for your patience during the re-review process.

Your manuscript has now been evaluated by Reviewers 1,3,and 4 from the previous rounds of review, whose comments are included at the end of this letter. In the light of our reviewers' advice, we are pleased to inform you that we will be able accept your Stage 2 manuscript, pending revisions to address reviewer comments and editorial requests.

To guide the scope of the revisions, the editors discuss the referee reports in detail within the team, including with the chief editor, with a view to (1) identifying key priorities that should be addressed in revision and (2) overruling referee requests that are beyond the scope of Stage 2 Registered Reports.

I. Reviewer 1 and Reviewer 3 make important points about the differences between preregistered and exploratory analyses. In your revised manuscript, please clearly distinguish between preregistered and exploratory analyses, and motivate all exploratory analyses. We ask that you also follow Reviewer 3's advice and exert caution when interpreting the results of exploratory analyses.

II. Reviewer 4 suggests that you conduct various exploratory analyses. Per our Registered Report

policy, it is entirely up to the authors whether they undertake at Stage 2 additional exploratory analyses that are recommended by the reviewers. Although we feel that these additional exploratory analyses will strengthen your work and its interpretation, it is at your discretion whether to carry them out or not.

One of the main reasons for delays in eventual acceptance is failure to fully comply with editorial policies and formatting requirements. To assist you with finalizing your manuscript for publication, I attach a checklist that lists all of our editorial policies and formatting requirements.

Please attend to *every item* in the checklist and upload a copy of the completed checklist with your submission. I have highlighted in the checklist items that require your attention. I also mention here a few points that are frequently missed and can cause delays:

1) Please insert the following Protocol Registration information in your manuscript:

"The Stage 1 protocol for this Registered Report was accepted in principle on 2023-04-20. The protocol, as accepted by the journal, can be found at <https://figshare.com/s/f431f656b53ec90396c0>."

2) Ensure that all corresponding authors have linked their ORCID to their account on our online manuscript handling system. This is very frequently missed and invariably causes delays in formal acceptance.

3) Ensure that you provide all of the materials requested in the attached checklist and below with your final submission.

Nature Human Behaviour offers a transparent peer review option for new original research manuscripts submitted from 1st December 2019. We encourage increased transparency in peer review by publishing the reviewer comments, author rebuttal letters and editorial decision letters if the authors agree. Such peer review material is made available as a supplementary peer review file. **Please state in the cover letter 'I wish to participate in transparent peer review' if you want to opt in, or 'I do not wish to participate in transparent peer review' if you don't.** Failure to state your preference will result in delays in accepting your manuscript for publication.

Please note: we allow redactions to authors' rebuttal and reviewer comments in the interest of confidentiality. If you are concerned about the release of confidential data, please let us know specifically what information you would like to have removed. Please note that we cannot incorporate redactions for any other reasons. Reviewer names will be published in the peer review files if the reviewer signed the comments to authors, or if reviewers explicitly agree to release their name. For more information, please refer to our FAQ page.

We hope to hear from you within [ENTER TIME PERIOD]; please let us know if the revision process is likely to take longer.

To submit your revised manuscript, you will need to provide the following:

- Cover letter
- Point-by-point response to the reviewers (if applicable)
- Manuscript text (not including the figures) in .docx or .tex format

- Individual figure files (one figure per file)
- Extended Data & Supplementary Information, as instructed
- Reporting summary
- Editorial policy checklist
- Third-party rights table (if applicable)
- Suggestions for cover illustrations (if desired)

Consortia authorship:

For papers containing one or more consortia, all members of the consortium who contributed to the paper must be listed in the paper (i.e., print/online PDF). If necessary, individual authors can be listed in both the main author list and as a member of a consortium listed at the end of the paper. When submitting your revised manuscript via the online submission system, the consortium name should be entered as an author, together with the contact details of a nominated consortium representative. See <https://www.nature.com/authors/policies/authorship.html> for our authorship policy and <https://www.nature.com/documents/nr-consortia-formatting.pdf> for further consortia formatting guidelines, which should be adhered to prior to acceptance.

Forms:

Nature Human Behaviour has now transitioned to a unified Rights Collection system which will allow our Author Services team to quickly and easily collect the rights and permissions required to publish your work. Once your paper is accepted, you will receive an email in approximately 10 business days providing you with a link to complete the grant of rights. If you choose to publish Open Access, our Author Services team will also be in touch at that time regarding any additional information that may be required to arrange payment for your article.

[REDACTED]

With best regards,

Samantha Antusch

Samantha Antusch, PhD
Senior Editor
Nature Human Behaviour

Reviewer #1:

Remarks to the Author:

Overall, the authors accurately reported the results of the pre-specified analyses and provide a valuable contribution to the literature- as interested parties have good evidence of a particular tactic

that does not work well. There are a few small instances where the exploratory versus preregistered analyses are a bit unclear, described below, and I do recommend a bit of additional explanation about at least one set of exploratory analyses. However, I see no substance issues beyond the minor recommendations, below.

As a matter of personal policy, I sign my reviews:

David Mellor, PhD

Director of Policy, Center for Open Science

<https://orcid.org/0000-0002-3125-5888>

Minor recommendations:

If accepted, please change the private, view only links to regular, public links (eg https://osf.io/m58zx/?view_only=95fd430f4b7e4ee99c9c8b472e31d6b3 to <https://osf.io/m58zx> after making that project public)]

The following statement is worded in a slightly odd way. "We exploratorily tested whether the psychological inoculations were able to protect participants' affect towards climate mitigation action against the influence of the very first climate disinformation statement only." I recommend structuring this sentence with a very brief justification for what motivated the exploratory analysis (whether it was simply noticing a trend that you wanted to investigate or because of some other reason, such as primacy bias for this exploratory analysis). For this one it could be: "We thought that perhaps the primacy bias could influence the first statement more than subsequent statements, so we conducted an exploratory analysis..." or "We noticed a seemingly stronger effect in the first statement so conducted an exploratory..."

Lines 427-430: It's possible that this explanation for the exploratory analyses could go in the discussion section.

Lines 344-345: Several exploratory analysis are described here, and it is unclear if the next set of results (starting in line 351) were also exploratory. Since this whole paragraph is listed as "preregistered analyses", I'd recommend removing those in 344 or continuing the convention with paragraph headers describing the analyses.

359-364: The authors do a good job of explaining the rationale for these additional tests (likewise for lines 400-404).

481-483: I'd also explain the value of knowing what does not work well- a major benefit of your findings shows the relevant community what will be insufficient (but at least not backfiring) interventions, and that is valuable information to have.

485-488: I anticipate that this possible strategy would be most effective and would look forward to a follow up study that establishes this as a condition.

Reviewer #3:

Remarks to the Author:

Congratulations to the authors on completing this important study. The apparent failure of the inoculation strategies is disappointing of course, but it is uplifting to see the diligent reporting of informative null results — this will move the field forward.

I've read the Stage 2 report and I'm generally satisfied — I have just one suggestion:

The authors carefully distinguish between preregistered and non-preregistered (exploratory) analyses in their reporting; however, it is also important that the status of the analyses influences one's confidence in the results. Specifically, a non-preregistered analysis is more susceptible to bias, and should thus be interpreted with greater caution (for details see Hardwicke & Wagenmakers, 2023; <https://rdcu.be/c4h3a>). Note that the risk of bias here is inherent to the data-dependent nature of the analyses and does not imply any kind of questionable research practices.

By my reading, the manuscript seems to imply equal confidence in the the results of the preregistered and non-preregistered analyses. For example, in this part of the discussion, the results exploratory analyses seem to be given equal weight (they are even mentioned first): "Exploratory analyses indicated that the psychological inoculations were able to counteract the negative impact of the first climate disinformation statement, but we found no evidence that the six psychological inoculations had a protective effect against the twenty climate disinformation statements, with the exception of the accuracy inoculation that significantly protected truth discernment ability."

In my view, it should be made clearer that the results of the non-preregistered analyses are more tentative. This can be done by prioritising the results of the preregistered analyses, explicitly stating that non-preregistered results have higher risk of bias and using hedging language when reporting non-preregistered results.

Reviewer #4:

Remarks to the Author:

I was interested to learn about the results of this ambitious study. Somewhat disappointingly, the findings of inoculation seem largely null in this context. Also of note, there were no backfire effects and no increase in response bias either.

I was glad to see the equivalence testing and appreciated the additional matched-by-disinformation-type analysis in the supplement.

However, call me old school but I was wondering if the authors could do something fairly simple for me? The multi-level model contains interaction terms (match by type) as well as many other variables and covariates. It's hard for me to understand the raw data in these complex models.

Could you not, e.g., do the model where you only have 1 variable (e.g. condition = scientific consensus inoculation vs control) and the DV would be belief in the scientific consensus disinformation. Basically a simple main effect where the intervention is matched by disinfo type (in isolation). What I'm trying to gauge is the simplest possible test on the raw data to understand whether inoculating people against a specific form of disinformation (consensus, trust) confers protection against that disinformation. I realize you attempted to do this, but with the power issues and many other variables in the model that could eat up the variance, it's not clear to me what the raw data is showing in terms of the experimental effect. You could also just show some plots of the raw data collapsing across condition (1 = inoculation X, 0 = control) and on the Y-axis we have belief in the specific disinfo - in a big plot for all six inoculations.

I know this is not pre-registered so there's no need to attach any value to it, I just think it would be interesting to see for the reader in the supplement. If the raw data shows absolutely nothing - which coincides with the MLM findings, you have a very strong case. In the event they diverge, you clearly should still interpret the pre-registered MLM but I'm a big fan of inspecting raw data patterns.

I'd also like to see some plots of the SDT analyses, e.g., the AUC curves?

I recommend this paper to be accepted (which is up to the editors of course) but as you did everything you said you would do in the RR I have no objections so my comments are just for a very minor brief addition to make sure we correctly conclude there were no experimental effects of inoculation, even in the raw data (not just the complex model with many other effects present).

p.s. one final remark, could it not be the case that the inoculations were simply not effective because the treatments were too weak? The authors do already discuss active vs passive variations, but e.g., in existing inoculation treatments (see Cook et al., 2017; van der Linden et al., 2017), the inoculation was a 600 word essay with graphs etc, how does that compare to the treatments in the current study?

Thanks and congrats on this important work

Author Rebuttal, second revision:

Reviewer #1:

Remarks to the Author:

Overall, the authors accurately reported the results of the pre-specified analyses and provide a valuable contribution to the literature- as interested parties have good evidence of a particular tactic that does not work well. There are a few small instances where the exploratory versus preregistered analyses are a bit unclear, described below, and I do recommend a bit of additional explanation about at least one set of exploratory analyses. However, I see no substance issues beyond the minor recommendations, below.

As a matter of personal policy, I sign my reviews:

David Mellor, PhD

Director of Policy, Center for Open Science

<https://orcid.org/0000-0002-3125-5888>

Minor recommendations:

1. If accepted, please change the private, view only links to regular, public links (eg https://osf.io/m58zx/?view_only=95fd430f4b7e4ee99c9c8b472e31d6b3 to <https://osf.io/m58zx> after making that project public)]

The following statement is worded in a slightly odd way. “We exploratorily tested whether the psychological inoculations were able to protect participants’ affect towards climate mitigation action against the influence of the very first climate disinformation statement only.” I recommend structuring this sentence with a very brief justification for what motivated the exploratory analysis (whether it was simply noticing a trend that you wanted to investigate or because of some other reason, such as primacy bias for this exploratory analysis). For this one it could be: “We thought that perhaps the primacy bias could influence the first statement more than subsequent statements, so we conducted an exploratory analysis...” or “We noticed a seemingly stronger effect in the first statement so conducted an exploratory...”

Lines 427-430: It’s possible that this explanation for the exploratory analyses could go in the discussion section.

Thank you for pointing us to this necessary clarification. We now explain that the additional exploratory analysis was motivated by the fact that psychological inoculations have been commonly tested against one single disinformation piece only (van der Linden et al., 2017; Cook et al., 2017). We therefore considered that a more comprehensive (and, it could be argued, fairer) test of the inoculations in our study should have also included this test. Thus, while the overall preregistered findings of this study seem to suggest that the particular psychological inoculations used in this study are generally ineffective, with the exploratory analysis we illustrate that, when using more commonly used experimental criteria of “treatment success”, all inoculations, with the exception of the transparent communications inoculations, were effective against a single climate disinformation statement. We now make our motivation for the exploratory test explicit in the Results section of the manuscript (pp. 13):

As psychological inoculations are traditionally tested against a single, not multiple, disinformation statement^{38; 67}, we exploratorily tested whether the psychological inoculations were able to protect participants’ affect towards

climate mitigation action against the influence of the very first climate disinformation statement only.

- 2. Lines 344-345: Several exploratory analysis are described here, and it is unclear if the next set of results (starting in line 351) were also exploratory. Since this whole paragraph is listed as “preregistered analyses”, I’d recommend removing those in 344 or continuing the convention with paragraph headers describing the analyses.**

We thank the reviewer for the suggestion. In accordance, the sentences, together with a justification for conducting the analyses, have been moved to the section “Effect of the inoculations on truth discrimination – exploratory analyses” (pp. 16-17).

We conducted additional exploratory analyses because recent studies found that psychological inoculations may increase participants’ bias towards judging any statement as false^[69, 173]. First, decomposing the effect of the accuracy inoculation across four types of climate statements (climate support versus climate delay, true or false) showed that accuracy-inoculated participants were better at discerning false statements delaying climate action ($t(1743.74)=2.204$, 95% CI[0.02, 0.27], $p=.024$, $\delta=0.11$), but not the other types of statements, i.e., true statements delaying climate action ($t(1745)=3.557$, 90% CI[-0.16, 0.07], $p<.001$), true statements supporting climate action ($t(1744.07)=2.614$, 90% CI[-0.19, 0.01], $p=.005$), false statements supporting climate action ($t(1744.69)=4.042$, 90% CI[-0.10, 0.09], $p<.001$).

Second, we applied Signal Detection Theory^[172] to further scrutinize performance in the truth discrimination task. In brief, Signal Detection Theory posits that stimulus detection is contingent upon people's discriminant ability and their overall response bias towards reporting all (dis)information as true or as false.

- 3. 359-364: The authors do a good job of explaining the rationale for these additional tests (likewise for lines 400-404).**

Thank you very much for the positive comment.

- 4. 481-483: I'd also explain the value of knowing what does not work well- a major benefit of your findings shows the relevant community what will be insufficient (but at least not backfiring) interventions, and that is valuable information to have.**

We thank the reviewer for pointing out the usefulness of our findings. The paragraph before the limitation paragraph is meant to do exactly that - explaining that the inoculations may only have a limited effect and redirecting readers to more promising interventions (pp. 20):

At the applied level, our findings indicate that psychological inoculations have no unintended consequences – e.g., backfire effects^[cf. 69; 173] – but may have no or only very limited capacity to protect against multiple disinformation related to climate change^[69; cf. 180]. Even the strategy of reinforcing the inoculated protection with follow-up “booster shots”^[70; 105; 181] may be untenable, as our evidence suggests that they would be needed after encountering a single climate disinformation already. Weighting the influence and discursive presence^[e.g., 29; 181-182] of climate disinformation with the partial effectiveness of

psychological inoculations, our findings join the recent discussion that behavioral science interventions do not seem to be efficacious enough to tackle systemic problems like climate disinformation by themselves^[59-60; 183-185]. Systemic interventions, such as content moderation, virality circuit breakers^[183], deplatforming^[181; 186], or changing online engagement metrics towards the accuracy of information^[187], may be better at curbing climate disinformation. However, systemic actions are enforceable only by the same platforms that might be incentivized to let climate disinformation spread^[187-192]. For this reason, both behavioral and systemic intervention approaches need to be further developed and applied, especially given the direct evidence provided here that climate disinformation drastically decreases climate-relevant judgments and behaviors.

We are unsure how to stress this point again without complicating the final paragraph. We meant the paragraph to end on a positive note, motivating researchers to develop better psychological inoculations in light of our findings and theoretical framework. We are of course open to suggestions on how to better implement the point raised by the reviewer.

- 5. 485-488: I anticipate that this possible strategy would be most effective and would look forward to a follow up study that establishes this as a condition.**

We hear you loud and clear!

Reviewer #3:

Remarks to the Author:

1. **Congratulations to the authors on completing this important study. The apparent failure of the inoculation strategies is disappointing of course, but it is uplifting to see the diligent reporting of informative null results — this will move the field forward.**

We thank the reviewer for the positive comments - we certainly hope that the work will have an impact!

I've read the Stage 2 report and I'm generally satisfied — I have just one suggestion:

2. **The authors carefully distinguish between preregistered and non-preregistered (exploratory) analyses in their reporting; however, it is also important that the status of the analyses influences one's confidence in the results. Specifically, a non-preregistered analysis is more susceptible to bias, and should thus be interpreted with greater caution (for details see Hardwicke & Wagenmakers, 2023; <https://rdcu.be/c4h3a>). Note that the risk of bias here is inherent to the data-dependent nature of the analyses and does not imply any kind of questionable research practices.**

By my reading, the manuscript seems to imply equal confidence in the results of the preregistered and non-preregistered analyses. For example, in this part of the discussion, the results exploratory analyses seem to be given equal weight (they are even mentioned first): "Exploratory analyses indicated that the psychological inoculations were able to counteract the negative impact of the first climate disinformation statement, but we found no evidence that the six psychological inoculations had a protective effect against the twenty climate disinformation statements, with the exception of the accuracy inoculation that significantly protected truth discernment ability."

In my view, it should be made clearer that the results of the non-preregistered analyses are more tentative. This can be done by prioritising the results of the preregistered analyses, explicitly stating that non-preregistered results have higher risk of bias and using hedging language when reporting non-preregistered results.

We wholeheartedly agree with the importance to distinguish between confirmatory and exploratory analyses. It was not our intention to treat these two types of analysis equally in our results section, but to provide additional information to the reader in order to help contextualize the study findings (in this case, the finding that the psychological inoculations may protect against one, but not multiple, instances of disinformation). We now emphasize more strongly the difference between the two analyses, hedging

the exploratory analyses and highlighting the confirmatory nature of the preregistered finding. The sentences now read (pp. 19):

Although exploratory analyses suggested that the psychological inoculations may have been able to counteract the negative impact of the first climate disinformation statement, we found no confirmatory evidence that the six psychological inoculations had a protective effect against the twenty climate disinformation statements, with the exception of the accuracy inoculation that significantly protected truth discernment ability.

Reviewer #4:

Remarks to the Author:

- 1. I was interested to learn about the results of this ambitious study. Somewhat disappointingly, the findings of inoculation seem largely null in this context. Also of note, there were no backfire effects and no increase in response bias either.**

I was glad to see the equivalence testing and appreciated the additional matched-by-disinformation-type analysis in the supplement.

Thank you very much for the positive assessment of our study. We equally were somewhat surprised about the null effects of this study, but we trust the findings will nevertheless provide informative results for the field at large.

- 2. However, call me old school but I was wondering if the authors could do something fairly simple for me? The multi-level model contains interaction terms (match by type) as well as many other variables and covariates. It's hard for me to understand the raw data in these complex models.**

Could you not, e.g., do the model where you only have 1 variable (e.g. condition = scientific consensus inoculation vs control) and the DV would be belief in the scientific consensus

disinformation. Basically a simple main effect where the intervention is matched by disinfo type (in isolation). What I'm trying to gauge is the simplest possible test on the raw data to understand whether inoculating people against a specific form of disinformation (consensus, trust) confers protection against that disinformation. I realize you attempted to do this, but with the power issues and many other variables in the model that could eat up the variance, it's not clear to me what the raw data is showing in terms of the experimental effect. You could also just show some plots of the raw data collapsing across condition (1 = inoculation X, 0 = control) and on the Y-axis we have belief in the specific disinfo - in a big plot for all six inoculations.

I know this is not pre-registered so there's no need to attach any value to it, I just think it would be interesting to see for the reader in the supplement. If the raw data shows absolutely nothing - which coincides with the MLM findings, you have a very strong case. In the event they diverge, you clearly should still interpret the pre-registered MLM but I'm a big fan of inspecting raw data patterns.

We thank the reviewer for proposing further analyses to test the preregistered hypotheses. Responding to the last suggestion first, we would like to avoid plotting these specific results for ethical reasons. Succinctly, we do not want to publicize easily interpretable figures showing which climate disinformation statements are more successful in increasing peoples' doubt about anthropogenic climate change. This information would be more useful to actors who spread disinformation rather than those who try to prevent disinformation.

However, we have followed the suggestion to test the simplest multilevel model possible to test whether there are between-participants differences within the affect updating on the climate disinformation statements matching the target inoculation (vs control; as a reminder, only four psychological inoculations had some thematic matches with one or more climate disinformation statements – see Table SI-2). As can be seen in the tables below, there is no evidence of an effect of the scientific consensus inoculation, even given the simpler model (see Table RE-1), nor of the other psychological inoculations (when accounting for multiple comparisons, $\alpha=0.0125$; Tables RE-2-4).

Table RE-1– Exploratory multilevel model for affect towards climate mitigation action between the passive control condition and the scientific consensus inoculation condition when processing thematically matched climate disinformation statements.

95% Confidence Intervals

Predictor	Estimate	SE	t-value	Lower	Upper	p
Intercept	59.11	0.81	72.740	57.52	60.70	<.2 ^{e-16}
Condition: Scientific consensus	1.96	1.16	1.695	-0.31	4.24	.09

Table RE-2 – Exploratory multilevel model for affect towards climate mitigation action between the passive control condition and the trust in scientists inoculation condition when processing thematically matched climate disinformation statements.

Predictor	Estimate	SE	t-value	95% Confidence Intervals		p
				Lower	Upper	
Intercept	57.58	0.81	70.931	55.99	59.17	<.2 ^{e-16}
Condition: Trust in scientists	2.74	1.15	2.388	0.49	4.99	.017

Table RE-3 – Exploratory multilevel model for affect towards climate mitigation action between the passive control condition and the transparent communications inoculation condition when processing thematically matched climate disinformation statements.

Predictor	Estimate	SE	t-value	95% Confidence Intervals		p
				Lower	Upper	
Intercept	58.35	0.85	68.790	56.69	60.01	<.2 ^{e-16}

Condition: Transparent communications	-0.09	1.20	1.695	-2.44	2.27	.94
---	-------	------	-------	-------	------	-----

Table RE-4 – Exploratory multilevel model for affect towards climate mitigation action between the passive control condition and the moralization inoculation condition when processing thematically matched climate disinformation statements.

Predictor	Estimate	SE	t-value	95% Confidence Intervals		p
				Lower	Upper	
Intercept	58.54	0.80	72.988	56.97	60.11	<.2 ^{e-16}
Condition: Moralization	2.15	1.13	1.912	-0.05	4.36	.056

As this was an a-posteriori analysis, we opt to maintain it in the publicly available response to reviewers, rather than the manuscript.

3. I'd also like to see some plots of the SDT analyses, e.g., the AUC curves?

Thank you for the suggestion. In the context of the manuscript, we would prefer to focus on the preregistered analyses. As mentioned in the response to the editors' comments, all data and scripts are openly available, and we welcome preregistered and rigorous secondary analyses of this dataset that may uncover additional findings and result in follow-up confirmatory studies.

I recommend this paper to be accepted (which is up to the editors of course) but as you did everything you said you would do in the RR I have no objections so my comments are just for a very minor brief addition to make sure we correctly conclude there were no experimental effects of inoculation, even in the raw data (not just the complex model with many other effects present).

4. p.s. one final remark, could it not be the case that the inoculations were simply not effective because the treatments were too weak? The authors do already discuss active vs passive variations, but e.g., in existing inoculation treatments (see Cook et al., 2017; van der Linden et al., 2017), the inoculation was a 600 word essay with graphs etc, how does that compare to the treatments in the current study?

We thank the reviewer for asking for this final specification. We would like to raise the point that the inoculations were not, in our opinion, ineffective: first, applying the standards of the papers mentioned by the reviewer, the inoculations *were* successful as they negated the negative effects of the first climate disinformation (while we freely admit the caveat of this being an exploratory analysis). It is rather the exposure to a stronger stimulus of multiple climate disinformation statements that reduced the effect of the psychological inoculations to a smaller-than-predicted effect size. It was currently unknown how the existing inoculation treatments would fare against this more severe test of multiple climate disinformation statements. Second, the point prediction of the effect sizes of the psychological inoculations presented in this study hover around the efficacy of recent, more detailed, broad-spectrum psychological inoculations (Roozenbeek et al., 2022) and active inoculations (Modirrousta-Galian et al., 2023). As we predicted a larger effect in our Stage 1 manuscript than the smaller ones described in these papers, we may not have had the statistical power to detect a potentially present protective effect of this size. Therefore, our conclusion is rather the effects of the psychological inoculations, if present, is significantly smaller than $\bar{d}=0.20$. We do however agree that future inoculations using the same theoretical components could benefit from multimodal delivery or active treatment paradigms to increase their effectiveness. Overall, we hope that our study will catalyze research using this more severe experimental paradigm – i.e., multiple consecutive (climate) disinformation statements – and a bigger sample size, to push forward our knowledge about psychological inoculations.

Thanks and congrats on this important work

Thank you for the thoughtful review!

Final Decision Letter:

Dear Mr Spampatti,

We are pleased to inform you that your Registered Report "Psychological inoculation strategies to fight climate disinformation across 12 countries", has now been accepted for publication in *Nature Human Behaviour*.

Please note that *Nature Human Behaviour* is a Transformative Journal (TJ). Authors may publish their research with us through the traditional subscription access route or make their paper immediately open access through payment of an article-processing charge (APC). Authors will not be required to make a final decision about access to their article until it has been accepted. Find out more about Transformative Journals

With best regards,

Samantha Antusch

Samantha Antusch, PhD
Senior Editor
Nature Human Behaviour